# Multidimensional chromatin profiling of zebrafish pancreas to uncover and investigate disease-relevant enhancers

Renata Bordeira-Carriço[1,2,9], Joana Teixeira [1,2,3,9], Marta Duque[1,2,3,9], Mafalda Galhardo [1,2,4], Diogo Ribeiro[1,2], Rafael D. Acemel[5], Panos. N. Firbas[5], Juan J. Tena [5], Ana Eufrásio[1,2,3], Joana Marques[1,2], Fábio J. Ferreira [1,2,6], Telmo Freitas[1,2], Fátima Carneiro [7,8], José Luís Goméz-Skarmeta[5] & José Bessa [1,2✉]

The pancreas is a central organ for human diseases. Most alleles uncovered by genome-wide association studies of pancreatic dysfunction traits overlap with non-coding sequences of DNA. Many contain epigenetic marks of cis-regulatory elements active in pancreatic cells, suggesting that alterations in these sequences contribute to pancreatic diseases. Animal models greatly help to understand the role of non-coding alterations in disease. However, interspecies identification of equivalent cis-regulatory elements faces fundamental challenges, including lack of sequence conservation. Here we combine epigenetic assays with reporter assays in zebrafish and human pancreatic cells to identify interspecies functionally equivalent cis-regulatory elements, regardless of sequence conservation. Among other potential disease-relevant enhancers, we identify a zebrafish *ptf1a* distal-enhancer whose deletion causes pancreatic agenesis, a phenotype previously found to be induced by mutations in a distal-enhancer of *PTF1A* in humans, further supporting the causality of this condition in vivo. This approach helps to uncover interspecies functionally equivalent cis-regulatory elements and their potential role in human disease.

[1] Instituto de Biologia Molecular e Celular (IBMC), Universidade do Porto, Porto 4200-135, Portugal. [2] Instituto de Investigação e Inovação em Saúde (i3S), Universidade do Porto, Porto 4200-135, Portugal. [3] Doctoral program in Molecular and Cell Biology (MCbiology), ICBAS—Instituto de Ciências Biomédicas Abel Salazar, Universidade do Porto, Porto, Portugal. [4] Instituto de Ciências, Tecnologias e Agroambiente (CIBIO), Universidade do Porto, 4051-401 Porto, Portugal. [5] Centro Andaluz de Biología del Desarrollo (CABD), Universidad Pablo de Olavide, Sevilla 41013, Spain. [6] Doctoral program in Areas of Basic and Applied Biology (GABBA), Instituto de Ciências Biomédicas Abel Salazar (ICBAS), Universidade do Porto, Porto 4050-313, Portugal. [7] Departamento de Patologia, Faculdade de Medicina da Universidade do Porto (FMUP), Porto 4200-319, Portugal. [8] Instituto de Patologia e Imunologia Molecular da Universidade do Porto (IPATIMUP), Porto 4200-465, Portugal. [9] These authors contributed equally: Renata Bordeira-Carriço, Joana Teixeira, Marta Duque. ✉email: jose.bessa@ibmc.up.pt

The mechanisms that tightly control transcription are essential for organ function. The transcriptional regulation of genes is controlled by non-coding cis-regulatory elements (CREs) spread over large genomic distances[1]. Genome-Wide Association Studies (GWAS) have identified many non-coding disease-associated alleles that have a hereditary component and overlap with CREs epigenetic signatures, suggesting that the disruption of CREs may be one of the genetic bases of human disease. This is the case of some pancreatic diseases, such as pancreatic cancer and diabetes[2–6], that have a heavy societal burden, with incidence and death rates increasing worldwide[7–12]. Many previous studies demonstrated an enrichment of diabetes-associated variants in adult human islet enhancers[2,5,13–16], corroborating the hypothesis of pancreatic diseases being caused by alterations in CREs. Likewise, experimental in vivo and in vitro enhancer reporter assays also showed that specific islet enhancer variants correlate with altered regulatory functions[14,17–20]. Studies of the role of CREs' mutations in the development of pancreatic diseases using in vivo models would provide invaluable insight given the complex regulatory networks involved. However, evidences from in vivo models of the role of CREs' mutations in the development of pancreatic diseases are still scarce[21–23].

The zebrafish is a vertebrate model suitable for genetic manipulation[24], with a pancreas that shares many similarities with the human pancreas, including similar transcription factors (TFs) and genetic networks of pancreatic development and function[25,26]. Thus, the zebrafish is a suitable in vivo model to validate causal regulatory variants. Yet, the identification of interspecies functionally equivalent CREs faces unsolved fundamental challenges, such as low conservation of interspecies non-coding sequences[27] and, for the minority of CREs whose sequence is conserved, their fast-evolving functionality[28]. Indeed, although sequence conservation of non-coding sequences has successfully been used to find enhancers, many with interspecies orthologous identities[29,30], it has also been demonstrated to be insufficient for identifying all enhancers within a genome and between species[31,32]. To bypass these limitations, in this work we profiled the chromatin state of zebrafish pancreas cells and chromatin interaction points. We were able to accurately identify zebrafish pancreatic enhancers and, by comparisons with similar human datasets, we predicted functionally equivalent pancreatic enhancers. These findings revealed a previously unidentified human enhancer in the landscape of the tumour suppressor ARID1A[33,34], with a potential role in the susceptibility to pancreatic cancer. Additionally, we explored the regulatory landscape of PTF1A, known to contain a human distal enhancer whose deletion leads to pancreatic agenesis/hypoplasia[35–38], and found a zebrafish distal ptf1a enhancer that contains similar regulatory information to its human counterpart. We further demonstrated its functional equivalency by showing that its ablation induces pancreatic agenesis, explained by a reduction in the pancreatic progenitor domain early in development. Taken together, the multidimensional chromatin profiling used here allowed the establishment of previously unknown functional connections between human and zebrafish enhancers. These bridges between different species are invaluable for the prediction of new disease-relevant enhancers and the study of their role in human disease.

## Results

### Zebrafish putative pancreatic enhancers share developmental roles.
When comparing the basic structure of the human and zebrafish adult pancreas we observed that the organ structure is analogous between the two species (Fig. 1a). We further extended this comparison to the cellular composition of the main cell types of the pancreas between zebrafish, mouse[39,40] and human[39–42],

and found that the predominance of the major cellular types is maintained in these three vertebrates (Supplementary Fig. 1). Because of these extended similarities between the zebrafish and mammal pancreas, the zebrafish has been used as a model to study pancreatic diseases[25,43]. Furthermore, these similarities hint at the existence of shared genetic networks that operate, likely through equivalent sets of CREs, in these three species. Thus, we explored the chromatin state and chromatin interaction points of zebrafish whole pancreas, to gather information about endocrine and exocrine cells, and compared it to human datasets. To identify CREs active in the zebrafish adult pancreas, we performed ChIP-seq for H3K27ac[44], a key histone modification associated with active enhancers, and ATAC-seq[45], an assay that identifies regions of open chromatin (Fig. 1b). We also performed HiChIP[17] against H3K4me3[46] to detect active promoters interacting with the uncovered enhancers (Fig. 1b). We found 14753 putative active enhancers, mostly in intergenic regions (57.8%), and 23298 putative active promoters corresponding to 9848 genes (Fig. 1c; Supplementary Dataset 1a–c). To identify a subset of pancreatic enhancers with higher tissue-specificity, we compared the H3K27ac data from adult zebrafish pancreas to whole zebrafish embryos at four developmental stages, Dome, 80% epiboly, 24 h post-fertilisation (hpf) and 48 hpf[47], since these comprise differentiated and non-differentiated cells from many different tissues. We found that 7115 putative enhancers (48.2%) are active only in the differentiated adult pancreas (PsE; Fig. 1c; Supplementary Dataset 1a–c) while the remaining 7638 (51.8%) are also broadly active in developing embryos (DevE), suggesting that their activity is not restricted to the pancreas. DevE presented 4 clusters (C1-4) with different H3K27ac abundance profiles during the different developmental stages (Fig. 1d; Supplementary Fig. 2a; Supplementary Dataset 1e–l), suggesting that, apart from their activity in the adult pancreas, these enhancers might function in other cell types. C1 and C4 show similar levels of H3K27ac in all developmental stages, compatible with a putative ubiquitous enhancer activity, while C2 and C3 show different levels of H3K27ac during development, which may reflect a dynamic state of repression (C2) and activation (C3) of enhancers, or alternatively, differences in the abundance of cells where these enhancers are active during development.

### Functional similarities between human and zebrafish pancreatic enhancers.
Pancreatic enhancers are expected to activate the expression of genes in the pancreas. To test if the predicted enhancers correlate with the expression of target genes in the pancreas, we identified the nearest genes to each putative pancreatic enhancer[48,49] and observed that genes nearby PsE are enriched for exocrine pancreas expression ($p < 4.27E{-}9$; Supplementary Fig. 2b; Supplementary Dataset 2a–c), detected by in situ hybridisation[48,49]. These results contrast with the ones obtained for DevE, for which nearby genes are enriched for expression in several other tissues, including epidermis and endothelial cells (Supplementary Fig. 2; Supplementary Dataset 2d–f), suggesting a higher tissue-specificity of PsE. Additionally, the presence of endothelial expression also in genes associated to the PsE group suggests the detection of endothelial enhancers, likely derived from the vasculature present in the zebrafish adult pancreas (Supplementary Dataset 2d–f).

To improve the enhancer to gene association, we used H3K4me3 HiChIP to detect chromatin interactions between active promoters and putative enhancers in the zebrafish adult pancreas (Fig. 1b; Supplementary Dataset 3a) and used RNA-seq to evaluate transcription (Fig. 1b;[50]). We found that, compared to all genes, PsE-associated genes have a higher average expression in multiple pancreatic cell types (Fig. 2a, Supplementary

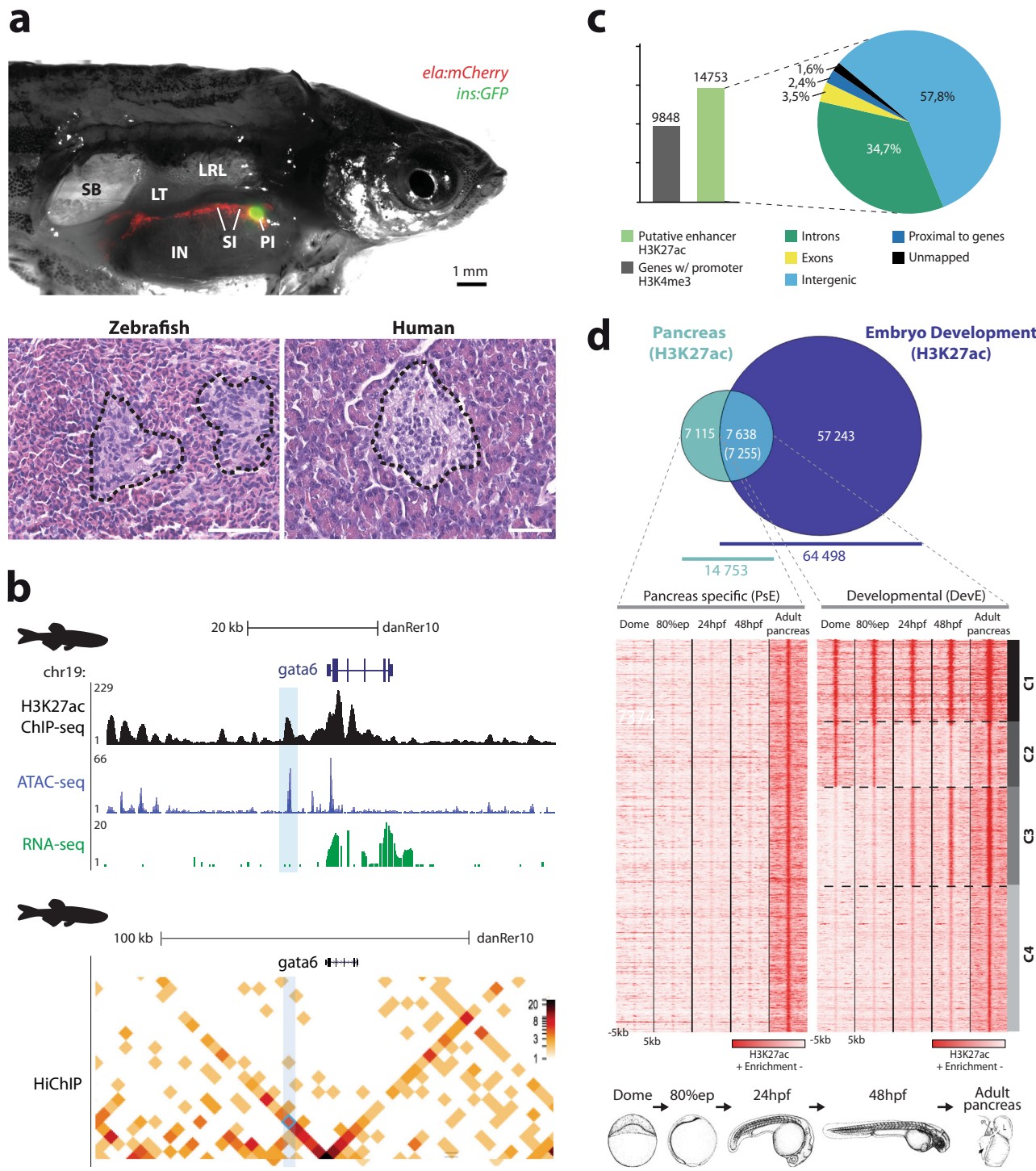

Dataset 3b). As expected, these expression results contrast with the lower average expression levels of the PsE-associated genes compared to all genes in a distantly related control tissue such as the muscle (Fig. 2a, Supplementary Dataset 3b). Similar results were obtained when analysing genes associated to the other identified clusters of pancreatic enhancers, specifically, DevE, C1-C4 and the total dataset of pancreatic enhancers altogether (PsEs+DevE; Supplementary Fig. 2c, d, Supplementary Dataset 3c–g), which had higher expression levels for at least one pancreatic adult tissue and lower expression levels in the muscle (control tissue), when compared to all transcribed genes. Next, we performed a similar analysis by calculating the ratio of the average expression level of genes associated to C1-4 and PsE putative enhancers (HC) divided by the average expression of all genes (AllG), using the previously published transcriptome of whole zebrafish embryos from 18 developmental stages[51]. We found that the genes associated to C1-4 and PsE have a HC/AllG ratio ≥ 1 (Fig. 2b; Supplementary Fig. 2e) and that the HC/AllG ratio of the DevE associated genes is higher than the one of PsE-associated genes, for most of the analysed developmental time points (Fig. 2b). These results suggest that DevE enhancers likely control gene expression during development in embryonic stages of the zebrafish. This hypothesis is further supported by the observed variation of the HC/AllG ratio during development that

**Fig. 1 The zebrafish pancreas, from histology to chromatin state. a** Comparison of the basic structure of the human and zebrafish adult pancreas. Above: Dissected adult male Tg(insulin:GFP, elastase:mCherry) zebrafish; insulin and elastase promoters drive GFP expression in beta-cells (green) and mCherry in acinar cells (red), respectively. IN, intestine; LRL, Liver right lobe; LT, left testis; PI, principal islet; SI, secondary islets; SB, swim bladder. Below: Histology of the pancreas; transverse sections with hematoxylin/eosin staining showing islets of Langerhans (black dashed lines) surrounded by exocrine tissue in zebrafish and human pancreas. Magnification: ×40 and scale bar: 1 mm. **b** Genomic landscape of *gata6* in the zebrafish adult pancreas showing the H3K27ac ChIP-seq profile (black) and ATAC-seq peaks (blue) from whole pancreas, RNA-seq from exocrine pancreas (green) and a heat map for chromatin interactions with *gata6* promoter detected by HiChIP for H3K4me3 from whole pancreas (below). A putative enhancer sequence that interacts with the *gata6* promoter is highlighted by the light blue box. **c** Bar plot (left panel) showing the number of genes with active promoters (defined by H3K4me3 signal, gray bar) and putative active enhancers in adult zebrafish pancreas (defined by H3K27ac mark, green bar), and their distribution throughout the regions of the genome (right panel). **d** Above: Venn diagram showing the overlap of putative active enhancers in adult zebrafish pancreas and stages of zebrafish embryonic development. Putative active enhancers exclusive to the adult pancreas form the pancreas-specific enhancers (PsE) group, while the shared enhancers belong to the developmental shared enhancers (DevE) group (Supplementary Dataset 1e, f). Below: Heat maps showing clusters of H3K27ac mark for PsE and DevE enhancers during embryonic development [dome, 80% epiboly (80%epi), 24 hpf, 48 hpf] and in adult pancreas. A window of 10 kb around the reference coordinates for each sequence was used and the density files were subjected to k-means clustering, obtaining four different clusters in DevE: C1, Cluster 1; C2, Cluster 2; C3, Cluster 3; and C4, Cluster 4. For **c**, **d**, source data are provided as a Source Data file.

partially reflects the variation of H3K27ac signal observed in the enhancers of the C1-4 clusters (Fig. 1d, Fig. 2b and Supplementary Fig. 2e). For instance, the C2 group that shows an increased presence of H3K27ac signal at Dome and 80% epiboly developmental time-points (Fig. 1d), also shows an increased HC/AllG ratio in the earliest developmental time points (BDO:blastula to G75: 75%epiboly; Fig. 2b and Supplementary Fig. 2e). These results suggest that C1-4 enhancers control gene expression in the adult differentiated pancreas, in addition to other cell types during development. Overall, these results increase the robustness of the pancreatic enhancers predictions, since it is possible to correlate with the transcription of the respective putative target genes.

To determine if the detected H3K27ac signal is a good predictor of active pancreatic enhancers, we performed in vivo enhancer reporter assays for 17 regions within the regulatory landscapes of known pancreatic genes. We selected sequences with detectable, but variable, H3K27ac signal overlapping with open chromatin, detected by ATAC-seq[45]. Of the 10 sequences with the highest H3K27ac values (-log10(*p*-value) from 36.5 to 92.1), 6 were validated in vivo as pancreatic enhancers (60%; Fig. 2c, d, Supplementary Fig. 3a and Supplementary Dataset 4a). Conversely, of the remaining 7 sequences with the lowest H3K27ac values (-log10(*p*-value) from 18.5 to 28.4), only 1 showed strong and reproducible evidence of pancreatic enhancer activity (14%, Supplementary Fig.3a–c and Supplementary Dataset 4a). Previous studies described similar percentages of validated enhancers from H3K27ac positive sequences[52–54]. These results validate the robustness of pancreatic enhancers prediction based on chromatin state and further suggest that the abundance of H3K27ac mark in genomic locations might improve such predictions.

We observed that out of 14753 putative zebrafish pancreatic enhancers, only 12.49% (*n* = 1842) could be directly aligned to the human genome[55] (Fig. 3a and Supplementary Dataset 3i–l). A similar proportion was found in the group of developmental enhancers (11.36%; 7326 out of 64,498; Fig. 3a). Using the corresponding human sequences from the pancreas and developmental enhancers groups, we found that they share similar PhastCons conservation scores (Fig. 3b; Supplementary Fig. 3d and Supplementary Dataset 3m–p). Next, we wanted to determine if the zebrafish putative pancreatic enhancers that align to the human genome also overlap with H3K27ac signal from human pancreas. Only a minority of interspecies aligned sequences shared H3K27ac signal (total pancreas data set: 227 out of 1842; PsE: 115 out of 1052; DevE: 112 out of 790). The human sequences, that shared H3K27ac signal with zebrafish, did not show a higher average

conservation score than the aligned sequences that showed H3K27ac signal in zebrafish alone (Fig. 3b and Supplementary Fig. 3e; Average sequence conservation score for H3K27ac non-shared vs shared signal, Pancreas: 0.40vs0.36, PsE:0.42vs0.41, DevE:0.36vs0.34). Notwithstanding the low absolute numbers of aligned sequences that share H3K27ac signal in human and zebrafish pancreas, these sequences represent a clear enrichment compared to the overlap obtained by randomized set of sequences in the human genome (3.21 times higher for pancreas, 2.79 times higher for PsE, 3.76 times higher for DevE and 1.76 times higher for embryo, Fig. 3c; Supplementary Dataset 3q). Overall, these results suggest that pancreatic enhancer function is not a strong condition to impose sequence conservation.

Following these data, we assessed whether functionally equivalent pancreatic CREs exist between human and zebrafish, despite an overall lack of sequence conservation. To explore this possibility, we investigated if the genes interacting with each cluster of zebrafish enhancers were enriched for homologs of human genes associated with pancreatic diseases, which would suggest the existence of functionally equivalent pancreatic CREs with potential biomedical relevance. Such enrichment was observed for the clusters of late development and adult pancreas (PsE, C3 and C4; Fig. 3d; Supplementary Dataset 3r, s). Human gene-disease associations were retrieved from DisGeNET[56] and we observed that 306 out of 836 zebrafish genes (36.6%) homologous to human pancreas disease-associated genes also interact with zebrafish pancreatic enhancers.

Enhancers can exist in their typical form, as short and restricted regions of DNA, or they can be present as large regions of hyperactive chromatin referred to as super enhancers[13,57,58]. Several computational approaches have been applied to identify super enhancers in vertebrate genomes, including in human and zebrafish[59]. We searched for super enhancers active in the pancreas of human and zebrafish (Supplementary Dataset 1m, n; 275 in zebrafish and 875 in human), to understand if pancreatic super enhancers control the same genes in both species, further suggesting an equivalency in function. Gene ontology for putative target genes showed a similar enrichment for transcriptional regulation in both species and several of these genes corresponded to the same orthologues (32 out of the 271 zebrafish genes; Supplementary Fig. 3f–g), some with important pancreatic functions, such as *INSR*, a critical regulator of glucose homoeostasis[60] and *GATA6*, which plays a crucial role in pancreas development and β-cell function[61] (Supplementary Fig. 3h). We further inquired if human and zebrafish enhancers might operate similarly, using equivalent TFs. To test this, we performed a motif enrichment search for TF binding sites (TFBS)

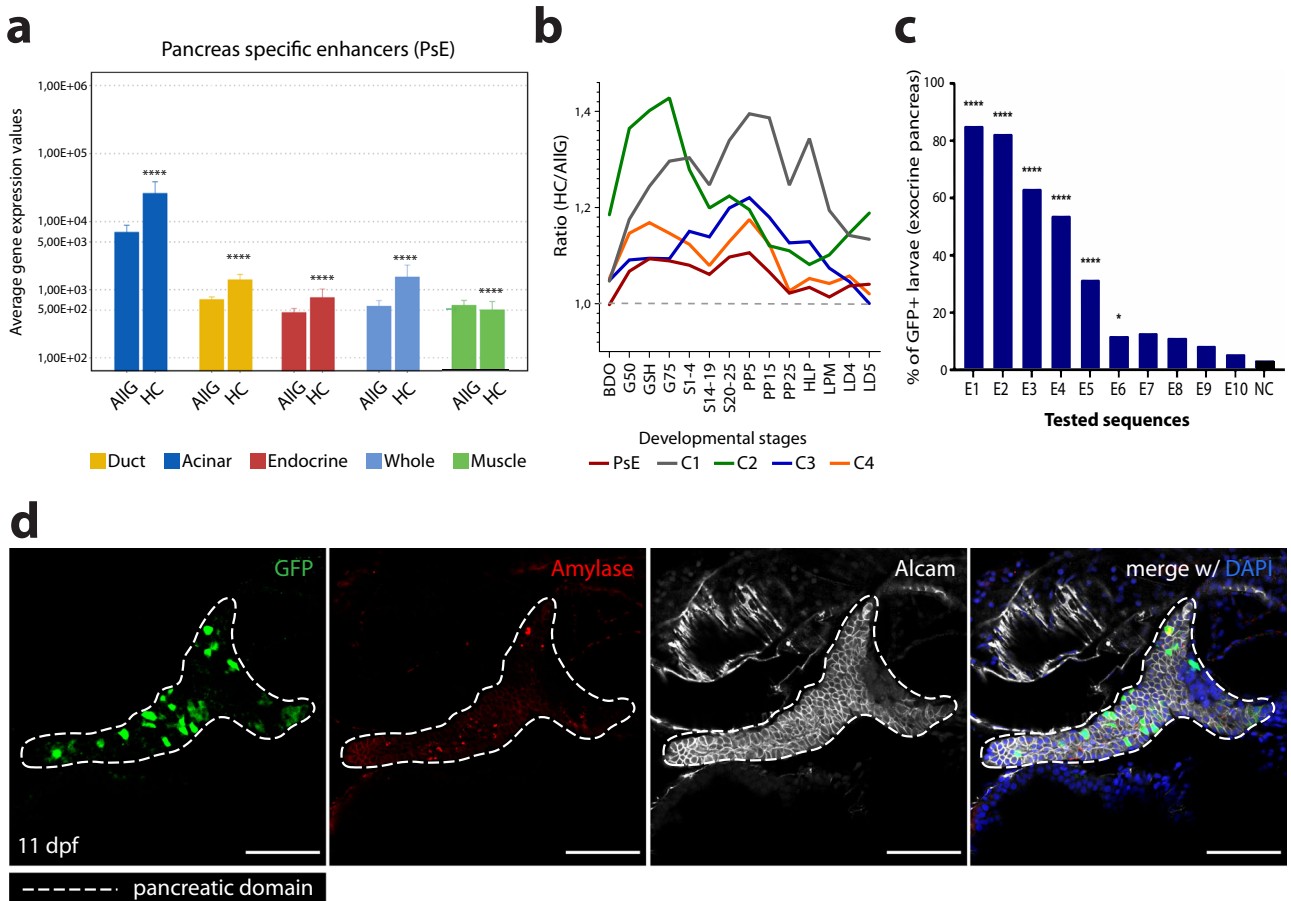

**Fig. 2 ChIP-seq and ATAC-seq data accurately predict functional pancreatic enhancers. a** Average expression of genes interacting with putative pancreas-specific enhancer sequences (PsE), detected by HiChIP for H3K4me3 (HC, $n = 6174$ genes), compared to the average expression of all genes (AllG, $n = 33737$ genes). Gene expression was determined from RNA-seq data from different pancreatic cells (acinar $n = 4$, duct $n = 3$, endocrine pancreas $n = 4$), whole pancreas ($n = 2$), and muscle (control; $n = 2$). One-sided Wilcoxon test (≥), $p$-values < 0.05 were considered statistically significant (****$p < 2E{-}16$). Error bars represent the 95% confidence interval. **b** Ratio between average expression of genes interacting with putative pancreatic enhancers (PsE, C1, C2, C3 and C4 clusters) and the average expression of all genes throughout zebrafish development. C1, C2, C3 and C4 are different clusters that compose the DevE category. BDO: blastula, dome; G50: gastrula, 50% epiboly; GSH: gastrula, shield; G75: gastrula, 75% epiboly, S1–4: segmentation, 1–4 somites; S14–19: segmentation, 14–19 somites; S20–25: segmentation, 20–25 somites; PP5: pharyngula, Prim-5; PP15: pharyngula, Prim-15; PP25: pharyngula, Prim-25; HLP: hatching, long-pec; LPM: larval, protruding-mouth; LD4: larval, day 4; LD5: larval, day 5. **c** Percentage of F0 zebrafish larvae with GFP expression in the exocrine pancreas following in vivo transient transgenesis reporter assays. The empty enhancer reporter vector was used as the negative control (NC). The depicted sequences (E1 to 10) represent the top 10 putative enhancer sequences with higher H3K27ac signal ("high H3K27ac" group). Values are represented as percentages and compared by two-sided Chi-square with Yates' correction test. $p$-values<0.05 were considered significant (****$p < 0.0001$, *$p < 0.05$). The exact $p$-value and $n$ are discriminated in Supplementary Dataset 4. **d** Representative confocal image of the in vivo transient transgenesis reporter assays for the E3 sequence ($n = 30$). depicted in **c**) showing expression of GFP (green) in 11 dpf zebrafish pancreas (white dashed line), labelled by anti-Alcam staining (white) and anti-Amylase (red) antibodies ($n = 30$, from 2 independent injections, with 63.33% of larvae showing GFP expression in the exocrine pancreas). Nuclei were stained with DAPI (blue). Images were captured with a Leica SP5II confocal microscope. Scale bar: 60 μm. For **a–c**, source data are provided as a Source Data file.

in regions of open chromatin identified by ATAC-seq[45], within the 14753 pancreatic enhancers, and found several TFBS for known pancreatic TFs (ZP; Fig. 3f, Supplementary Fig. 4a, and Supplementary Dataset 3t, u). We also performed a similar analysis using available human whole pancreas datasets (HP[62]; Datasets summarised in Supplementary Dataset 4g). To compare the extent of overlap of enriched motifs in human and zebrafish pancreatic enhancers with motifs enriched in other pancreas unrelated enhancers, we have performed a similar motif enrichment search for datasets of zebrafish embryos (D80, dome and 80%epiboly; 24 HPF, 24 hpf) and human heart ventricle (V[62]; Datasets summarised in Supplementary Dataset 4g). We selected the top 140 enriched motifs from each dataset and observed that the majority of the common motifs were found in zebrafish (ZP)

and human (HP) pancreas datasets (ZP,HP:98; ZP,D80:63; HP,D80:61) (Fig. 3g, Supplementary Fig. 4b), while comparisons with the human ventricle (V) showed that ZP,HP was the second largest group following HP, V (Supplementary Fig. 4c).

Several TFs, such as Ptf1a, Pdx1, Pax6 and Sox9, are known to be important for pancreas function or development in several vertebrate species, including human and zebrafish[2,63–65]. As shown above, human and zebrafish pancreatic enhancers are enriched for many shared TFBS, therefore it is reasonable to expect that many of these TFBS are from TFs known to have an important pancreatic function. To test this hypothesis, we have selected 25 TFs known to be required for pancreas function and development and calculated the distribution of the respective TFBS motifs within the previously identified enriched motifs

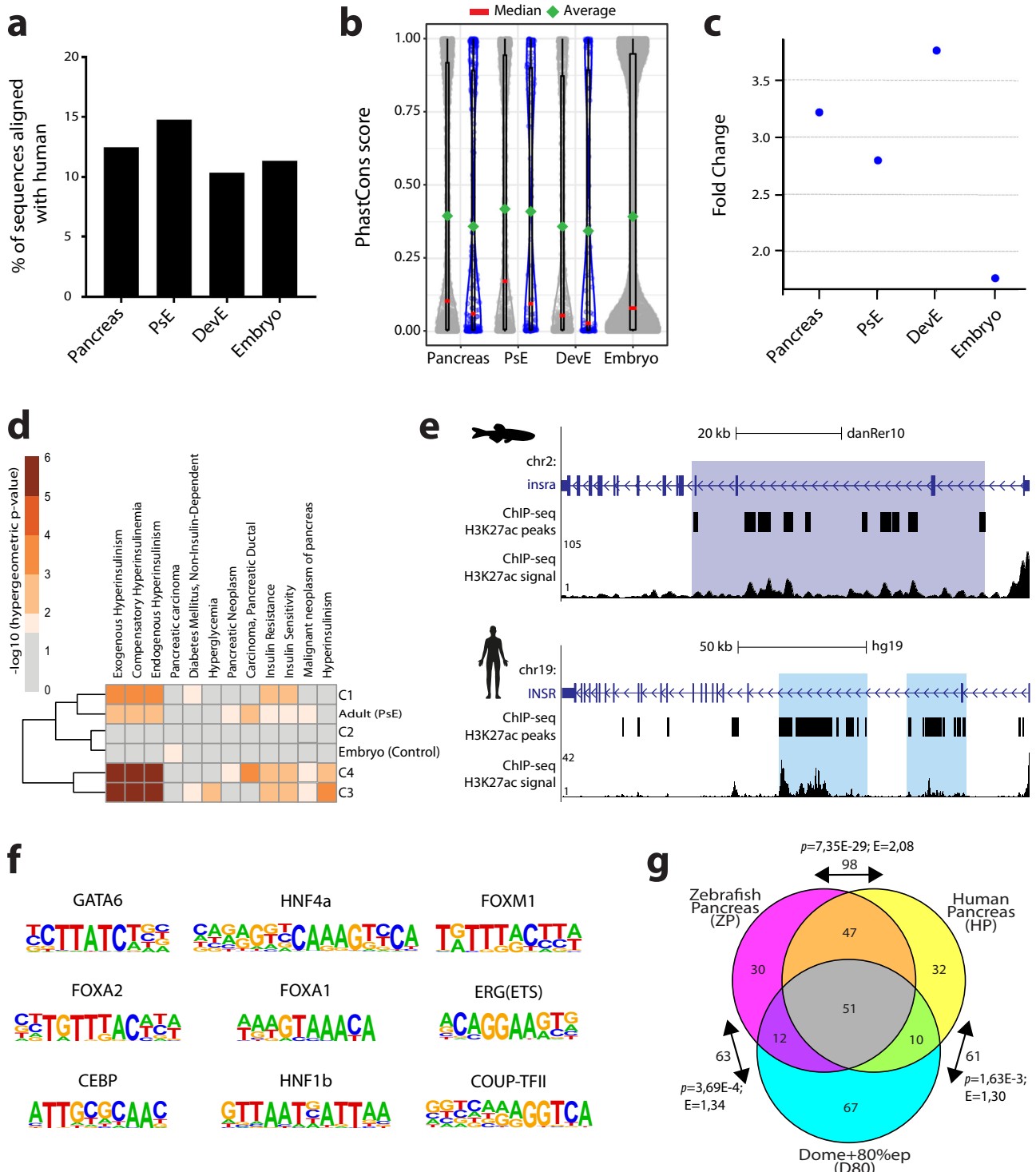

described in Supplementary Dataset 3t. We found that the majority of the TFBS motifs from the pancreatic TFs were within the ZP,HP overlapping datasets, regardless of the compared groups (Supplementary Fig. 4d–f). These results suggest that the same set of TFs operates in zebrafish and human pancreatic enhancers. Overall, these results argue in favour of interspecies functional equivalency of enhancers.

**Landscape of *arid1a* reveals potential pancreatic cancer associated enhancer.** To better address the hypothesis of interspecies

functional equivalency of enhancers, we focused on the regulatory landscape of a gene that is potentially linked to human pancreatic diseases. We selected *arid1ab*, the orthologue of human *ARID1A*, a tumour-suppressor gene associated with cancer in several different cell types[33,34], including pancreatic ductal adenocarcinoma[66]. ARID1A plays a key role in the regulation of DNA damage repair, by promoting an efficient processing of double-strand breaks into single-strand ends, being required to sustain DNA damage signalling and repair, hence suppressing tumorigenesis[67].

We identified several putative enhancers (zA.E1-4, Fig. 4a), that we tested in vivo using enhancer reporter assays (Supplementary

**Fig. 3 The zebrafish and human pancreas share cis-regulatory similarities. a** Percentage of predicted zebrafish pancreatic enhancer sequences aligned to the human genome. Sequences are grouped in different clusters: "Pancreas" that includes PsE and DevE; "PsE"; "DevE"; "Embryo" that include putative enhancers active only during embryonic development. **b** PhastCons scores (99 vertebrate genomes against hg38) for human sequences converted from zebrafish putative enhancers. Grey dots label conserved sequences that do not overlap with H3K27ac mark in human pancreas (Pancreas-1801, PsE-1017, DevE-784 and Embryo-6792). Blue dots label conserved sequences that also show H3K27ac signal in human pancreas (ENCODE data; Pancreas-227, PsE-112, DevE-115). Green diamonds: average (grey dots: 0.40, 0.42, 0.36, 0.39; blue dots: 0.36, 0.41, 0.34, respectively for Pancreas, PsE, DevE and Embryo). Red line: median (grey dots: 0.10, 0.17, 0.05, 0.08; blue dots: 0.06, 0.09, and 0.03, respectively for pancreas, PsE, DevE and Embryo). The Embryo dataset is composed by different developmental stages (Dome, 80% Epiboly, 24 hpf and 48 hpf). **c** Ratio between the number of human sequences conserved with the zebrafish putative active enhancers (Pancreas-3.21, PsE-2.79, DevE-3.76 or Embryo-1.76) overlapping H3K27ac signal in human pancreas (ENCODE data) over the average of a $10^5$ random shuffling of human sequences overlapping with H3K27ac signal in human pancreas (Supplementary Dataset 3q; empirical $p$-value < 1E−5). **d** Heatmap showing -$\log_{10}$($p$-values) from hypergeometric enrichment test for pancreatic disease association on the genes linked by HiChIP to each enhancer cluster. Represented values meet the criteria: $q$-value ≤ 0.05 and fold enrichment ≥ 1.5. **e** Genomic landscape of the human *INSR* gene (top) and zebrafish *arid1ab* ortholog (bottom), showing H3K27ac signal and predicted super-enhancers (blue). **f** Relevant pancreas transcription factors whose binding motifs are enriched in zebrafish pancreas H3K27ac ChIP-seq data. **g** Venn diagram of the top 140 enriched TFBS motifs in H3K27ac positive sequences in three different datasets: zebrafish pancreas (ZP), human pancreas (HP) and dome+80%epiboly embryos (D80). Number of motifs shared between pairs of groups (arrows). $p$-values are described ($p$; hypergeometric enrichment test). The enrichment of the observed *vs* expected is represented (E). $p$-values ≤ 0.05 were considered significant. For **a**–**d**, **g**, source data provided in Source Data file.

Dataset 4a). Of these, zA.E2 and zA.E4 were validated as pancreatic enhancers. zA.E4 was the most robust pancreatic enhancer of this set (Fig. 4a and Supplementary Dataset 4a), driving expression in endocrine, acinar and duct cells of the zebrafish pancreas (Fig. 4b and Supplementary Fig. 5a) and interacting with the promoter of *arid1ab* (Fig. 4a and Supplementary Fig. 5b). Additionally, we detected a human/zebrafish syntenic block containing the zebrafish zA.E4 enhancer and a human pancreatic CRE (hA.E4) (Fig. 4a). In vivo enhancer assays for hA.E4 demonstrated its ability to drive expression in endocrine cells of the zebrafish pancreas, and in vitro in a human pancreatic duct cell line (hTERT-HPNE), suggesting a functional equivalency to the zebrafish zA.E4 enhancer (Fig. 4b, c and Supplementary Fig. 5a). To study the influence of this human enhancer on *ARID1A* expression, we deleted the hA.E4 enhancer in the hTERT-HPNE cell line, relevant for the pancreatic tumour suppressor role of *ARID1A*, through CRISPR-Cas9 system (Fig. 4d and Supplementary Fig. 5c–e), using a deletion in an unrelated genomic region[16] as a control. We observed lower levels of ARID1A upon deletion of hA.E4 compared to the control (Fig. 4e, f and Supplementary Fig. 5e), suggesting that the loss of this enhancer may interfere with the DNA-damage response, with possible implications in the increased risk for pancreatic cancer[68,69].

**A *ptf1a* enhancer explains pancreatic agenesis causal variant in vivo**. To further evaluate the interspecies functional equivalency of enhancers and their role in human pancreatic diseases, we focused on the human *PTF1A* locus, known to be controlled by a distal downstream enhancer whose deletion causes pancreatic agenesis[35] (Fig. 5a; hP.E3). Concomitantly, we detected a zebrafish distal *ptf1a* enhancer, downstream of *ptf1a* (zP.E3), as well as two previously identified proximal enhancers (zP.E1 and zP.E2;[70]). zP.E3 interacts with the promoter of *ptf1a*, observed by Hi-ChIP and 4C-seq (Fig. 5a and Supplementary Fig. 5b), and could correspond to the functional equivalent enhancer whose deletion causes pancreatic agenesis in humans (hP.E3), although its sequence partially aligns with a more distal human sequence likely inactive in human pancreatic cells (Supplementary Fig. 6). In vivo enhancer assays for zP.E3 and hP.E3 showed strong and robust expression in progenitor cells (Fig. 5b), a result that is in agreement with the described activity of hP.E3 in vitro as a human developmental enhancer[35]. These results suggest that the human and zebrafish enhancers share some regulatory information. This is further supported by binding sites for FOXA2 and PDX1 in the human hP.E3, also predicted to bind to the zebrafish zP.E3 (Supplementary Fig. 7a, b;[71]). To further evaluate the role

of zP.E3, we generated genomic deletions in the zP.E3 sequence (Fig. 5c–g, Supplementary Fig. 8 and Fig. 9). Deletion1, a 632 bp deletion that includes the predicted Foxa2 and Pdx1 binding sites and the majority of transposase-accessible chromatin within zP.E3 (Supplementary Fig. 9a), results in a decrease of the pancreatic progenitor domain area in homozygous mutants (Fig. 5 c, d, f), as well as a reduction in the expression levels of *ptf1a* (Supplementary Fig. 9b). Furthermore, after pancreatic differentiation, the Deletion1 mutants displayed pancreatic hypoplasia (Fig. 5e, g; Supplementary Fig. 9c–e), and we observed the same phenotype for multiple independent deletions of zP.E3 generated in somatic cells (Supplementary Fig. 8). In contrast, no phenotypes were observed for a 517 bp deletion within the zP.E3 enhancer, adjacent to Deletion1, which excludes the majority of accessible chromatin and predicted TF binding sites (Deletion2; Supplementary Fig. 9a, d, e), suggesting that the functional core of zP.E3 coincides with the regions of available chromatin that overlap with the predicted binding of Foxa2 and Pdx1. In agreement with the observed phenotypes, pancreatic hypoplasia is compatible with the described loss-of-function of *ptf1a* in zebrafish[70] and the loss of hP.E3 function in humans[35]. In light of these results, we suggest that pancreatic hypoplasia is the consequence of the reduction in the pancreatic progenitor domain caused by decreased levels of *ptf1a* due to the loss of an important pancreatic progenitor enhancer.

Later on, after pancreatic differentiation, zP.E3 and hP.E3 enhancers acquire distinct activity patterns. The zebrafish zP.E3 enhancer is able to drive a consistent expression in differentiated pancreatic cells from late embryos up to adults (Supplementary Fig. 10), including acinar and duct cells, while the human hP.E3 enhancer shows almost a total lack of activity in differentiated acinar and duct cells, as previously observed in vitro[35] driving expression only in very few cells (Supplementary Fig. 10). Overall, these results suggest that zebrafish and humans share a functionally equivalent distal enhancer of *PTF1A* during development, whose loss-of-function results in a reduction of the pancreatic progenitor domain, elucidating, in vivo, the causal link between the disruption of this enhancer in humans and pancreatic agenesis.

## Discussion
Cis-regulatory mutations and sequence variations are associated with pancreatic cancer and diabetes[2–6]. However, the in vivo implications of these genetic changes are still unknown. Here, we explore the chromatin state of the zebrafish pancreas to uncover pancreatic enhancers and establish comparisons with humans,

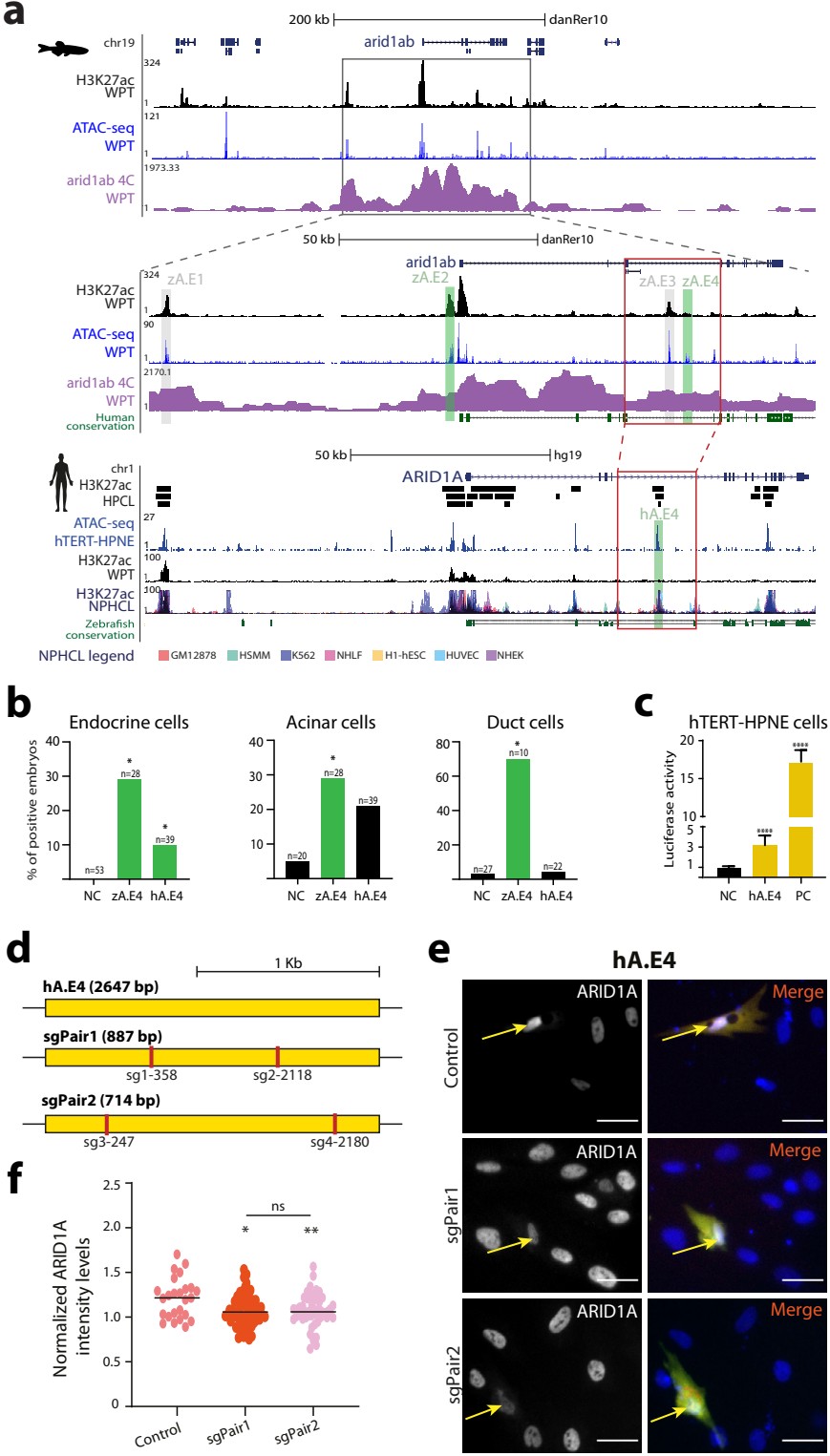

so that we can predict and model human pancreas disease-associated enhancers. We found that, although most of the zebrafish pancreatic enhancers do not share significant sequence identity with human pancreatic enhancers, they share many TFBS and their target genes are enriched for human pancreas diseases. These results suggest the existence of functionally equivalent enhancers in zebrafish and humans, as proposed for other tissues and species[72,73]. Indeed, recent studies looking into highly divergent species as human and sponges have located similarly

functional enhancers within microsyntenic regions that, although do not share significant sequence identity, clearly recapitulate similar expression patterns in enhancer reporter assays, arguing in favour of functional equivalency[74]. This is likely the consequence of enhancers being fast-evolving sequences operating with a high degree of sequence flexibility[75]. Several mechanisms that may operate together during evolution can illustrate the potential for sequence flexibility of enhancers while retaining a consistent TFBS code. Among them, nucleotide alterations within

**Fig. 4 The zebrafish and human *arid1ab/ARID1A* regulatory landscapes contain an equivalent pancreatic enhancer. a** Genomic landscape of the zebrafish *arid1ab* gene, showing profiles for H3K27ac ChIP-seq (black), ATAC-seq (blue) and 4 C with viewpoint in the *arid1ab* promoter (magenta) in adult zebrafish pancreas (top); zoom-in in *arid1ab* regulatory landscape (middle). Human *ARID1A* genomic landscape (bottom) with H3K27ac enriched intervals from human pancreatic cell lines (HPCL, black bars, top-to-bottom: PT-45-P1, CFPAC-1 and HPAF-II), H3K27ac profile from human pancreas (WPT, black) and from non-pancreatic human cell lines (NPHCL; GM12878, H1-hESC, HSMM, HUVEC, K562, NHEK and NHLF; Data from ENCODE). Human/zebrafish sequence conservation (dark green). Tested putative enhancers are highlighted in grey (zA.E1 and zA.E3; no enhancer activity) and green (zA.E2, zA.E4 and hA.E4; enhancer activity). Zebrafish/human syntenic box (red box). **b** Transient in vivo enhancer reporter assays of zA.E4 and hA.E4 showing the percentage of zebrafish embryos with GFP expression in endocrine, acinar and duct cells (two-sided chi-square test with Yates correction; *$p < 0.05$; Endocrine cells: zA.E4, $p = 0.0001$; hA.E4, $p = 0.0294$; Acinar cells: zA.E4, $p = 0.0391$; hA.E4, $p = 0.1167$; Duct cell: zA.E4, $p = 0.00001$; hA.E4, $p = 0.9731$). Number of analysed embryos (*n*). Negative control (NC). **c** Luciferase enhancer reporter assays performed in human hTERT-HPNE cells for hA.E4, showing luc2/Nluc ratios, relative to the negative control (two-sided *t*-test; ****$p < 0.0001$; hA.E4 *p*-value = 0.0001; PC *p*-value < 0.0001). Data from three biological replicates (grey dots, $n = 3$) and Mean±SD (error bar). Negative control (NC). Positive control (PC). **d** Strategy for CRISPR-Cas9 deletions in the hA.E4 locus, indicating sgRNA target sites. **e** Representative images of transfected hTERT-HPNE human cells expressing pairs of sgRNAs and Cas9 (arrows). In control, sgRNAs target a H3K27ac depleted region, while sgRNAs in sgPair1 and sgPair2 target the hA.E4 locus. Left column show anti-ARID1A (grey) and right column GFP (green), mCherry (red) and DAPI (blue; nuclei). Representative images from three biological replicates. Scale bar: 40 μm. **f** Normalized ARID1A levels from immunocytochemistry images. Two-sided *t*-test depicted for $p \leq 0.05(*)$, $p \leq 0.01(**)$ and not significant (ns; *p*-values of: Control vs sgPair1 = 0.0208, Control vs sgPair2 = 0.0044, sgPair1 vs sgPair2 = 0.6227). A black line represents the mean of values. Data from three biological replicates. Data included in Source Data file for **b**, **c**, **f**.

similar TFBS[76], reshuffle of TFBSs within enhancers, compatible with a billboard model[77,78], and substitution of enhancer's sequence by acquisition of redundant enhancers within the same regulatory landscape[79]. In the current work we show several examples compatible with the potential for enhancers' sequence flexibility. Focusing on the regulatory landscape of *Arid1a*, a tumour-suppressor gene active in the pancreas[66,68] and other tissues[33], we show that within a microsyntenic region within the *arid1a* locus in humans and zebrafish, there are pancreatic enhancers that share regulatory information, although not sharing significant sequence identity. We further show that the deletion of the human *ARID1A* pancreatic enhancer impairs ARID1A expression, defining a locus for non-coding mutations that may increase the risk for pancreatic cancer. We further explored the potential of functional equivalency for an enhancer of *ptf1a*[80], in which both zebrafish and human enhancers share regulatory information and biological requirements during pancreas development. The loss-of-function of the zebrafish enhancer results in a decrease of the pancreatic progenitor domain and ultimately in pancreatic hypoplasia, a phenotype consistent with the impact of mutations described in the human regulatory landscape, which are associated with pancreatic agenesis[35]. The reduction of the pancreatic progenitor domain in zebrafish may explain the phenotype observed in humans, contributing to the clarification of its molecular and cellular origin. Interestingly, the deletion of the zebrafish *ptf1a* enhancer does not show a complete phenotypic penetrance, with ~25% of the embryos having a pancreas morphologically similar to the controls, suggesting that other redundant enhancers may exist in the zebrafish regulatory landscape of *ptf1a*, compatible with a shadow enhancer identity[81]. Additionally, human and zebrafish *ptf1a* enhancers show divergent functions after differentiation. While the human enhancer shows very little activity in differentiated pancreatic cells, the zebrafish enhancer drives persistent reporter expression, suggesting that the phenotype in zebrafish after pancreatic differentiation could have the extra contribution of this late zebrafish specific function of the *ptf1a* enhancer.

Sequence conservation of CREs can be a good predictor of sequence functionality, however it holds important limitations in the prediction of equivalent functions. This is observed in the current work, where the vast majority of the zebrafish pancreatic enhancers that could be aligned to the human genome did not share marks of enhancer activity in pancreatic cells. This is further illustrated by zP.E3, which shows some partial alignment with a human sequence that has no active marks of enhancer in

pancreatic cells. Many examples have been described showing how conserved sequences among divergent species might harbour divergent functions. These include differences in conserved enhancer sequences resulting in functional divergence[82,83], to more striking examples of coding exons sequences repurposed to cis-regulatory functions[79]. Additionally, recent studies have shown that the ultra-conservation at sequence level observed in some enhancers is not necessary for the maintenance of tissue specific regulatory functions, suggesting that sequence constraint may partially result from other regulatory or unknown functions[75].

The use of animal models to understand the role of CREs in the development of human diseases requires the identification of functionally equivalent sequences. As discussed above, sequence conservation is not a reliable predictor of functional conservation[84] and functional equivalent sequences might not present high sequence conservation[85]. This problem can be partially bypassed by combining the use of biochemical marks associated to CREs activity with enhancer reporter assays to identify similar regulatory information harboured by such sequences. In the current work we used this strategy, allowing us to identify and test in vivo enhancers that, when altered, can affect the expression of disease-associated genes. This strategy can help to identify where in the genome disease-causing non-coding mutations may occur by predicting disease-relevant CREs based on phenotypic description of CRE's loss-of-function. Furthermore, in the near future this strategy may be further improved by computational methods as well as the detection of TFBS in both species. These improvements could help to establish a correspondence of enhancers' identity genome wide.

The pancreas is a complex structure composed by multiple cell types. In this work we assessed the chromatin state of the whole pancreas of adult zebrafish in order to identify pancreatic CREs and their target genes. By associating CREs to the expression of target genes, we have shown that our dataset includes exocrine and endocrine CREs. This broad pancreatic enhancer map is very advantageous since it allows us to approach different biological and biomedical questions related with different pancreatic cell types. The pancreas also contains other cell types that are heavily intertwined, as is the case of endothelial cells. Indeed, several of our observations indicate the presence of endothelial enhancers in the described CREs datasets, namely the enrichment of endothelial expressing genes located nearby DevEs (Supplementary Dataset 2d–f) and the extended overlap of common motifs between pancreatic enhancers and heart ventricle enhancers.

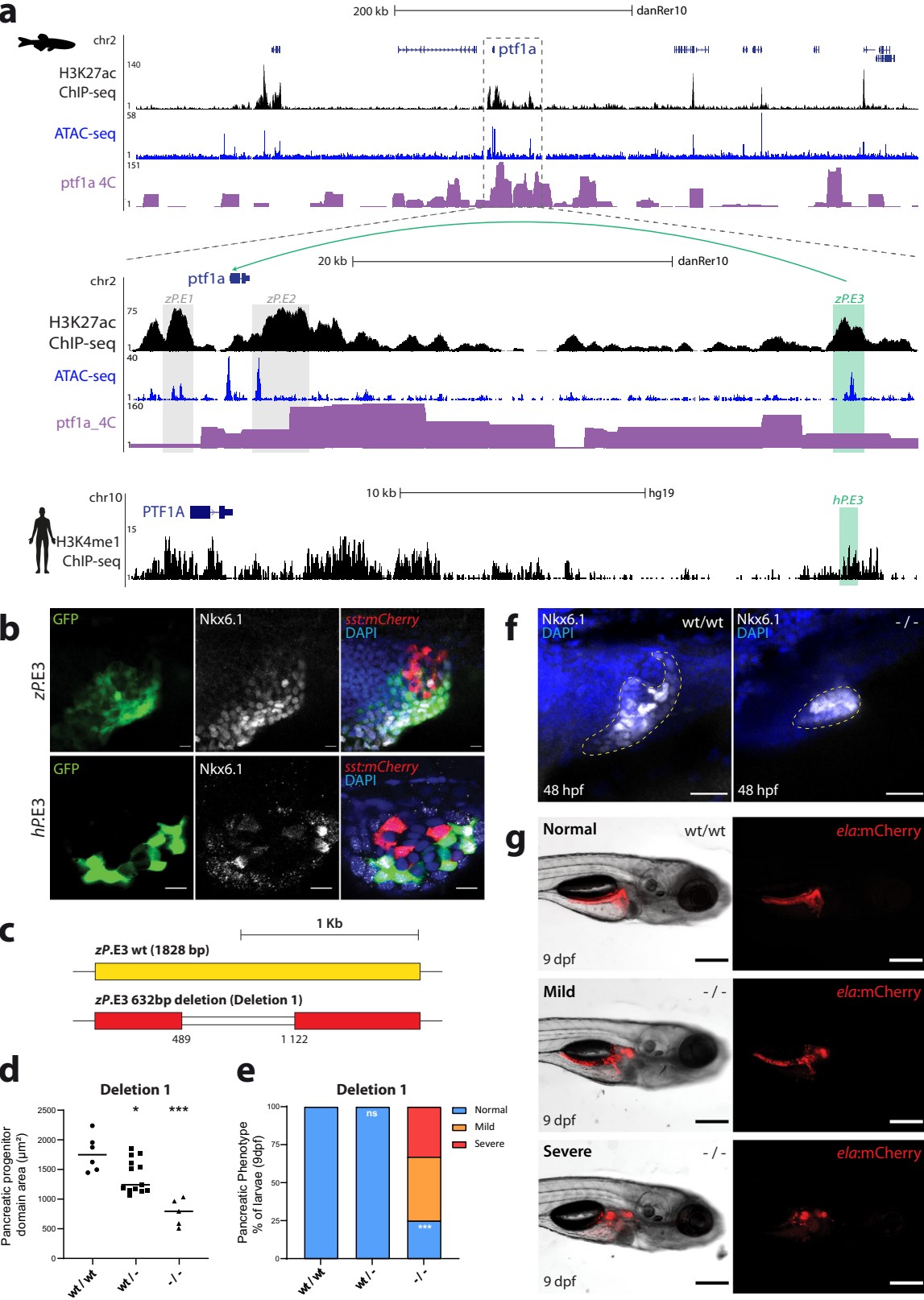

Enhancers can be highly tissue specific, while others can be active in multiple tissues, as observed by the identification of PsE and DevE. The former showed H3K27ac profiles more restricted to the zebrafish adult pancreas, while the latter had broad profiles throughout development, suggesting their activity to be present in multiple tissues. The zP.E3 enhancer is not detected in the embryonic H3K27ac

dataset, likely because its activity is highly restricted to pancreatic progenitor cells during development, resulting in its inclusion in the PsE group. A detailed analysis of the activity of this enhancer, from the larval stage to adulthood, shows it to be almost exclusively active in exocrine pancreatic cells (Supplementary Fig. 10e), illustrating the expected tissue specificity of PsE enhancers.

**Fig. 5 The zebrafish and human *ptf1a/PTF1A* regulatory landscapes contain a functional equivalent enhancer essential for pancreas development.** **a** UCSC Genome Browser view of the zebrafish *ptf1a* and human *PTF1A* genomic landscapes showing H3K27ac ChIP-seq (black), ATAC-seq (blue) and *ptf1a* 4 C interactions (purple) from whole zebrafish pancreas samples (upper panel), with a zoom-in (middle panel), and H3K4me1 ChIP-seq data[2] (black) from human embryonic pancreatic progenitors (lower panel). Grey boxes highlight two previously validated zebrafish enhancers, zP.E1 and zP.E2 in the vicinity of the *ptf1a* gene. Green boxes highlight a distal enhancer in zebrafish, zP.E3, and the location of its putative human functional ortholog hP.E3. **b** Confocal images of zebrafish reporter stable transgenic lines Tg(zP.E3:GFP) (n = 10) and Tg(hP.E3:GFP) (n = 3), showing co-localization of GFP expression (green) with Nkx6.1 (white), a marker of pancreatic progenitors, at 48 hpf. Delta-cells of the endocrine pancreas express mCherry (red) and nuclei are labelled with DAPI (blue). Scale bar: 25 μm. **c** Schematic depiction of the CRISPR-Cas9 mediated 632 bp deletion (Deletion 1) of the zP.E3 enhancer. **d** Pancreatic progenitor domain area, defined by Nkx6.1 (white), of homozygous (−/−; n = 5), heterozygous (wt/−; n = 13) and wild type (wt/wt; n = 6) embryos for Deletion1 of zP.E3, at 48 hpf. Unpaired student's t-test (two-tailed), p-values < 0.05 were considered significant (*p = 0.017, ***p = 0.0002). **e** Percentage of larvae (−/−, n = 12; wt/−, n = 14 and wt/wt, n = 12) with different pancreatic phenotypic defects (normal, mild and severe) at 9 dpf. Fisher's exact test (two-sided), p-values < 0.05 were considered significant (***p = 0.0003). **f** Representative confocal images (maximum intensity projections) of the pancreatic progenitor domain (yellow dashed line) of zP.E3wt/wt (n = 6) and zP.E3−/− sibling embryos (n = 5) at 48 hpf. Nuclei are stained with DAPI. Scale bar: 25 μm. **g** Epifluorescence live images of representative phenotypes quantified in **e**). Scale bar: 250 μm. ela elastase, sst somatostatin. For **d**, **e**, source data are provided as a Source Data file.

In this work, we identified pancreatic CREs in zebrafish, a model organism that is amenable to genetic manipulation and phenotyping. By establishing a correlation between human and zebrafish pancreatic CREs, functional testing of CREs can be performed in vivo, helping to clarify the role of CREs in pancreatic function and disease. In summary, the combination of techniques used in this work, allowed the identification of human cis-regulatory elements involved in disease. We show that transcriptional cis-regulation of the human and zebrafish adult pancreas have a high degree of similarity, allowing the functional exploration of cis-regulatory sequences in zebrafish, with the potential of translation to human pancreatic diseases.

## Methods

### Experimental procedures

*Zebrafish stocks, husbandry, breeding and embryo rearing.* Adult zebrafish AB/TU WT strains where obtained from the Gomez-Skarmeta's laboratory in Seville (CABD). WT, transgenic and mutant lines were maintained at 26–28 °C under a 10 h dark/14 h light cycle in a recirculating housing system according to standard protocols[86]. Embryos were grown at 28 °C in E3 medium [5 mM NaCl (#S/3161/60, Fisher Chemical), 0.17 mM KCl (#2676.298, VWR), 0.33 mM CaCl₂•2H₂O (#C3881, Sigma-Aldrich), 0.33 mM MgSO₄•7H₂O (#63140, Sigma-Aldrich) and 0.01% methylene blue (#66120, Sigma-Aldrich), pH 7.2] or E3 supplemented with 0.01% PTU (1-phenyl-2-thiourea, #P7629, Sigma-Aldrich)[87]. For the in vivo enhancer assays, embryos were anesthetized by adding tricaine (MS222; ethyl-3-aminobenzoate methanesulfonate, #E10521-10G, Sigma-Aldrich) to the medium and selected by the internal positive control of transgenesis. For the establishment of transgenic and mutant zebrafish lines, embryos were microinjected, selected, bleached and grown until adulthood. Adult F0s were outcrossed with WT adults and the offspring screened for the internal control of transgenesis and the pattern of expression of the regulatory element, or for the respective mutations, by genotyping. In vivo reporter lines, Tg(ela:mCherry) and Tg(sst:mCherry), were used to label the exocrine and endocrine domain, respectively. The i3S animal facility and this project were licensed by *Direcção Geral de Alimentação e Veterinária (DGAV)* and all the protocols used for the experiments were approved by the i3S Animal Welfare and Ethics Review Body.

*Cell culture.* hTERT-HPNE (ATCC CRL-4023) cells were cultured in a 5% CO₂-humidified chamber at 37 °C in DMEM (1×, 4.5 g/L D-glucose with pyruvate; #D6429, Gibco, ThermoFisher Scientific), supplemented with 10% fetal bovine serum (#BCS0615, biotecnomica), 10 ng/mL human recombinant EGF (#11343406, Immunotools) and 750 ng/mL puromycin (#P8833-25MG, Sigma-Aldrich) in TC Dish 100 (SARSTEDT). When cells reached 90% of confluence, they were split using TrypLE Express (#12604-021, Gibco, ThermoFisher Scientific; ~0.5 mL per 10 cm²).

*ChIP-seq.* Whole pancreas was dissected from 25 adult zebrafish (~50 × 10⁶ cells; both genders and with 12–24 months), kept on ice in PBS [137 mM NaCl (#S/3161/60, Fisher Chemical), 2.7 mM KCl (#2676.298, VWR), 10 mM NaHPO4 (#1.06342.0250, Merk), and 1.8 mM KH2PO4 (#1.06585.1000, Merk)] with 1x Complete Proteinase Inhibitor (#11697498001, Roche), fixed in 2% formaldehyde (#F1635-500ML, Sigma-Aldrich) for 10 min, and stored at −80 °C. ChIP was performed as previously described for zebrafish embryos[31] with minor alterations. Cell lysis was performed on ice, using a 15 mL Tenbroeck Homogenizer, in cell lysis buffer [10 mM Tris-HCl pH7.5 (Tris Base #BP152-1, Fisher bioreagents, HCL

#20255.290, VWR), 10 mM NaCl (#S/3161/60, Fisher Chemical), 0.5% NP-40 (#85124, ThermoFisher Scientific), 1x Complete Proteinase Inhibitor (#11697498001, Roche)] for 15 min. Nuclei were washed and re-suspended in nuclei lysis buffer (50 mM Tris-HCl pH7.5 (Tris Base #BP152-1, Fisher bioreagents, HCL #20255.290, VWR), 10 mM EDTA (#20301.290, VWR), 1% SDS (#MB11601, NZYTech), 1x Complete Proteinase Inhibitor (#11697498001, Roche)). Chromatin was sheared using a BioruptorPlus (Diagenode) device with the following cycling conditions: 10 min high–30 s on, 30 s off; 15 min on ice; 10 min high–30 s on, 30 s off. The sonicated chromatin had a size in the range of 100–500 bp and was incubated overnight at 4 °C with the anti-H3K27ac antibody (1:2, #ab4729, Abcam). Samples were incubated for 1 h at 4 °C with Dynabeads Protein G for Immunoprecipitation (#10003D, Invitrogen, ThermoFisher Scientific). Final DNA was purified with MinElute (#28004, Qiagen) and sequenced on Illumina HiSeq 2000 platform.

*ATAC-seq.* ATAC-seq was performed as previously described[88], with minor changes. Whole pancreas was dissected from 2 to 3 adult zebrafish (both genders and with 12–24 months). Following cell lysis, 50000-100000 nuclei were submitted to tagmentation with Nextera DNA Library Preparation Kit (#FC-121-1030, Illumina). ATAC-seq libraries were amplified using KAPA HiFi HotStart PCR Kit (#KK2500, Roche) with the primers Ad1, Ad2.2 and Ad2.3[45], and further purified with PCR Cleanup Kit (#28104, Qiagen).

*4C-seq.* 4C-seq was performed as previously described[88], with minor alterations. Whole pancreas was dissected from 6 to 12 adult zebrafish (7–15 × 10⁶ cells; both genders and with 12–24 months), kept on ice in PBS [137 mM NaCl (#S/3161/60, Fisher Chemical), 2.7 mM KCl (#2676.298, VWR), 10 mM NaHPO4 (#1.06342.0250, Merk), and 1.8 mM KH2PO4 (#1.06585.1000, Merk)] with 1x Complete Proteinase Inhibitor (#11697498001, Roche), fixed in 2% formaldehyde (#F1635-500ML, Sigma-Aldrich) for 10 min, and stored at −80 °C. Cell lysis was performed on ice, with a 15 mL Tenbroeck Homogenizer, not exceeding 10 min. Ligation was performed with 60U T4 DNA Ligase (#EL0012, ThermoFisher Scientific). The restriction enzymes used were DpnII (#R0543M, NEB) and Csp6I (#ER0211, ThermoFisher Scientific) for the first and second cuts, respectively. Chromatin was purified by Amicon Ultra-15 Centrifugal Filter Device (#UFC901024, Milipore). 4 C libraries were prepared for Illumina sequencing by the Expand Long Template Polymerase (#11759060001, Roche) with primers targeting the TSSs of each gene and including Illumina adapters (Supplementary Dataset 4c). Final PCR products were purified with the High Pure PCR Product Purification Kit (#11796828001, Roche) and AMPure XP PCR purification kit (#B37419AB, Agencourt AMPure XP).

*HiChIP-seq.* HiChIP-seq was performed as previously described[89], with minor alterations. Whole pancreas, from both genders and with 12–24 months, was dissected, fixed in 1% formaldehyde (#F1635-500ML, Sigma-Aldrich) and cells lysed as described for 4C-seq. Immediately after lysis, samples were washed with HiChIP Wash Buffer [Tris-HCl pH 8 50 mM (Tris Base #BP152-1, Fisher bioreagents, HCL #20255.290, VWR), NaCl 50 mM (#S/3161/60, Fisher Chemical), EDTA 1 mM (#20301.290, VWR)]. Chromatin was sonicated using the BioruptorPlus (Diagenode) with the following cycling conditions: 10 min high–30 s on, 30 s off; 15 min on ice, to obtain a size in the range of 100–500 bp. Samples were incubated with anti-H3K4me3 antibody (1:5, #AB8580, Abcam) and Dynabeads Protein G for Immunoprecipitation (#10003D, Invitrogen, ThermoFisher Scientific) and purified with DNA Clean and Concentrator columns (#D4004, Zymo Research). Up to 150 ng of the DNA was then biotinylated with Streptavidin C-1 beads (#65001, ThermoFisher Scientific). Tagmentation was performed using Nextera DNA Library Preparation Kit (#FC-121-1030, Illumina). Libraries were amplified using NEBNext® High-Fidelity 2X PCR Master Mix (#M0541S, NEB)

with primers Ad1, Ad2.23 and Ad2.24[45]. The final product was purified with DNA Clean and Concentrator kit (#D4004, Zymo Research).

*Generation of plasmids for enhancer assays.* Putative enhancer sequences were selected based on the overlap between H3K27Ac ChIP-seq and ATAC-seq signal in non-coding regions within the landscape of each pancreas-relevant gene. Sequences were PCR amplified from zebrafish genomic DNA using the primers in Supplementary Dataset 4b (designed to span the ChIP-seq and ATAC-seq signals) (Sigma-Aldrich), with the proof-reading iMax ™ II DNA polymerase (#25261, INtRON Biotechnology) following the manufacturer's instructions for a standard 20 μl PCR reaction. PCR products were visualised by electrophoresis on an 1% agarose gel, the bands excised, purified with NZYGelpure kit (#MB011, NZYTech) and cloned into the entry vector pCR®8/GW/TOPO (#250020 Invitrogen, ThermoFisher Scientific) according to manufacturer's instructions. The vectors were then recombined into the destination vectors Z48[90], for transient enhancer assays, and ZED[91,92], for stable transgenic lines, using Gateway® LR Clonase® II Enzyme mix (#11791020, Invitrogen, ThermoFisher Scientific), following manufacturer's instructions.

Standard chemical transformation was performed with MultiShotTM FlexPLate Mach1TM T1R (#C8681201, Invitrogen, ThermoFisher Scientific), grown O.N. at 37 °C. Vector selection was performed with 100 μg/ml Spectinomycin (#S4014, Sigma-Aldrich) in the growth medium for the pCR®8/GW/TOPO vectors, or 100 μg/ml Ampicillin (#624619.1, Normon) for the Z48 and ZED vectors. Plasmids were purified with NZYMiniprep kit (#MB010, NZYTech) and confirmed by Sanger sequencing using the primers in Supplementary Dataset 4b. Final plasmids were purified with phenol/chloroform (#A931I500 and #C/4920/15, Fisher Chemical) and concentration was determined by NanoDrop 1000 Spectrophotometer (ThermoFisher Scientific).

*In vitro mRNA synthesis, microinjection and transgenesis.* Z48 and ZED zebrafish lines were generated through TOL2-mediated transgenesis[93]. TOL2 cDNA was transcribed by Sp6 RNA polymerase (#EP0131, ThermoFisher Scientific) after Tol2-pCS2FA vector linearization with NotI restriction enzyme (#IVGN0016, Anza, Invitrogen, ThermoFisher Scientific). TOL2 mRNA was purified as previously described[91]. One-cell stage embryos were injected with 1nL solution containing 25 ng/μL of transposase mRNA, 25 ng/μL of phenol/chloroform (#A931I500 and #C/4920/15, Fisher Chemical) purified plasmid (Z48 or ZED), and 0.05% phenol red (#P0290, Sigma-Aldrich).

*Luciferase reporter assays.* The h.A.E4 enhancer were cloned in the pGL4.23 GW[luc2/minP] vector (Addgene #60323[2]) and co-transfected along with pNL1.1PGK[Nluc/PGK] (Promega #N1441) in hTERT-HPNE cells using Lipofectamine 3000 (#L3000008, ThermoFisher), following manufacturer's instructions. The promoter of tyrosine kinase was cloned into the pGL4.23 GW[luc2/minP] vector and used as positive control (pGL4.23 GW[luc2/Tkp])[94]. As negative control, a region without marks of enhancer activity (H3K27ac) was cloned into the pGL4.23 GW [luc2/minP] vector. The luciferase activity was measured 48 h post transfection with the Nano-Glo Luciferase Assay System (#N1610, Promega) on a Synergy 2 microplate reader (BioTek). Results were presented as luc2/Nluc ratios, relative to the negative control. Two-sided *t*-test was used to calculate statistical significance. Three independent replicates of the transfection were performed.

*Cas9 target design, sgRNA synthesis and mutant generation.* Small guide RNAS (sgRNAs) targeting regions flanking zP.E3 were designed using the CRISPRscan algorithm[95] to include H3K27ac ChIP-seq and ATAC-seq signal (Supplementary Dataset 4f). Oligonucleotides (1.5 μL at 100 μM each, from Sigma-Aldrich) were annealed in vitro by incubation at 95 °C for 5 min in 2x Annealing Buffer [10 mM Tris, pH7.5-8.0 ((Tris Base #BP152-1, Fisher bioreagents, HCL #20255.290, VWR), 50 mM NaCL (#S/3161/60, Fisher Chemical), 1 mM EDTA (#20301.290, VWR)] followed by slow cooling at RT, and inserted into 100 ng of pDR274 vector (#42250, Addgene) previously cut with BsaI (#IVGN0366, Anza, Invitrogen, ThermoFisher Scientific; 1:10). The pDR274 vectors carrying sgRNA sequences were linearized with HindIII (#IVGN0168, Anza, Invitrogen, ThermoFisher Scientific; 1:10), purified with phenol/chloroform (#A931I500 and #C/4920/15, Fisher Chemical) and transcribed with T7 RNA polymerase (#EP0111, ThermoFisher Scientific). Final sgRNAs were purified as described previously[91]. One cell-stage zebrafish embryos were co-injected with two sgRNAs (40 ng/μl each) and Cas9 protein (300 ng/μl; #CP01-50 PNA Bio, Inc). Zebrafish mutant lines for zP.E3 deletion were generated using the combinations sgRNA1 + sgRNA2 (sgPair1) and sgRNA3 + sgRNA2 (sgPair2; Supplementary Dataset 4f). Enhancer deletions in zebrafish were detected with PCR using HOT FIREPol DNA Polymerase (#01-02-00500, Solis BioDyne) with the flanking primers used to amplify the enhancers (Supplementary Dataset 4b). PCR products were visualised by electrophoresis in 2% agarose gel and confirmed by Sanger sequencing. The mutations were further verified in the F1 mutants by sequencing.

*CRISPR-Cas9 in human cell lines.* Four single-guide sequences named sg1, sg2, sg3, sg4, targeting hA.E4 enhancer were designed (Supplementary Dataset 4f). sg1 and sg3 were designed upstream of the enhancer, while sg2 and sg4 were designed downstream, based on H3K27ac ChIP-seq and ATAC-seq signal. Two

complementary oligonucleotides containing the single-guide sequences and BbsI ligation adapters were synthesised by Sigma. Two single-guide sequences designed to delete a genomic region lacking enhancer activity marks (based on H3K27ac), named ng1 and ng2, were used as negative control of the experiment[16]. Oligonucleotides were annealed in T4 Ligation Buffer (ThermoFisher Scientific). sgRNA was cloned into the BbsI-linearized pSpCas9-T2A-GFP (#R3539S, NEB; 48138, Addgene) (sg1, sg3, ng1) and pU6-(BbsI)CBh-Cas9-T2A-mCherry (#64324, Addgene) (sg2, sg4, ng2) vectors using T4 Ligase (ThermoFisher Scientific). The plasmid DNA was purified with Plasmid Midi Kit (#12143, Qiagen).

hTERT-HPNE cells were seeded in six-well plates ($1.1 \times 10^5$ cells/well, at early passage number) and transfected (~70–90% of confluency) using the following combinations: ng1 + ng2 (control); sg1 + sg2 (sgPair1); sg3 + sg4 (sgPair2). The transfection (1.5 μg of each sgRNA plasmid) was performed using Lipofectamine 3000 (#L3000008, ThermoFisher Scientific), according to the manufacture instructions. Then, cells were changed to fresh culture medium after 24 h. Three independent replicates of the transfection were performed. After 48 h of recovery, cells were used in subsequent experiments.

*Nucleic acid extraction from zebrafish and human cell lines.* Genomic DNA was extracted from whole zebrafish embryos at 24 hpf, after removal of the chorion, with a standard phenol-chloroform DNA extraction (#A931I500 and #C/4920/15, Fisher Chemical), and used as template for PCR amplification in order to genotype the tested conditions (Supplementary Dataset 4b). The DNA samples were resuspended in 20 μl of TE buffer with RNase [10 mM Tris, pH 8.0 (Tris Base #BP152-1, Fisher bioreagents, HCl #20255.290, VWR); 1 mM EDTA pH 8.0 (#20301.290, VWR) and 100 μg/ml RNAse (#10109142001, Sigma-Aldrich)], incubated for 1 h at 37 °C, and stored at −20 °C.

Genomic DNA from hTERT-HPNE cells was extracted 48 h after transfection and used as template for PCR amplification in order to genotype the tested conditions (Supplementary Dataset 4b).

RNA was extracted from zebrafish embryos, pancreas and muscle, with 500 μl TRIzol (#15596026, Invitrogen, ThermoScientific), following the manufacturer's instructions. Samples were incubated 30 min at 37 °C with 1 μl DNAse I (#EN0521, ThermoScientific), 1 μl 10x reaction buffer and 0.5 μl NZY Ribonuclease Inhibitor (40U/μl; # MB084, NZYTech) at 0.05 μl/μl final concentration. After adding 1 μl EDTA (#20301.290, VWR) 50 mM per 1 μg of estimated RNA, final volume was completed to 60 μl with H2O, phenol-chloroform (#A931I500 and #C/4920/15, Fisher Chemical) standard purification was performed and the RNA stored at −80 °C.

Zebrafish pancreatic progenitor cells were extracted from 48 hpf embryos, immediately following euthanasia by rapid chilling, by repeated pipetting up and down in a gentle motion with 300 μL of Ginzburg fish Ringer's solution [55 mM NaCl (#S/3161/60, Fisher Chemical), 1.8 mM KCl (#2676.298, VWR), 1.25 mM NaHCO3 (# S5761, Sigma-Aldrich)]. Embryos were allowed to settle to the bottom and the suspension containing the detached pancreatic progenitor cells and yolk was collected, washed with PBS [137 mM NaCl (#S/3161/60, Fisher Chemical), 2.7 mM KCl (#2676.298, VWR), 10 mM NaHPO4 (#1.06342.0250, Merk), and 1.8 mM KH2PO4 (#1.06585.1000, Merk)], and RNA was extracted using Quick-RNA Microprep Kit (#R10150, Zymo Research), according to the manufacturer's instructions. For Real-time qPCR, RNA samples were treated with DNaseI (#EN0521, ThermoScientific) and reverse transcribed using the iScript cDNA Synthesis Kit (#1708890, Bio-Rad) according to the manufacturer's instructions.

*Immunohistochemistry in zebrafish embryos and human cell lines.* Zebrafish embryos/larvae were euthanized by prolonged immersion in 200–300 mg/L tricaine (MS222; ethyl-3-aminobenzoate methanesulfonate, #E10521-10G, Sigma-Aldrich). Whenever necessary the chorion was removed, and the zebrafish were fixed in formaldehyde 4% (#F1635-500ML, Sigma-Aldrich) for 1 h at RT (8–12 dpf larvae) or O.N. at 4 °C (48 hpf embryos). Permeabilization was carried out by incubation with 1% Triton X-100 (#X100, Sigma-Aldrich) in PBS [137 mM NaCl (#S/3161/60, Fisher Chemical), 2.7 mM KCl (#2676.298, VWR), 10 mM NaHPO4 (#1.06342.0250, Merk), and 1.8 mM KH2PO4 (#1.06585.1000, Merk)] for 1 h at RT, followed by blocking with 5% bovine serum albumin (BSA; #MB04602, NZYTech) in 0.1% Triton X-100 (#X100, Sigma-Aldrich) for 1 h at RT. Zebrafish were incubated with the primary antibody diluted in blocking solution at 4 °C O.N., and then incubated with the secondary antibody plus DAPI (1:1000, D1306 Invitrogen, ThermoFisher Scientific) diluted in blocking solution for 4 h at RT. After each antibody incubation, embryos were washed 6 times in PBS-T (0.5 % Triton X-100 (#X100, Sigma-Aldrich) in PBS-1x[137 mM NaCl (#S/3161/60, Fisher Chemical), 2.7 mM KCl (#2676.298, VWR), 10 mM NaHPO4 (#1.06342.0250, Merk), and 1.8 mM KH2PO4 (#1.06585.1000, Merk)]) 5 min at RT. Embryos were stored in 50% Glycerol/PBS (#BP229-1, Fisher bioreagents) at 4 °C before microscopy slides preparation in the mounting medium 50% Glycerol/PBS; (#BP229-1, Fisher bioreagents)). Images were acquired with a Leica TCS SP5 II confocal microscope (Leica Microsystems, Germany; LAS AF software (v.2.6.3.8173) and processed by ImageJ software (v.1.8.0). Primary antibodies: rabbit anti-Amylase (1:50, #A8273-1VL, Sigma-Aldrich), mouse anti-Alcam (1:50, #ZN-8, DSHB) and mouse anti-Nkx6.1 (1:50, #F55A10, DSHB). Secondary antibodies: goat anti-mouse AlexaFluor647 (1:800, #A-21236 Invitrogen, ThermoFisher Scientific), goat anti-rabbit AlexaFluor568 (1:800, #A-11036 Invitrogen, ThermoFisher Scientific).

The hTERT-HPNE cells were fixed at 48 h after transfection in formaldehyde 4% (#F1635-500ML, Sigma-Aldrich) in PBS [137 mM NaCl (#S/3161/60, Fisher Chemical), 2.7 mM KCl (#2676.298, VWR), 10 mM NaHPO4 (#1.06342.0250, Merk), and 1.8 mM KH2PO4 (#1.06585.1000, Merk)] for 15 min at RT, permeabilized with 1% Triton X-100 (#X100, Sigma-Aldrich) in PBS and blocked with 2% BSA (#MB04602, NZYTech) in PBS for 20 min at RT. Incubation with primary antibody in 2% BSA/PBS (#MB04602, NZYTech) was O.N. at 4 °C and in secondary antibody plus DAPI (1:1000, D1306 Invitrogen, ThermoFisher Scientific) was 3 h at 4 °C in 2% BSA/PBS (#MB04602, NZYTech) for 3 h. Human cells were washed once after fixation and permeabilization, and three times after each incubation with primary and secondary antibodies with PBS for 10 min at RT. Fluorescence images were obtained at ×40 magnification on a Leica DMI6000 FFW microscope (v.3.7.4.23463). Primary antibody used: anti-ARID1A (1:1000; #HPA005456 Sigma-Aldrich). Secondary antibody used: anti-rabbit Alexa Fluor 647 (1:1000, #A31573, ThermoFisher Scientific). In hTERT-HPNE immunohistochemistry images, the ARID1A nuclear staining was measured for each cell GFP + /mCherry+ and normalized for the average staining of the nucleus of all other cells in the same field (ratio=ARID1A expression/mean of ARID1A expression in the field). Then, we normalized the ratios using the control values.

*Flow cytometry.* The whole pancreases were dissected from double transgenic adult zebrafish [Tg(ins:GFP, ela:mCherry), Tg(ins:GFP, gcga:mCherry), and Tg(ins:GFP, sst:mCherry)] and fixed using 4% formaldehyde (#F1635, Sigma-Aldrich) in xPBS [137 mM NaCl (#S/3161/60, Fisher Chemical), 2.7 mM KCl (#2676.298, VWR), 10 mM NaHPO4 (#1.06342.0250, Merk), and 1.8 mM KH2PO4 (#1.06585.1000, Merk)]. Cells were dissociated, on ice, using a 15 mL Dounce homogenizer in 1 mL of ice-cold sort buffer [1% EDTA (#20301.290, VWR), 2 mM HEPES (#83264, Sigma-Aldrich) pH 7.0 in 1xPBS], and then passed through a 40-μm cell strainer. Immediately following dissociation, the mCherry and GFP fluorescence were analysed on a BD FACS-ARIA™ II cell sorter (BD Biosciences).

*Statistical analysis.* Two-tailed paired Student's t-test was applied to area quantifications, and in expression analysis. Chi-square test was applied to the in vivo validation of selected putative pancreatic enhancers and TFs motif comparisons. Wilcoxon test was applied to gene-to-enhancer association by chromatin interaction points comparisons. Fisher's exact test was applied to analyse the percentage of larvae in each phenotypic class. In all analyses, $P < 0.05$ was required for statistical significance and calculated in GraphPad Prism 5 (San Diego, CA, USA).

## Processing and bioinformatic analysis

*ChIP-seq analysis.* High quality raw reads for the two replicates of H3K27ac ChIP-seq (FASTQC v.0.11.5[96], Supplementary Data 1 and 2) were aligned to the zebrafish genome (GRCz10/danRer10) using Bowtie2 (v.2.2.6) with default settings[97]. Before the alignment, the sequencing adapters were removed from the raw reads applying Skewer (v.0.2.1)[98]. The alignment file was converted into a bed file(Bedtools v.2.27)[99] and the data extended 300 bp, bigwig tracks generated and uploaded to UCSC Genome Browser (Fig.1b). Highly enriched regions (peaks) were obtained by MACS14 (v.1.4.2) with the parameters "--nomodel, --nolambda and --space=30"[100]. During the ChIP-seq analysis the two replicates were processed independently. Reproducibility of the two biological replicates was measured by Pearson's correlation coefficient[101] in R. The same pipeline was applied to analyse human dataset from the ENCODE project (https://www.encodeproject.org/): ENCSR340GAZ; ENCSR748TFF. Regarding the embryo ChIP-seq datasets from the work by Bogdanovic and colleagues[47], the data processed by the authors was used.

*Identification of putative enhancers.* To identify the best putative active enhancers in the zebrafish adult pancreas, we intersected the peaks from the two H3K27ac ChIP-seq replicates, generated by peak calling, selecting only the enriched regions present in both replicates (Bedtools intersect v.2.27 with the default parameters[99]). Since H3K27ac is also present in promoter regions, we excluded peaks overlapping with TSS by intercepting our set of putative active enhancers with the TSS coordinates (Bedtools intersect with the parameter "-v"). To determine the presence of unreliable peaks, a "blacklist" was generated using H3K27ac ChIP-seq of different zebrafish tissues to identify putative false positive peaks. The used datasets from the DANIO-CODE consortium were the following(https://danio-code.zfin.org).: DCD002894SQ, DCD002921SQ, DCD003653SQ, DCD003654SQ, DCD003671SQ and DCD002742SQ. Then, MACS software was performed in these datasets using the same parameters described in the last section and the peaks that were present in at least 5 out 6 datasets were selected. This analysis generated 156 peaks, from which 102 overlapped with 69 peaks from the list of 14,753 putative pancreatic enhancers, representing less than 0,5% of the total dataset. We have used a published human "blacklist" of unreliable peaks[102] and observed that these represent 192 out of 102,548 of the human pancreas H3K27ac ChIP-seq called peaks (0.2% of the identified peaks). The zebrafish and human "backlist" of peaks is included in Supplementary Dataset 1o and annotated in Supplementary Dataset 1a.

The genomic distribution of putative enhancers was performed using the annotatePeaks.pl module of HOMER(v.4.11.1)[103](Fig. 1c). The adult pancreas putative active enhancer dataset (PsE+DevE) was crossed with the H3K27ac zebrafish embryonic dataset (dome, 80% epiboly, 24 hpf and 48 hpf) (Supplementary Dataset 4g)[47] to identify enriched regions present only in adult

pancreas (PsE) (Fig. 1d). All genomic intersections were performed using Bedtools "intersect"[99]. We superimposed the H3K27ac mapped reads from adult pancreas and the embryonic dataset with the adult pancreas H3K27ac peaks using seqMINER (v1.3.4) with default settings (Fig. 1d), showing read densities ±5 kb from the acetylation peak centre[104]. Gene enrichment and functional annotation of our dataset were obtained with GREAT (v.3.0.0)[48,49], using the basal plus extension association rule (proximal: 5 kb upstream, 1 kb downstream, plus distal: up to 1000 kb (Supplementary Fig. 2b).

*ATAC-seq analysis.* High quality raw reads for the two replicates of pancreas ATAC-seq (FASTQC v.0.11.5)[96] were trimed for adapter sequences using Skewer (v.0.2.1)[98]. All libraries were sequenced on Illumina HiSeq 2500 platform and raw reads were mapped to the reference zebrafish genome (GRCz10/danRer10) using Bowtie2 (v.2.2.6) with parameters "-X 2000 and --very-sensitive"[97]. To avoid clonal artefacts, the duplicated mapped reads were removed using Samtools (v.1.9)[105]. Mapped reads were filtered by the fragment size (≤120 bp) and mapping quality (≥10). For a better visualisation, data were extended 10 bp, generated bigwig tracks and uploaded to the UCSC browser (Fig. 1b). To call for enriched regions, MACS2 (v.2.1.0)[100] was used with the parameters "--nomodel, --keep-dup 1, --llocal 10000, --extsize 74, --shift −37 and -p 0.07". For the ATAC-seq analysis, the two replicates were processed independently. Reproducibility of the biological replicates was measured using the Pearson's correlation coefficient[101] in R. Then, we applied the Irreproducible Discovery Rate (IDR, v.2.0.4) in order to obtain a confident and reproducible set of peaks[106]. The same pipeline was applied to analyse human dataset from ENCODE project (https://www.encodeproject.org/; ENCSR340GAZ; ENCSR515CDW) and ATAC-seq dataset from the work by Bogdanovic and colleagues[47].

*4C-seq analysis.* 4C-seq libraries were first inspected for quality control using FASTQC[96] (v.0.11.5, Supplementary Data 3-5) and demultiplexed using the script "demultiplex.py" from the FourCSeq package[107], allowing for 1 mismatch in the primer sequence. 4C-seq data were analysed as previously described[108,109]. Briefly, reads were aligned to the zebrafish genome (GRCz10/danRer10) using Bowtie (v.1.1.2)[110], keeping only uniquely mapping reads (-m 1). Reads within fragments flanked by restriction sites of the same enzyme or if fragments smaller than 40 bp were filtered out. In addition, reads falling ±5 kb from the viewpoint were filtered out. Mapped reads were then converted to reads-per-first-enzyme-fragment-end units, and smoothed using a 30 fragment mean running window algorithm (Figs. 4a and 5a).

*HiChIP-seq analysis.* H3K4me3 HiChIP-seq analysis from paired-end fastq files to pairs of interacting chromatin fragments were performed using a custom python script based on the default function of the pytadbit python library[111]. This library first uses GEM mapper (v.3.6)[112] to map paired reads independently to the zebrafish reference genome (GRCz10/danRer10, flags used by GEM mapper --max-decoded-strata 1; --min-decoded-strata 0; -e 0.04). Then, reads are associated to a particular restriction fragment and paired together according to their read names. Once the reads are paired, the pairs of reads are filtered so that only those belonging to different restriction fragments are kept. Compressed sparse matrix files in cooler and hic formats were generated from those filtered reads using Cooler ("cload pairix" utility) and Juicer tools ("pre" utility) respectively for both visualisation and further analysis. From the hic file we obtained contact matrices detailing the coordinates of 2 interacting 5 kb chunks and the respective number of interactions, using Juicer tools ("dump" utility) and filtering for ≥2 interactions between chunks ≤100 kb apart. To predict the target promoters of putative active enhancers, only contacts connecting zebrafish pancreas active TSSs and putative active enhancers given by H3K27ac ChIP-seq peaks from whole pancreas, adult pancreas (PsE), developing pancreas (DevE) and the different enhancer clusters (C1-C4) were selected. An output table was produced with genes targeted by enhancers, per enhancer cluster (Supplementary Dataset 3a-g). Custom scripts are provided in a GitLab repository (https://gitlab.com/rdacemel/pancreasregulome).

*Identification of active promoters.* H3K4me3 sequencing datasets (2 replicates performed in the HiChIP assay; Supplementary Data 6–9) were aligned to the zebrafish genome (GRCz10/danRer10) using Bowtie2 (v.2.2.6) with default settings. Highly enriched regions (peaks) were obtained by MACS14 (v.1.4.2) algorithm with the parameters "--nomodel, --nolambda and --space=30"[100]. Then, the peaks present in both replicates were filtered with the transcription start site (TSS) position to identify the active promoters using Bedtools "intersect"(v.2.27)[99].

*RNA-seq analysis.* Total RNA extracted from adult zebrafish (exocrine, endocrine and muscle) and sequenced on Illumina HiSeq 2000 platform was inspected for quality control using FASTQC[96] (v.0.11.5, Supplementary Data 10–17). Then, sequences were trimmed to remove adaptors, sequencing artefacts and low-quality reads (Q < 20)[113]. The BWA-MEM software (v.0.7.17) was used to map the clean reads to the reference genome (ZV9/danRer7) with the parameters "-w 2 and -c 3"[114]. Gene expression was measured from the mapped reads using HT-seq-count (v0.9.0)[115]. In addition, two public RNA-seq datasets were used (Supplementary Dataset 4g).

*Gene expression barplots.* The average expression of genes associated with each enhancer cluster (PsE, DevE, C1-C4), as defined by HiChIP, was compared to the

average expression of all genes present in the RNA-seq datasets using R and ggplot for drawing barplots (Fig. 2a, Supplementary Fig. 2c, Supplementary Dataset 3h, Fig. 2a R in https://gitlab.com/rdacemel/pancreasregulome).

*Identification of Human/zebrafish syntenic blocks.* Human/zebrafish syntenic blocks were defined by two aligned regions between both species that kept their relative position among each other. Pre-existing alignments available in the UCSC genome browser were used. Then, enhancers were searched within these blocks in both species.

*Conservation between zebrafish and human and PhastCons scores.* To obtain the percentage of zebrafish putative active enhancers conserved with human, the coordinates of putative active enhancers from adult zebrafish pancreas and embryos at different development stages (GRCz10/danRer10) were used as input to the UCSC genome coordinate conversion tool (https://genome.ucsc.edu/cgi-bin/hgLiftOver, liftover (v.1.04.00) to hg19, October 2019) (Fig. 3a). To visualise the conservation of the respective sequences, liftOver (v.1.04.00) to hg38 was done and their average PhastCons conservation score plotted (Fig. 3b). For this, we downloaded PhastCons scores in bigWig format from a 100-way multiple species alignment of 99 vertebrates against human (hg38) (hg38.phastCons100way.bw, October 2019)[116] and converted to BedGraph text format using the UCSC's utility *bigWigToBedGraph* (v.1.04.00). Then, the Bedtools[99] suite (v.2.27) was used to intersect and map different putative enhancer clusters in bed format with the conservation scores, storing for each putative enhancer the median and average PhastCons score. To know for which of them overlap putative active enhancers in human pancreas, we used the Bedtools "intersect" tool with default ≥1 bp of overlap (Fig. 3b, blue). To calculate the Fold Change (FC) of the graph displayed in Fig. 3c, we have quantified the number of zebrafish H3K27ac positive sequences aligned with the human genome that also showed H3K27ac signal in human pancreas (ZebraHumanK27). As a control, we have performed a similar analysis, randomizing the aligned human sequences, quantifying the number of those that also showed H3K27ac signal in human pancreas, repeating this operation $10^5$ times (randomZebraHumanK27). FC was calculated by the ratio: ZebraHumanK27/ [average(randomZebraHumanK27)] (Supplementary Dataset 3q). This was performed for the different populations of zebrafish enhancers (Pancreas, PsE, DevE, and embryo).

*Transcription factor binding motifs enrichment.* To refine our data, H3K27ac peaks were filtered with the ATAC-seq peaks. Then, the transcription factor binding site (TFBS) predictor program Hypergeometric Optimization of Motif EnRichment (HOMER v.4.11.1) was used to identify conserved sequence motifs enriched[103]. To evaluate our results, we also analysed, using HOMER, different acetylation data from: human pancreas, human ventricle, zebrafish embryos at 24 hpf and at dome +80%epiboly (Supplementary Dataset 3t, u and 4g). From the resulting analysis, we selected the top 140 enriched motifs for each dataset. These motifs were selected based on ranking and the groups were compared by performing hypergeometric enrichment tests. Fisher exact test from GraphPad Prism 7 (v.7.04) was performed to evaluate the enrichment in 25 known pancreas-related TFs (with Bonferroni correction). The HOMER software was also similarly applied in PsE, C1, C2, C3 and C4 in order to identify TFBS (Supplementary Fig. 3f, g, Fig. 4 and Supplementary Dataset 3t, u).

*Identification of super-enhancers.* We applied ROSE (Ranking Ordering of Super-Enhancers, v.1) algorithm with default parameters to define super-enhancers in our whole pancreas acetylation data and in human pancreas acetylation data[58]. Then, we performed gene ontology analysis in both data using PANTHER software (v.14.0, on April 2019) and compared the molecular functions obtained (http://pantherdb.org). To identify the genes shared between the two groups, we identified the human orthologous genes in our zebrafish list using Biomart (https://www.ensembl.org/biomart; on April 2019) and compared the groups (Fig. 3e, Supplementary Fig. 3h).

*Disease association enrichment of genes from different enhancer clusters.* To know whether the genes interacting with the pancreatic enhancer sets (PsE, C1-C4) include homologs of human genes associated with pancreatic diseases in a higher proportion than expected by chance, we took human gene-disease associations from DisGeNET (v.6.0)[56], for the available pancreatic diseases. Then, we derived for each disease, the set of zebrafish genes homologous to the human disease-associated genes. In detail, pancreatic diseases and their associated genes were selected from the file containing all gene-disease links from DisGeNET (all_gene_disease_associations.tsv, downloaded from the DisGeNET website on April 2019, v6.0, http://www.disgenet.org/, Integrative Biomedical Informatics Group GRIB/ IMIM/UPF), filtering for associations with a score > 0.1 to exclude those based only on text-mining. The disease search term used was "pancrea*", followed by manually filtering for pancreas-related diseases and their human associated genes.

Gene annotations were obtained from Ensembl via BioMart on April 2019 selecting protein coding genes in zebrafish and gene homologs between human and zebrafish. We required a minimum of 15 zebrafish genes relating to a disease to avoid significant gene set enrichments only due to small group ratios without real over/under representations, yielding 16 pancreatic diseases totalling 836 zebrafish homologs of human genes associated to pancreatic diseases

(Supplementary Dataset 3r). To check whether the genes interacting with various enhancer clusters (Embryo only, C1, C2, C3, C4, PsE) are enriched for pancreas disease-association, we performed hypergeometric tests for gene set enrichment with the 16 pancreatic diseases left (R phyper function, X: number of genes in disease Ai and in enhancer set Bi; M: number of genes in disease Ai, N: non-disease genes – number of zebrafish protein coding genes minus M; K: number of genes in enhancer set Bi). The R package "qvalue" was used to correct for multiple testing using FDR and convert unadjusted $p$-values into $q$-values[117]. Hypergeometric enrichment was obtained as the ratio "(number disease genes in clusterX/number of genes in clusterX)/(number disease genes/number of protein coding genes)". Finally, diseases with an absolute enrichment ≥ 1.5 and a $q$-value ≤ 0.05 were considered significantly enriched in the respective cluster (Fig. 3d).

**Reporting summary**. Further information on research design is available in the Nature Research Reporting Summary linked to this article.

## Data availability

All raw sequencing data generated within this study has been submitted to ENA under accession number "PRJEB40292". The analysed data are available on "USCS browser [http://genome-euro.ucsc.edu/s/VDR_group_public_data/Carrico_et_al_2020_ZebrafishPancreasRegulome]" and in Supplementary material.

Other datasets used in this study can be downloaded from ENCODE project (https://www.encodeproject.org/): ChIP-seq and ATAC-seq of Human pancreas "ENCSR340GAZ", ChIP-seq and ATAC-seq of left ventricle "ENCSR464TTP"; from Expression Atlas: data (http://www.ebi.ac.uk/gxa/experiments/): RNA-seq of zebrafish development stages "E-ERAD-475"; NCBI Gene Expression Omnibus (GEO)(https://www.ncbi.nlm.nih.gov/geo/): ChIP-seq of developmental stages of zebrafish "GSE32483"; European Nucleotide Archive (ENA) browser(https://www.ebi.ac.uk/ena): RNAseq of the pancreatic acinar, alpha, beta and delta cells from zebrafish "PRJEB10140", RNA-seq of developmental stages of zebrafish "PRJEB12296"; "PRJEB7244"; "PRJEB12982". ChIP-seq from the DANIO-CODE consortium to create the blacklist were the following(https://danio-code.zfin.org): "DCD002894SQ", "DCD002921SQ", "DCD003653SQ", "DCD003654SQ", "DCD003671SQ" and "DCD002742SQ". All other relevant data supporting the key findings of this study are available within the article and its Supplementary Information files or from the corresponding author upon reasonable request. Source data are provided with this paper.

## Code availability

The custom code for analysis of optical action potential traces is available in gitbub (https://gitlab.com/rdacemel/pancreasregulome)[118] and in Zenodo (https://doi.org/10.5281/zenodo.6340878).

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

## Acknowledgements

We thank to the i3S Scientific Platform: Advanced Light Microscopy, member of PPBI (POCI-01-0145-FEDER-022122); Translational Cytometry; Cell Culture and Genotyping and i3s HPC facility. We also thank to Carla Oliveira for statistical support; Mafalda Sousa for the automated cell analysis; Guilherme Cardoso from the Histology Service and Isabel Guedes for maintenance of the zebrafish lines. From CABD, we thank to Elisa de la Calle-Mustienes (ChIP-seq), Sandra Jimenez Gancedo (ChIP-seq), Ana Fernandez-Minñán (4C-seq) and Ensieh Farahani (ATAC-seq) for protocol support.

This study was supported by the European Research Council (ERC) under the European Union's Horizon 2020 research and innovation programme (ERC-2015-StG-680156-ZPR and ERC-2016-AdG-740041-EvoLand to J.L.G.-S.). J.B. is supported by an FCT CEEC grant (CEECIND/03482/2018). J.L.G.-S. is supported by the Spanish Ministerio de Economía y Competitividad (BFU2016-74961-P), the Marató TV3 Fundacion (Grant 201611) and the institutional grant Unidad de Excelencia María de Maeztu (MDM-2016-0687). R.B.C. was funded by FCT (ON2201403-CTO-BPD), IBMC (BIM/04293-UID991520-BPD) and EMBO (Short-Term Fellowship). J.Tx. (SFRH/BD/126467/2016), M.D. (SFRH/BD/135957/2018), A.E. (SFRH/BD/147762/2019), and F.J.F. (PD/BD/105745/2014) are PhD fellows from FCT. M.G. was supported by the EnvMetaGen project via the European Union's Horizon 2020 research and innovation programme (grant 668981). This work was funded by National Funds through FCT—Fundação para a Ciência e a Tecnologia, I.P., under the project UIDB/04293/2020".

## Author contributions

J.B. designed and supervised the study. J.L.G.-S. supervised the work and gave important inputs to the study. R.B.C. obtained biological material and generated ATAC-seq, ChIP-seq, 4C-seq and HiChIP data from zebrafish pancreas. R.B.C. and F.J.F. collected biological material and generated RNA-seq data from zebrafish pancreas and muscle. J.T., M.G., R.D.A., P.N.F. and J.Tx. performed computational analyses and data interpretation. R.B.C., M.D., A.E. and D.R. performed enhancer-assays in zebrafish and CRISPR-Cas9 in zebrafish as well as immunohistochemistry, microscopy acquisition and analysis. J.Tx. performed transfection, CRISPR-Cas9 and image acquisition in human cell lines with support of F.F. F.C. contributed with histology of human pancreas. T.F. and J.M. contributed for plasmid and zebrafish lines generation. J.B. wrote the manuscript with input from all authors and all contributed for the development and discussion of the work.

## Competing interests

The authors declare no competing interests.
