## [Peer Review File · Nature Communications]

REVIEWER COMMENTS

Reviewer #1 (Remarks to the Author):

This is an interesting study by Bordeira-Carriço et al. who aimed to uncover similarities between the pancreases of humans and zebrafish at chromatin level, with the goal of better modelling and understanding genetic susceptibility to human disease.

The work presented throughout the manuscript is of high quality and the methodology is presented thoroughly, including the inclusion of quality control checks for all sequencing experiments.

I have however a few concerns in relation to the broader message that the authors are trying to convey with this work. I provide a point-by-point review below:

1. I would argue that the current evidence from GWAS pointing towards a key contribution of pancreatic islet enhancers is a lot more than “very limited” as the authors point out in the introduction. There have been several reports showing enrichment of diabetes variants specifically at adult human islet enhancers (see for example Parker et al 2013, Pasquali et al 2014, Fushbergen et al 2016, Mahajan et al 2018, Khetan et al 2018, Greenwald et al 2019, Miguel-Escalada et al 2019). And quite a few reports have shown some line of experimental evidence that specific islet enhancer variants are functional regulatory variants (Gaulton et al. 2010, Roman et al 2017, Kycia et al 2018, Khetan et al 2018, and several of the papers listed above as well). Multiple reviews have also approached this topic in recent years. Therefore, a more nuanced statement in this section, elaborating on the fact that there is still very little experimental/in vivo interrogation of diabetes risk variants at islet enhancers would be more appropriate to justify the present study.
2. In figure 1, the authors compare the pancreatic structure and cellular composition between human and zebrafish. This is a very valid point, but not novel. The authors should acknowledge previous art in this field. Furthermore, the comparison provided is not as detailed as the text suggests. For example, pancreatic islets contain several different endocrine cell types, which are important for glucose homeostasis, but only insulin positive cells are counted. This accompanying supplementary figure also lacks human quantifications for comparison. A better appreciation of the study design limitations should be provided and of previous work using zebrafish as a model to study pancreatic disease would have been desirable.
3. The authors set out to characterise the enhancer landscape of the zebrafish pancreas. To identify tissue-specific enhancers, the authors compare adult pancreatic enhancers with whole embryos. Given that enhancers are not only spatially restricted, but also temporally, wouldn't comparing pancreatic enhancers with enhancers active in other adult tissues address this point better?
4. The analysis of DevE enhancers is rather confusing. It is mentioned that the results suggest that DevE enhancers control gene expression in both adult and developmental settings. Given that the authors

defined this set of enhancers as being active in both stages, isn't this expected?

5. Figure 3c, the authors are claiming that there is enrichment of overlap of H3K27ac shared signal between enhancer sequences that are conserved between the two species, but only fold-changes are presented. Have the authors derived p-values for these fold-changes? The graphical representation of this result is also confusing, why is there a dotted line between the different comparisons?

6. Working with zebrafish has its obvious limitations, but could the authors expand on their thoughts on what can really be retrieved from enhancer maps derived from full embryos and full pancreases? What proportion of pancreatic developmental enhancers can be truly captured in a mixed population of cells from the full embryo? Similarly, what proportion of pancreatic islet enhancers can be captured from full pancreases, given the fact that (as shown in Figure S1a) the largest bulk of pancreatic tissue is acinar, not endocrine? This point has major implications for the validity of the authors' conclusions and should be addressed in more detail in the manuscript.

7. In the motif analysis, the authors detect a very large overlap of common motifs between pancreatic enhancers and ventricle enhancers. This is rather unexpected, as I suspect that the authors used ventricle as a negative control dataset. What is the interpretation of these results? Could the observed results be driven by common cell types, such as endothelial cells? The enrichment for the endothelial master regulator ERG seems to suggest this.

8. The authors performed very interesting experiments to investigate the concept of cross-species enhancer functional equivalency. Nevertheless, I am not entirely convinced that the results presented in Figure 4b suggest functional equivalency, as the activity in ductal cells is markedly distinct for the zebrafish and human sequences. The choice of a ductal cell line to knockout the human enhancer of ARID1A is confusing, as the results from Figure 4b show that the human sequence does not act as an enhancer in ductal cells. Presumably the authors chose a ductal cell line as ARID1A is associated with ductal adenocarcinoma? This point should be more explicit. Additionally, linking back this figure to the title of the manuscript, the authors do not provide evidence associating this ARID1A enhancers to human disease. For example, are there PDAC risk polymorphisms in enhancers of ARID1A?

9. In the final part of the study, the authors focused on an enhancer of PTF1A. In the original paper describing the human pancreas agenesis enhancer mutations (Weedon et al. 2014), the human enhancer is highly tissue and temporal specific, being exclusively active in pancreatic progenitor cells. Therefore, the claim of a functionally equivalent enhancer in adult zebrafish pancreas is inconsistent with the previous report. The presentation of the data is slightly misleading, as the authors compare H3K27ac from adult zebrafish pancreas with H3K4me1 from human pancreatic progenitors. To be a true functionally equivalent enhancer, the authors should demonstrate that the enhancer is active in the developing zebrafish pancreas, not in the adult. In humans, the enhancer that harbours agenesis-causing mutations is not active in adult human pancreatic islets, nor in whole pancreas. The results obtained with the enhancer deletion are certainly interesting, but do not demonstrate functional equivalency. PTF1A is expressed in the adult pancreas, in acinar tissue. Thus, it is expected that a number of adult pancreas enhancers drive PTF1A expression in acinar tissue in the adult. The fact that a deletion of an adult enhancer causes pancreas agenesis is also not indicative of functional equivalency per se, as

PTF1A, as an important developmental gene, is expected to have multiple enhancers controlling its expression throughout development.

10. Whilst the work seems to have been motivated by the observation that regulatory sequence variants associate with pancreatic cancer and diabetes, the authors do not actually show much evidence of enrichment of type 2 diabetes common variants in shared zebrafish-human pancreatic enhancers. This is an expected outcome, as type 2 diabetes is predominantly underlined by defects in the endocrine compartment, which accounts for a minority of cells in the pancreas, but this limitation of the study should be better addressed in the discussion.

11. A final point, the authors claim that this work has identified a region at ARID1A whose mutation may increase the risk of PDAC. This may be true, but it is not clear why this specific region (the common zebrafish-human enhancer) would be more important than other enhancers detected by ATAC-seq or H3K27ac ChIP-seq in human pancreas. Is the claim that shared enhancers are somehow more important to establish/maintain pancreatic transcriptional programmes?

The major strength of this work resides in the quality of the data and in this concept of a shared lexicon between pancreatic enhancers of humans and zebrafish. This is an important outcome, relevant to the wider field. The link with human disease genetics is nevertheless questionable with the current presentation of the data and should be re-addressed in more detail by the team.

Minor points:

- It is suggested that the authors revise the title of the manuscript, as all human pancreas profiling data seems to have been retrieved from ENCODE.

- The flow of the manuscript could benefit from some restructuring. For example, it is unclear why the HiChIP experiment is presented in figure 1, when it becomes more relevant to the analyses shown in figure 2 to assign enhancers to target genes. Figure 1b could actually be improved by highlighting a specific enhancer element that interacts with *gata6*. The current 4C-seq representation does not make it clear which fragments show significant interactions with the promoter.

- There is some inconsistency between instances where “pancreas enhancers” is used instead of “pancreatic enhancers” (the latter is preferred)

- Figure 1c: the two yellow hues are very difficult to discern, but I assume that the 57.8% corresponds to intergenic? (as is mentioned in the text). I suggest changing one of the colours for clarity.

- Line 67: “which” should be “whose”

- The authors present the examples of *INSR* and *GATA6* as genes with important pancreatic functions. The manuscript would benefit by the presentation of a brief description of these genes, as Nature Comms is not a journal specialised on pancreas/diabetes biology. The authors should note that *INSR* is actually not a good example of a tissue-specific gene.

- The authors selected the top 140 enriched motifs from different enhancer datasets. This threshold seems rather arbitrary. Could the authors provide an explanation?
- Figure S2k does not seem to be referenced in the manuscript. Furthermore, this figure shows many redundant motifs, the manuscript would be more informative if it presented non-redundant motifs.
- The results from figures 4d,e are very interesting, but the downregulation is very mild (20% perhaps?). This may be due to the chosen readout. Was there a particular reason to quantify ARID1A protein levels by immunostaining, rather than ARID1A by qPCR on clonal populations?

Inês Cebola

Reviewer #2 (Remarks to the Author):

Overview:

Bordeira-Carriço et al describe the discovery of putative zebrafish pancreas regulatory elements using H3K27ac ChIP-seq and compare this data to existing H3K27ac datasets obtained from whole developing zebrafish embryos. After classifying a set of putative pancreas enhancers of interest the authors link these elements to target genes using HiChIP. The authors go on to functionally test a subset of these enhancers in zebrafish. Overall the data and analyses the authors performed will be of wide interest as will the functionally validated pancreatic enhancers.

The topic is interesting, the zebrafish validation work appears to be of high quality and the authors clearly make progress in defining and validating zebrafish pancreas enhancers. However, the manuscript is in general challenging to read and the analyses do not always appear to be systematic. While there are many grammatical issues that add uncertainty to the manuscript, I am most concerned with the statements and conclusions made around the concept of functionally equivalent enhancers. I am convinced that the authors discovered examples of such enhancers (based on their functional validations) but how these results are used to make broader conclusions are not always justified by the analysis methods used. As written and presented now the paper would be stronger without trying to force the concept of functional equivalency (which I realize is not what the authors intend).

Three major concerns are that the paper does not acknowledge or use the appropriate cross species analysis methods; that the analysis and selection of enhancers for functional validation does not appear to be systematic; and the use of pancreas related disease regions/genes does not appear to be systematic. I have done my best to be specific/helpful in my comments. However, this has required me to quote parts of the manuscript which is a bit tedious and is mostly used to indicate where the comments pertain. While in many instances I could guess what the authors are saying or want to say, I could not always be sure. I have spent more time than average reviewing this paper and my comments are more than usual and intended to be constructive as possible.

Specific points:

1. When the authors start to describe their functional validations, the PsE and DevE concept never

comes back. 10 regions with strong H3K27ac and 7 regions with low H3K27ac activity were tested. Are they all DevEs? Were any PsE elements tested to confirm they are indeed only active in adult pancreas?

2. What is the logic for choosing the zebrafish and human *arid1ab*/ARID1A regulatory elements and the *ptf1a* enhancer for functional testing? Do they fall into different categories? For example, are they both PsEs? I would assume the zP.E3 belongs to DevEs as it shows activity in larval fish. What about the other distal enhancers at this locus? For example in Fig 4 there looks like the 4C data indicates an upstream enhancer that corresponds to an upstream enhancer in human. I think for these two loci it would have been relevant to acknowledge and screen all putative zebrafish and human enhancers in zebrafish to reveal which one(s) drive similar expression patterns. This would allow the authors to more definitively make their case for functional equivalency.

3. The authors have 7 sets of enhancers: PsE, DevE, PsE+DevE, C1, C2, C3, C4. The authors examined the average expression of genes interacting with enhancers belong to each of the category (Fig2a and FigS1d). However, when presenting the ratio between the average expression of genes interacting with a given category of enhancers and the average expression of all genes throughout zebrafish development (Fig2b), it did not seem to be presented for all categories (it seems that DevE and PsE+DevE are missing). Similarly, for FigS1e, DevE is missing.

4. The liftover method and alignments used are not the state of the art for zebrafish comparative genomics. Work done by many groups have established a comprehensive set conserved elements using methods including conserved synteny, ancestral reconstruction and intermediary species. These resources seem not to be used or acknowledged in the paper. While reference 19 is very relevant it seems to be used out of context in the introduction (this method does not simply take the nearest gene). Also the same reference (19) is also used as a reference for the enrichment tool GREAT later in the paper which is clearly an error. I did not have time to cross-check all the references but this is a major concern in addition to my main point that the manuscript does not seem to acknowledge key literature in the experimental design and data interpretation. For example, a quick look at the *zptf1aE3* (1.8k) region in the context of existing zebrafish conserved non-coding elements resources, show it encompasses a zebrafish conserved non-coding element which has an orthologous human region 300 kb away (hg19: chr10:23579080-23579212) from the *hPTF1AE3*. This conserved element was not mentioned in the paper and contradicts the authors statement that there was not enough sequence identity to link *zptf1aE3* to the human genome. While this region does not appear to have the enhancer features than *hPTF1AE3*, it is not mentioned by the authors and its likely there are other examples of this in the dataset. While I would expect very few mammal-zebrafish conserved elements regardless of the alignments used, a more thorough analysis and more representative acknowledgement/citation of work in this area would be needed if the aim of this paper is to make statements about lack of sequence conservation of regulatory elements and functional equivalency.

5. It is not clear how the authors defined syntenic blocks and their corresponding enhancers in methods section. For example, how can one rule out that the *zA.E3* instead of *zA.E2* is the functional equivalent of *hA.E3*?

6. To confirm the functional equivalency, the authors should consider testing if the *zA.E3* can rescue the *hA.E3* deletion cell line and restore the expression level of ARID1A.

7. To show that *zPtf1aE3* is the functional equivalent enhancer of *hPtf1aE3*, the author should show that the expression level of *PTF1A* gene is affected upon loss of the enhancer.

8. Abstract states "Most disease-associated alleles overlap with non-coding cis-regulatory elements of

DNA, suggesting that alterations in regulatory sequences contribute to pancreatic diseases.” Is this really true? I would argue that most disease associated alleles affect protein structure/function. I would agree that most GWAS hits fall in non-coding regions and that understanding how this genetic variation impacts gene expression is essential. I am sure the authors know this, and I highlight it only as an example of where a lack of precise language impacts the scientific message.

9. There are also many instances in the manuscript where two different ideas are presented within a single sentence. Also there are many examples where two adjacent sentences do not have a clear conceptual link. I am not a professional writer, but as a scientist interested in the authors work, I found this to be an issue when trying to follow the logic of the paper. Examples of this can first be seen in the abstract: “Most disease-associated alleles overlap with non-coding cis-regulatory elements of DNA, suggesting that alterations in regulatory sequences contribute to pancreatic diseases. However, the interspecies identification of equivalent cis-regulatory elements required for in vivo testing face fundamental challenges, including lack of sequence conservation.” [a) how does the first part of the sentences connect to pancreas and b) what is the connection between disease associated alleles and interspecies studies?]

10. A dramatic societal burden of pancreatic diseases is mentioned however this statement would better be supported by citations or specific diseases and numbers.

11. Intro: “However, evidences that support this hypothesis are very limited, and for the most of them, they are inferred by in vitro assays.” What hypothesis is being referred to here?

12. “Yet, the identification of interspecies functionally equivalent CREs face unsolved fundamental challenges, such as low conservation of interspecies non-coding sequences (10) and, for the minority of CREs that are conserved, their fast-evolving functionality (11)”. This is an important statement and deserves more elaboration/clarification and citations. I think this concept of shadow enhancers, which is not acknowledged by the authors, directly relates to enhancer evolution and the regulatory elements one would naturally expect to find around transcription factors that have been shown to play a key role pancreas development in humans and fish. Citing work that has compared regulatory elements between human and fish would also be warranted (e.g early work in zebrafish with RET enhancers is but one clear example of this concept PMID: 16556802 that was not acknowledged). One could almost get the impression that the functional equivalency was a new concept being proposed. I certainly believe that more work is needed on this topic and the authors have provided additional examples.

13. It seems like the authors do not really integrate their data generated but rather generate a series of important comparisons using specific datasets. This is of course fine, however I wondered why the ATAC-seq data was collected but did not seem to be used in the downstream analysis. Presumably the ATAC-seq data would give more refined putative enhancer regions for functional testing.

14. “Half of the Zebrafish Pancreas Enhancers Share Developmental Roles” – this should say putative enhancers.

15. “Interestingly, DevE presented 4 clusters (C1-4) with different activity dynamics during development (Fig.1d; Fig.S1b; Table S1).” The authors should clarify what is the interesting aspect of this.

16. “We found that PsE-associated genes have a higher average expression in a variety of pancreatic cell types when comparing to all transcribed genes, contrasting with transcription in the muscle (Fig.2a, Table S3).” It would be important to know how many such genes were detected using the HiChIP versus the previous analysis that used a defined list pancreas expressed genes. (It would be interesting to know how using HiChIP alters the list of enhancers near pancreas genes of interest compared to the proximal gene analysis).

17. "Performing a similar assay using the transcriptome of whole zebrafish embryos from 18 developmental stages (20), genes associated to DevE have shown an increased average expression comparing to all transcribed genes, with a similar dynamic to the enhancer activation during development (Fig.2b; Fig.S1e)." It is not clear what it means to perform a similar assay. Was this HiChIP-seq? If this was a typo and it was meant to say analysis, it is not clear how one can do a similar analysis if the specific data used are not clearly defined.

18. In the abstract "Among several disease-associated sequences, we identified a zebrafish ptf1a distal enhancer whose deletion generates pancreatic agenesis, demonstrating the causality of this condition in humans." The logic of this sentence is unclear and as written suggests that ptf1a zebrafish enhancer was disease associated (from the manuscript citation there was a human enhancer that was clearly shown to be disease-causing based on human genetics and functional studies). Since others have shown that homozygous human mutations in a distal PTF1A enhancer lead to pancreatic agenesis, it is not clear how studying a non-orthologous zebrafish enhancer near the ortholog of PTF1A can further demonstrate the disease causality of the homozygous deletion in the distal human enhancer. Given the demonstration that homozygous mutations in one human PTF1A enhancer causes pancreatic agenesis (PMC4131753), I fully agree the authors chose a relevant and exciting model to study. However looking over the human sequence coordinates given (either in hg19 or hg38) suggests that the authors did not test the actual enhancer that where mutations were demonstrated to cause the human disease (PMC4131753: positions 23508305A>G, 23508363A>G, 23508365A>G, 23508437A>G and 23508446A>C on chromosome 10). The choice for not including the actual disease associated enhancer in the comparative analysis of this locus and validation strategies should be explained given the emphasis on the disease association in in the abstract and the overall presentation of the paper. Given the clear pancreatic expression and function of this pancreatic transcription factor it is not surprising that mouse and human both have enhancers that drive reporter gene expression in the zebrafish pancreas. However, it is still of interest knowing what the repertoire of pancreatic enhancers are and if there are potentially redundant enhancer or "shadow enhancers" underlying PTF1A/ptf1a regulatory logic and a more systematic analysis of the enhancers within the authors regions of interest.

19. "Overall, we show that chromatin profiling can uncover interspecies functional equivalency of cis-regulatory elements". This claim suggests that a systematic analysis was performed and presented in the paper. I do not see how chromatin profiling can do this. I do agree that the authors have done excellent work to demonstrate that specific human enhancers at syntenic loci in human and zebrafish can drive similar expression, but a lot of the decisions on what to validate likely came from the authors intuition/prior knowledge rather than from the data they generated.

20. "To determine if the identified regulatory sequences are active pancreatic enhancers, we have performed in vivo reporter assays for 10 regions with strong H3K27ac signal and 7 with low levels of this mark (Fig.2c-d; Fig.S2a). From the first set, we have found that 5 out of 10 tested sequences (H3K27ac: $-\log_{10}(\text{p-value}) \geq 35$) are pancreatic enhancers (50%; Fig.2c-d and Table S4). In contrast, from the regions with low H3K27ac signal, only 1 out of 7 tested (14%, Fig.S2a) showed strong and reproducible pancreatic enhancer activity ($-\log_{10}(\text{p-value}) < 35$). These results validate the robustness of the enhancers prediction based on chromatin state."

It is not clear how these regions were chosen for testing and what defines high and low levels of H3K27ac. Without appreciating the criteria for selecting these regions the statistical tests performed are not as informative as it could be and would be difficult to reproduce. I think it is well established that cell-type enhanced H3K27ac signal is a useful criteria for selecting enhancer active in a particular cell-

type. However, its not clear here how this select set of 17 regions were chosen. (i.e. to what extent was evolutionary conservation considered or ATAC-seq signal or the literature on nearby genes considered). The reason I ask this is that the authors use these examples to make broad conclusions about predicting enhancers with “functional equivalency”. To what extent would studying putative enhancers with H3K27ac signal that surround orthologous genes with known pancreas expression or disease associations, with or without evidence for 3D chromatin interactions, serve as a predictor of functional equivalency?

24. Page 6 “Although only a minority of interspecies aligned sequences shared H3K27ac signal (Total pancreas data set: 229 out of 1842; PsE: 116 out of 1052; DevE: 113 out of 790), there is a clear enrichment comparing to random sequences (Fig.3c), although not showing a higher average sequence conservation score (Fig.3b). These results suggest that pancreatic enhancer function is not a strong constraint for sequence conservation.” There is a lot of information here that would benefit from further explanation. Intuitively, sequences that can be aligned between fish and human should show more sequence constraint than random sequences unless such a large window was taken that it obscures such sequence constraint. Could the ATAC-seq peak within the H3K27ac peaks be used for this calculation? What does “pancreatic enhancer function is not a strong constraint for sequence conservation” mean?

25. “Then, we wanted to assess whether functionally equivalent pancreatic CREs might exist between human and zebrafish, despite an overall lack of sequence conservation. To explore this possibility, we analysed if the human ortholog genes coupled to each cluster of zebrafish enhancers were enriched for human pancreatic diseases” Until this point, Clusters of enhancers or super enhancers were not mentioned and it seems odd that the authors did not acknowledge prior comparative work comparing human and zebrafish super enhancers (PMID: 27965291). It would be relevant to know how calling super-enhancers in each species, linking them to target genes with HiChIP; and then looking into aspects of conservation/functional equivalency and disease association compares to the current stratification of H3K27ac regions where adult zebrafish H3K27ac peaks where compared to a reference dataset of bulk zebrafish embryo H3K27ac CHIP-seq?

26. While performing TF analysis is an essential step and should be performed, the analysis as presented is challenging to follow and in places seems arbitrary. For example, why were 140 motifs chosen? It was not clear in the main text or methods what human data was used. Ideally the data type and a citation of the original publications of the data used in the manuscript would be given.

27. “These results suggest that the same set of TFs might operate in zebrafish and human pancreas enhancers.” I would think that it would be expected a priori that vertebrate pancreas TFs would be shared between species and that there would be ample literature to cite and suggest targets. It would be useful to clarify and/or make such expectations in the introduction.

28. Details about the number of replicates for RNA-seq for adult zebrafish were not given. The DATA tab in the supplemental spreadsheet does not indicate which biological replicates for the authors own data as well as public data were used.

27. There seems to be information missing regarding how the authors processed their own CHIP-seq data as well as published CHIP-seq data (it would not be possible to redo this analysis with the information given). How were biological replicates handled? It is mentioned for the H3K27ac that biological replicates were compared but not how the data was used (e.g. were the peaks combined or were the replicates combined prior to peak calling.) There does not appear to be any mention of the use of input controls for CHIP-seq peak calling analysis which is highly unusual. This should be clarified and

the justification for not using input controls should be clearly made in the method if this is not the case and how this impacts the data analysis. Were input controls also not used for the human data and developmental datasets utilized by the authors? This seems unusual.

28. The DisGeNET analysis seems to be a central aspect of defining regions for study in this paper. It is not clear in the main text how pancreas disease genes were obtained. Reading the methods, it was not clear how many genes the authors obtained and how the putative enhancers associated with these genes compares with the set of zebrafish pancreas enhancers. Given the prominence of the disease angle in the abstract, I think this idea should be more formally and systematically introduced and used in the paper.

29. The authors used H3K4me3 HiChIP to capture the enhancers that interacts with gene promoters and RNA-seq as a read out for gene expression. In Fig.1b, they should show a track of RNA-seq signal as well for their gata6 example.

30. In Figure 2b, the author examined the HC/AllG ratio across development. It would be interesting to add in an adult timepoint to see if there is a significant increase for the genes interacting with PsE.

31. In theory genes interacting with PsE, even if they are expressed in development, they are not activated by PsEs. In Figure 2b, the author still observed higher expression level of these genes compared to all genes in certain developmental stages (e.g. S20-25). Are these PsE genes being regulated by other sets of enhancers such as DevEs? What is the percentage of enhancers that are overlapped between PsE associated genes and DevE associated genes? For those overlapped genes, how do their gene expression change during development?

32. Is it possible for the authors to show the percent sequence identity between human and fish for the two enhancers they tested (A.E3 and P.E3)? Just wanted to get an idea of how poorly conserved they are. Or is it simply they don't align?

33. When the author deleted the enhancer for functional test, more rationale about the part of the enhancer, the TFBS motifs contained etc would be important. I am curious if and how the ATAC-seq data was used for designing these experiments.

34. In the zebrafish zPtf1aE3 deletion line, are the Foxa2 and Pdx1 binding sites still present? The author used two pairs of gRNAs to generate zebrafish mutants. How many different alleles were recovered? Are the phenotypes consistent across independent lines?

35. Enhancers regulate tissue specific gene expression. Are the enhancers tested restricted to pancreas even though some of the genes they regulate are not pancreas specific? Are they activate elsewhere? For example, arid1ab could be broadly expressed in many tissues.

Reviewer #3 (Remarks to the Author):

In this paper the authors perform a combined analysis using ATAC-seq, ChIP-seq, 4C-seq and HiChIP-seq in zebrafish and human pancreatic cells to identify interspecies functionally equivalent cis-regulatory elements. This is an innovative work that utilizes the power of inter species comparison of chromatin traits and conservation of them to identify regulatory elements regulating pancreatic genes linked to human diseases. Overall the study is robust and well presented and I would recommend it for publication in Nature Communications.

Minor comments.

1. There are a few places in the manuscript that the authors refer to interaction points of chromatin. For instance “To bypass these limitations, we profiled the chromatin state of zebrafish pancreas cells and interaction points” in lines 61,62. I assume the authors refer to chromatin interactions but this should be clarified in several parts of the text.

2. In lines 103 to 113 the authors state “We found that PsE-associated genes have a higher average expression in a variety of pancreatic cell types when comparing to all transcribed genes, contrasting with transcription in the muscle (Fig.2a, Table S3). Similar results were obtained when analysing genes associated to the remaining clusters of pancreatic enhancers (PsEs+DevE, DevE and C1-4; Fig.S1d). Performing a similar assay using the transcriptome of whole zebrafish embryos from 18 developmental stages 20, genes associated to DevE have shown an increased average expression comparing to all transcribed genes, with a similar dynamic to the enhancer activation during development (Fig.2b; Fig.S1e). These results suggest that DevE enhancers control gene expression in the adult differentiated pancreas and during development.”

- This paragraph is very confusing and hard to understand. The authors need to clarify it. Particular points are:

3. PsE associated genes have a higher expression in pancreatic cell types compared to all transcribed genes, contrasting with the muscle. Why do the authors compare the PsE associated genes transcription with the muscle? Are the authors measuring the expression of the PsE associated genes in the muscle as well? This is unclear in the text.

4. What do the authors mean with “the remaining clusters of pancreatic enhancers” and what does the PsE+DevE category refers to? This should be better explained in the main text.

5. “...genes associated to DevE have shown an increased average expression comparing to all transcribed genes, with a similar dynamic to the enhancer activation during development (Fig.2b; Fig.S1e). These results suggest that DevE enhancers control gene expression in the adult differentiated pancreas and during development.” What do the authors mean by enhancer activation?. Clarify in the main text. PsE are the one’s driving gene expression in the adult pancreas preferentially over DvE correct? The last sentence of the paragraph is confusing.

6. The authors state in lines 115-117, “From the first set, we have found that 5 out of 10 tested sequences (H3K27ac: $-\log_{10}(p\text{-value}) \geq 35$) are pancreatic enhancers (50%; Fig.2c-d and Table S4).” Can the authors comment on the fact that 50% of H3K27ac peaks are not pancreatic enhancers and if these elements have other features (accessibility, chromatin contacts) that could explain this phenomenon? Have they checked for other enhancer features as active transcription or p300 abundance? It would be interesting to include a comment on this matter somewhere in the paper as the field more or less assumes that a peak of H3K27ac will be an enhancer or promoter.

7. It would be important for the authors to discuss why if they can detect TFBS enriched at the enhancers in both human and zebra fish the sequence conservation seems to be low. Also in this regard

it would be interesting for the author's to comment what features are preserved then, if not sequence, between the enhancers across species.

** See Nature Research's author and referees' website at www.nature.com/authors for information about policies, services and author benefits.

Point-by-point response to reviewers:

Reviewer1

This is an interesting study by Bordeira-Carriço et al. who aimed to uncover similarities between the pancreases of humans and zebrafish at chromatin level, with the goal of better modelling and understanding genetic susceptibility to human disease.

The work presented throughout the manuscript is of high quality and the methodology is presented thoroughly, including the inclusion of quality control checks for all sequencing experiments.

I have however a few concerns in relation to the broader message that the authors are trying to convey with this work. I provide a point-by-point review below:

1) *I would argue that the current evidence from GWAS pointing towards a key contribution of pancreatic islet enhancers is a lot more than “very limited” as the authors point out in the introduction. There have been several reports showing enrichment of diabetes variants specifically at adult human islet enhancers (see for example Parker et al 2013, Pasquali et al 2014, Fushbergen et al 2016, Mahajan et al 2018, Khetan et al 2018, Greenwald et al 2019, Miguel-Escalada et al 2019). And quite a few reports have shown some line of experimental evidence that specific islet enhancer variants are functional regulatory variants (Gaulton et al. 2010, Roman et al 2017, Kycia et al 2018, Khetan et al 2018, and several of the papers listed above as well). Multiple reviews have also approached this topic in recent years. Therefore, a more nuanced statement in this section, elaborating on the fact that there is still very little experimental/in vivo interrogation of diabetes risk variants at islet enhancers would be more appropriate to justify the present study.*

Response:

We thank Reviewer1 for raising this point and we agree with the comment. The contribution of modifications in enhancers to increased risk of diabetes development has been shown in several works and it should not be considered “very limited”, although the number of reports focusing on pancreatic cancer is considerably smaller. We have now improved this paragraph, clarifying what

is known about this contribution and we have included a more comprehensive list of citations on these works (Parker et al 2013, Pasquali et al 2014, Mahajan et al 2018, Khetan et al 2018, Greenwald et al 2019, Miguel-Escalada et al 2019; Gaulton et al. 2010, Roman et al 2017, Kycia et al 2018, Khetan et al 2018). Additionally, and following Reviewer1's suggestion, we have now focused on the restricted number of studies that show still very little *in vivo* interrogation of diabetes risk variants at islet enhancers, to highlight the importance of the current work.

To facilitate the identification of the changes performed in the revised version of the manuscript addressing each specific point raised by Reviewer1, we have annotated the respective point in a "comment" in the manuscript version that contains "track changes". E.g. this particular change has been annotated as "Reviewer1, point 1".

2) In figure 1, the authors compare the pancreatic structure and cellular composition between human and zebrafish. This is a very valid point, but not novel. The authors should acknowledge previous art in this field. Furthermore, the comparison provided is not as detailed as the text suggests. For example, pancreatic islets contain several different endocrine cell types, which are important for glucose homeostasis, but only insulin positive cells are counted. This accompanying supplementary figure also lacks human quantifications for comparison. A better appreciation of the study design limitations should be provided and of previous work using zebrafish as a model to study pancreatic disease would have been desirable.

Response:

We thank Reviewer1 for raising this point. In line with Reviewer1's comments we have improved the description of the cellular composition of the zebrafish pancreas, establishing a comparison with human and mice pancreas. The improved description of the zebrafish pancreas also includes the main cellular components of the pancreas, with acinar cells for the exocrine part and alpha, beta and delta cells for the endocrine part. Additionally, we have included references to previous works that assess the cellular composition of human and mouse pancreas (Supplementary Fig. 1), as well as some citations that support the use of the zebrafish as a reliable model to study

pancreatic diseases. Finally, we have improved the discussion, in line with points 6, 7, 9 and 10 raised by Reviewer1, to better clarify the limitations of the current work.

3) The authors set out to characterise the enhancer landscape of the zebrafish pancreas. To identify tissue-specific enhancers, the authors compare adult pancreatic enhancers with whole embryos. Given that enhancers are not only spatially restricted, but also temporally, wouldn't comparing pancreatic enhancers with enhancers active in other adult tissues address this point better?

Response:

We thank Reviewer1 for giving us the opportunity to clarify this point. To understand which of the putative enhancers could have a higher degree of tissue specificity towards the pancreas, we have asked if those sequences also show H3K27ac mark during development. We decided to use H3K27ac ChIP-seq data-sets from whole embryos at different developmental stages (Dome, 80% Epiboly, 24hpf and 48hpf) because whole embryos comprise differentiated and differentiating cells. Therefore, these datasets allow us to assess the potential activity of enhancers in already differentiated tissue as well as non-differentiated cells using a single model, without having to specifically dissect each organ from an adult zebrafish and performing multiple tissue versus tissue comparisons. Aside from this, the availability of datasets of H3K27ac in zebrafish adult tissues is limited, and is mostly restricted to the brain, muscle, heart, intestine and testis (PMID: 27965291; PMID: 28245924; PMID: 28245924), which would not allow us to make generalizations from such comparison. For these reasons we consider that the whole embryo is an adequate and advantageous model for the purpose of this comparison. This has now been made clearer in the manuscript within the results section “Zebrafish putative pancreatic enhancers share developmental roles”.

4) The analysis of DevE enhancers is rather confusing. It is mentioned that the results suggest that DevE enhancers control gene expression in both adult and developmental settings. Given that the authors defined this set of enhancers as being active in both stages, isn't this expected?

Response:

We thank Reviewer1 for giving us the opportunity to clarify this point. Indeed, we have identified the DevE putative enhancers by identifying the presence of H3K27ac mark in the adult differentiated pancreas and during embryo development. Next, we wanted to test these predicted putative enhancers for enhancer activity *in vivo*. However, it would have been unfeasible to test the total set of putative enhancers one-by-one using enhancers reporter assays, which is why we only performed such tests for a restricted number of sequences (Fig.2; Supplementary Table 4a Fig.S3b). Therefore, to understand if the overall set of enhancer predictions were robust, we tested a correlation between the enhancer predictions with the transcription of their putative target genes. In this scenario, and as Reviewer1 pointed out, it is expected that the target genes of the H3K27ac-positive sequences have increased average transcriptional levels than the control group within the tissue/developmental stage where the H3K27ac signal was detected. As expected, we observed said correlation between enhancer activity and gene expression, both in the adult pancreas (Fig.2) and in different stages of embryo development (Supplementary Fig.2). These results, in turn, illustrate the robustness of our dataset of predicted enhancers.

We have now clarified this point in the revised manuscript within the “Functional similarities between human and zebrafish pancreatic enhancers “ section of the Results.

5) Figure 3c, the authors are claiming that there is enrichment of overlap of H3K27ac shared signal between enhancer sequences that are conserved between the two species, but only fold-changes are presented. Have the authors derived p-values for these fold-changes? The graphical representation of this result is also confusing, why is there a dotted line between the different comparisons?

Response:

We thank Reviewer1 for the constructive comment. We agree with Reviewer1 that the graphical representation is confusing, mostly because of the presence of the dotted line between the different comparisons. We have now redone this graph, excluding the dotted line.

Regarding this experiment, we wanted to determine if zebrafish putative pancreatic enhancers (H3K27ac) that aligned with the human genome showed a higher number of events of overlap with human putative pancreatic enhancers (H3K27ac) than with randomly selected sequences. Therefore, to calculate the Fold Change of the graph displayed in Fig.3c, we quantified the number of zebrafish H3K27ac-positive sequences aligned with the human genome that have H3K27ac signal in human whole pancreas tissue (ZebraHumanK27). As a control, we performed a similar analysis, randomizing the aligned human sequences, quantifying the number of those that also showed H3K27ac signal in human pancreas, and repeating this operation 10^5 times (randomZebraHumanK27). Fold Change was calculated by the ratio: ZebraHumanK27/[average(randomZebraHumanK27)], (Supplementary Table 3q). This was performed for the different populations of zebrafish enhancers (Pancreas, PsE, DevE, and embryo). For this, we do not have *p-values*. However, we recorded how many times the overlap between one random shuffle was equal to or higher than that observed between the zebrafish and human sequences, being this number 0 out of 10^5 for all the 4 groups (Pancreas, PsE, DevE, and embryo) shown in Fig.3c. This allowed us to calculate an empirical *p-value* $<1E-5$, now presented in the legend of Fig.3c.

Regarding the values of the graph plotted in Fig.3c, this data was made available in the previous version of the manuscript, in Table S3 "EnhShufflStats(FIG3C)", now renamed Supplementary Table 3q). However we did not have a reference in the text to this table. We apologize for this.

To improve this point, we have now: 1) Added a reference to this table in the manuscript, immediately after the reference to Fig.3c. 2) We have reformulated the titles of the sheets of the supplementary tables, for easy access to the reader. 3) We have improved Fig.3c legend, to better explain this assay, indicating an empirical p-value. 4) We have improved the "Methods" section, point "Conservation between zebrafish and human and PhastCons scores", to better explain this experiment, including a reference to Supplementary Table 3q. 5) We have improved the display and format of Supplementary Table 3, including renaming the corresponding sheet to "FoldChange_Fig3c".

6) Working with zebrafish has its obvious limitations, but could the authors expand on their thoughts on what can really be retrieved from enhancer maps derived from full embryos and full pancreases? What proportion of pancreatic developmental enhancers can be truly captured in a mixed population of cells from the full embryo? Similarly, what proportion of pancreatic islet enhancers can be captured from full pancreases, given the fact that (as shown in Figure S1a) the largest bulk of pancreatic tissue is acinar, not endocrine? This point has major implications for the validity of the authors' conclusions and should be addressed in more detail in the manuscript.

Response:

We thank Reviewer1 for letting us clarify these points. Regarding the first question, “*could the authors expand on their thoughts on what can really be retrieved from enhancer maps derived from full embryos and full pancreases?*”, the full pancreas enhancers maps allow us to identify enhancers in multiple cells of the pancreas, not being restricted to only one pancreatic subdomain or specific cell type. This broad pancreatic enhancer map is very advantageous since it allows us to approach different biological and biomedical questions related with different pancreatic cell types, spanning from the endocrine to the exocrine components of the pancreas. Regarding full embryos enhancers map, as approached above in point 3, raised by Reviewer1, we have used these datasets to understand the tissue specificity of enhancers and their potential to accumulate functions, apart from their possible roles in the adult differentiated pancreas.

Regarding the following point raised by Reviewer1, and that is immediately linked with the previous response “*What proportion of pancreatic developmental enhancers can be truly captured in a mixed population of cells from the full embryo? Similarly, what proportion of pancreatic islet enhancers can be captured from full pancreases, given the fact that (as shown in Figure S1a) the largest bulk of pancreatic tissue is acinar, not endocrine? This point has major implications for the validity of the authors' conclusions and should be addressed in more detail in the manuscript.*”, indeed, as pointed out by Reviewer1 (point 4), a correlation between active enhancers and transcription of target genes is expected. We tested this hypothesis by analyzing the transcription levels of the genes that are interacting with the predicted sequences within each cluster and verified that, when compared to all genes, genes interacting with DevE or PsE sequences had higher transcription levels in at least one of the tissues of the zebrafish pancreas (whole pancreas, duct,

acinar, and endocrine pancreas) (Fig.2a and Supplementary Fig.2c), or all pancreatic tissues in the case of PsE. A summary of these results is now represented in Supplementary Fig.2d for a clearer visualization. This important association between transcriptome and enhancer activity show us that our dataset of putative enhancers contains pancreatic enhancers from all these differentiated pancreatic cell types.

DevE's activity is not exclusive to the differentiated pancreas (Fig.1d) and the enhancers in this cluster likely have additional roles throughout development. These developmental roles of DevE may pertain to pancreatic progenitor cells, but we cannot state whether they do or do not, nor is it our intent to do so.

As suggested by Reviewer1 we have now expanded the text focusing on the rationale of looking to enhancer maps derived from full embryos and full pancreases. This has been included in new segments within the Results section “Zebrafish putative pancreatic enhancers share Developmental roles” and in the Discussion.

7) In the motif analysis, the authors detect a very large overlap of common motifs between pancreatic enhancers and ventricle enhancers. This is rather unexpected, as I suspect that the authors used ventricle as a negative control dataset. What is the interpretation of these results? Could the observed results be driven by common cell types, such as endothelial cells? The enrichment for the endothelial master regulator ERG seems to suggest this.

Response:

We thank Reviewer1 for raising this interesting question and for highlighting this point. Indeed, we agree with Reviewer1, that these results likely point to the presence of endothelial enhancers in our data set. This is further supported by the enrichment of genes, nearby DevEs, that show endothelial expression (please see Supplementary Table 2d-f). These results are likely explained by the intertwined presence of endothelial tissue in the whole pancreas. We have now addressed this point in the second section of the Results “Functional similarities between human and zebrafish pancreatic enhancers”, and in the Discussion section.

8) *The authors performed very interesting experiments to investigate the concept of cross-species enhancer functional equivalency. Nevertheless, I am not entirely convinced that the results presented in Figure 4b suggest functional equivalency, as the activity in ductal cells is markedly distinct for the zebrafish and human sequences. The choice of a ductal cell line to knockout the human enhancer of ARID1A is confusing, as the results from Figure 4b show that the human sequence does not act as an enhancer in ductal cells. Presumably the authors chose a ductal cell line as ARID1A is associated with ductal adenocarcinoma? This point should be more explicit. (Duct>Pancreatic cancer) Additionally, linking back this figure to the title of the manuscript, the authors do not provide evidence associating this ARID1A enhancers to human disease. For example, are there PDAC risk polymorphisms in enhancers of ARID1A?*

Response:

We thank Reviewer1 for giving us the opportunity to improve this point. We have now clarified, in the revised version of the manuscript, that: 1) *ARID1A* is associated with pancreatic adenocarcinoma in pancreatic ducts and 2) we selected the hTERT-HPNE cell line because of its relevance for the tumor suppressor role of *ARID1A*. These clarifications can now be found in the section of Results of the revised version of the manuscript “Landscape of *arid1a* reveals a potential pancreatic cancer associated enhancer”.

Regarding the doubt pointed out by Reviewer1 about the functional equivalency of the zA.E4/hA.E4 enhancers (previously annotated as zA.E3/hA.E3), we have improved Fig.4 and we performed some additional experiments: 1) We added to Fig.4a a track containing an ATAC-seq profile for hTERT-HPNE, in order to show that we have accessible chromatin in the region of the hA.E4 enhancer; 2) We performed an enhancer reporter assay in hTERT-HPNE in order to test the enhancer activity of hA.E4 in this duct cell line, showing that this sequence is an enhancer in this cell line (this result was added to Fig.4c). Although the enhancer reporter assays in zebrafish do not support the activity of hA.E4 in duct cells, the ATAC-seq profile and the *in vitro* reporter assay in hTERT-HPNE cells, together with the fact that the deletion of hA.E4 impairs *ARID1A* expression in hTERT-HPNE, clarify that indeed this enhancer has activity in human pancreatic duct cells.

Regarding the last part of this question, “*Additionally, linking back this figure to the title of the manuscript, the authors do not provide evidence associating this ARID1A enhancers to human disease. For example, are there PDAC risk polymorphisms in enhancers of ARID1A?*”, we did not find SNPs associated with PDAC, overlapping with the *ARID1A* enhancer described by us. Nevertheless, we would like to highlight that there is a very restricted number of PDAC associated SNPs, which likely reflects the limited number of cases and studies focusing on PDAC, when compared to other more broadly studied diseases such as Diabetes. This discrepancy can be appreciated in some of the latest publications regarding GWAS for Type 2 Diabetes (T2D) and PDAC, where about 403 SNPs associated to T2D are described from a total of 74,124 patients (PMID: 30297969), comparing to 38 PDAC-associated SNPs identified in 11,537 PDAC patients (PMID: 29422604). Apart from the scarcity of described PDAC-associated SNPs, it is important to highlight the rationale of choosing *ARID1A* as a case study. Loss-of-function of *ARID1A* is strongly associated with the development of PDAC, including validations in animal models. In the current work, we were able to show that there is a regulatory element of *ARID1A*, whose disruption leads to a downregulation of this gene in human cells. We agree with Reviewer1 that we did not demonstrate that this CRE is indeed associated with PDAC, accordingly, we do not state such a conclusion in the manuscript. However, we discuss the possible implications that our results may have, stating that “...We observed lower *ARID1A* expression upon deletion of hA.E4 compared to the control (Fig.4e-f and Supplementary Fig.5e), suggesting that the loss of this enhancer may interfere with the DNA-damage response, with possible implications in the increased risk for pancreatic cancer”. With this in mind, we have now improved the discussion of the revised version of the manuscript.

9) *In the final part of the study, the authors focused on an enhancer of PTF1A. In the original paper describing the human pancreas agenesis enhancer mutations (Weedon et al. 2014), the human enhancer is highly tissue and temporal specific, being exclusively active in pancreatic progenitor cells. Therefore, the claim of a functionally equivalent enhancer in adult zebrafish pancreas is inconsistent with the previous report. The presentation of the data is slightly misleading, as the authors compare H3K27ac from adult zebrafish pancreas with H3K4me1 from human pancreatic progenitors. To be a true functionally equivalent enhancer, the authors should*

demonstrate that the enhancer is active in the developing zebrafish pancreas, not in the adult. In humans, the enhancer that harbours agenesis-causing mutations is not active in adult human pancreatic islets, nor in whole pancreas. The results obtained with the enhancer deletion are certainly interesting, but do not demonstrate functional equivalency. PTF1A is expressed in the adult pancreas, in acinar tissue. Thus, it is expected that a number of adult pancreas enhancers drive PTF1A expression in acinar tissue in the adult. The fact that a deletion of an adult enhancer causes pancreas agenesis is also not indicative of functional equivalency per se, as PTF1A, as an important developmental gene, is expected to have multiple enhancers controlling its expression throughout development.

Response:

We thank Reviewer1 for such an insightful question. Regarding the first point raised by Reviewer1, “*To be a true functionally equivalent enhancer, the authors should demonstrate that the enhancer is active in the developing zebrafish pancreas, not in the adult.*”, we would like to highlight that the enhancer that we have found in zebrafish has activity in pancreatic progenitor cells. Indeed, we have compared the enhancer activity of the human *PTF1A* enhancer, named in our work as hP.E3, and the zebrafish enhancer (zP.E3), showing that both, human and zebrafish enhancers drive expression in pancreatic progenitor cells (Nkx6.1 positive; Fig.5b). Furthermore, we show that the zebrafish zP.E3 deletion induces a decrease in the pancreatic progenitor field, demonstrated for the first time *in vivo*, as represented in Fig.5f and quantified in Fig.5d, along with a decrease in *ptf1a* expression levels (Supplementary Fig.9b). This demonstrates that zP.E3 is indeed active and functional during pancreatic development. In this sense, we consider that there is an equivalency in function of zP.E3 and hP.E3 in pancreatic progenitor cells, since both are required for the proper expression of *ptf1a/PTF1A* during pancreatic development.

Despite the high degree of similarity between activity of hP.E3 and zP.E3 during early pancreatic development, we did find divergences of function in later stages, after pancreatic differentiation. Consistent information from mosaic transient reporter assays and stable transgenic reporter lines show that zP.E3 is able to drive expression in acinar and duct cells, while hP.E3 drives an extremely restricted expression in these two cell types, as can be appreciated by the stable transgenic lines documented in Supplementary Fig.10 alter, not showing an homogeneous

expression in these cell types. We have now improved the results section of the manuscript “A ptf1a enhancer explains pancreatic agenesis causal Variant *in vivo*”, pointing out these clear differences. We have also improved the discussion section.

Because there is a considerable and significant decrease of the number of pancreatic progenitor cells upon disruption of the zP.E3 enhancer, we raised the possibility that the pancreatic agenesis phenotype is most likely caused by the loss of the early developmental role of zP.E3. Nevertheless, we cannot exclude the possibility that this role of zP.E3 in differentiated cells may contribute to the pancreatic agenesis phenotype observed in zebrafish after pancreatic differentiation. We have now discussed these points in the revised version of the manuscript, highlighting the late divergent functions of the zP.E3 and hP.E3 enhancers.

10) *Whilst the work seems to have been motivated by the observation that regulatory sequence variants associate with pancreatic cancer and diabetes, the authors do not actually show much evidence of enrichment of type 2 diabetes common variants in shared zebrafish-human pancreatic enhancers. This is an expected outcome, as type 2 diabetes is predominantly underlined by defects in the endocrine compartment, which accounts for a minority of cells in the pancreas, but this limitation of the study should be better addressed in the discussion.*

Response:

We thank Reviewer1 for raising such an interesting topic. As Reviewer1 suggested, this work was motivated by the observation that common disease variants often fall within non-coding regions, many with enhancer marks, as stated in the text. We have interpreted this observation as an indication that enhancers constitute potential ‘hot-spots’ for previously undiscovered disease relevant loci and we set out to mine the zebrafish genome in search of pancreatic enhancers with the ultimate goal of relating the data back to the human genome. As Reviewer1 mentioned, we do not present evidence of enrichment of type 2 diabetes common variants in shared zebrafish-human pancreatic enhancers. This is because CREs show very low conservation between zebrafish and humans, severely hindering the identification of such equivalent sequences. This is an important point that we have approached in the manuscript with, we consider, enough depth. We look beyond sequence conservation, arguing that functional similarities can exist, even in absence of regions

with high sequence conservation. The main limitations are centered in the identification of such functional equivalent sequences in a genome wide manner. Following the suggestion of Reviewer 1, we have now better addressed this point in the discussion.

11) *A final point, the authors claim that this work has identified a region at ARID1A whose mutation may increase the risk of PDAC. This may be true, but it is not clear why this specific region (the common zebrafish-human enhancer) would be more important than other enhancers detected by ATAC-seq or H3K27ac ChIP-seq in human pancreas. Is the claim that shared enhancers are somehow more important to establish/maintain pancreatic transcriptional programmes?*

Response:

We thank Reviewer 1 for raising this point. We selected and tested 4 of the most prominent putative enhancers within the regulatory landscape of *arid1ab*, zA.E1, zA.E2, zA.E3 and zA.E4 (Fig.4a; please note that the nomenclature of these enhancers has changed and former zA.E3 is now referred to as zA.E4). Out of these sequences, zA.E1 and zA.E3 did not show pancreatic enhancer activity in *in vivo* reporter assays, and we observed that zA.E2 was a very weak pancreatic enhancer. On the other hand, zA.E4 displayed more robust enhancer activity in the pancreas. For this reason we focused on the zebrafish zA.E4 and its putative human correspondent enhancer (hA.E4). We have improved this description in the revised version of the manuscript. We would also like to clarify that we do not claim that this enhancer is more important than other *ARID1A* enhancers, since the loss of function of other enhancers was not tested. We simply used the above described pipeline to identify a potential candidate in human cells that could affect *ARID1A*. Finally, following deletion of hA.E4 we observed a reduction in *ARID1A* levels. Therefore, functionally, the remaining enhancers that may exist in the landscape of *ARID1A* were not sufficient to maintain *ARID1A* levels, which is the main conclusion that we reach from this experiment.

12) *The major strength of this work resides in the quality of the data and in this concept of a shared lexicon between pancreatic enhancers of humans and zebrafish. This is an important outcome, relevant to the wider field. The link with human disease genetics is nevertheless*

questionable with the current presentation of the data and should be re-addressed in more detail by the team.

Minor points:

12.1) *It is suggested that the authors revise the title of the manuscript, as all human pancreas profiling data seems to have been retrieved from ENCODE.*

Response:

Thank you for the suggestion. We have now modified the title to: “Multidimensional chromatin profiling of zebrafish pancreas to uncover and validate disease-related enhancers”.

12.2) *The flow of the manuscript could benefit from some restructuring. For example, it is unclear why the HiChIP experiment is presented in figure 1, when it becomes more relevant to the analyses shown in figure 2 to assign enhancers to target genes. Figure 1b could actually be improved by highlighting a specific enhancer element that interacts with gata6. The current 4C-seq representation does not make it clear which fragments show significant interactions with the promoter.*

Response:

We have now improved Fig.1b, according to the Reviewer1 suggestion, by highlighting a predicted enhancer within the landscape of *gata6*. Additionally, we decided to present the complete dataset in Fig.1b, including the HiChIP data, so that we can present an initial summary regarding the datasets we have generated and used in this manuscript. Finally, as suggested, we have excluded the 4C track since it is redundant together with the HiChIP heat map. The extensive revision of the manuscript included some restructuring, which helped to improve the flow of the manuscript.

12.3) *There is some inconsistency between instances where “pancreas enhancers” is used instead of “pancreatic enhancers” (the latter is preferred).*

Response:

We thank Reviewer1 for the suggestion. We are now consistent with the nomenclature, having changed all the terms to “pancreatic enhancers”.

12.4) *Figure 1c: the two yellow hues are very difficult to discern, but I assume that the 57.8% corresponds to intergenic? (as is mentioned in the text). I suggest changing one of the colours for clarity.*

Response:

We have now changed the colors of the graph in Fig.1c. We have also improved the selection of the color pallet to be easily read by color blind people. Yes, 57.8% corresponds to the percentage of intergenic sequences.

12.5) *Line 67: “which” should be “whose”.*

Response:

This point has been corrected.

12.6) *The authors present the examples of INSR and GATA6 as genes with important pancreatic functions. The manuscript would benefit by the presentation of a brief description of these genes, as Nature Comms is not a journal specialised on pancreas/diabetes biology. The authors should note that INSR is actually not a good example of a tissue-specific gene.*

Response:

In the revised version of the manuscript, we have made a brief description regarding the functions of *INSR* and *GATA6* in the pancreas.

12.7) *The authors selected the top 140 enriched motifs from different enhancer datasets. This threshold seems rather arbitrary. Could the authors provide an explanation?*

Response:

The threshold was arbitrary. 140 motifs correspond to 32% of the total dataset of motifs used for the motif enrichment discovery. We ensured that for this threshold, all the enriched motifs had $p < 0.01$. To better clarify this point, we improved the Material and Methods section.

12.8) *Figure S2k does not seem to be referenced in the manuscript. Furthermore, this figure shows many redundant motifs, the manuscript would be more informative if it presented non-redundant motifs.*

Response:

In the former Figure S2k (renamed (Supplementary Fig.4a) we presented the 3 top motifs, regardless of their similarities. To complement this table, we now present in Supplementary Table 3t-u the complete list of motifs and their respective p-values. Finally, as Reviewer1 pointed out, the figure was not referenced in the manuscript, we have now corrected the mistake in the revised text, within the Results section.

12.9) *The results from figures 4d, e are very interesting, but the downregulation is very mild (20% perhaps?). This may be due to the chosen readout. Was there a particular reason to quantify ARID1A protein levels by immunostaining, rather than ARID1A by qPCR on clonal populations?*

Response:

We thank Reviewer1 for pointing out this detail. Indeed, our first approach was to isolate clonal populations of hTERT-HPNE cells, however, we were not able to do so, either from enhancer deleted cells or negative control deleted cells. To circumvent this problem, we performed immunohistochemistry, since we are able to discriminate cells that incorporate the combination of sgRNAs. We believe that this is a good and valid readout, and possibly even better than a transcriptional output, regarding the impact of the genomic deletions in the activity of the gene, since several other post-transcriptional mechanisms may modulate the protein production and availability. Consequently, in this set of results, we show that there is an effective impact in ARID1A protein availability within the cell.

Reviewer2

Overview:

Bordeira-Carriço et al describe the discovery of putative zebrafish pancreas regulatory elements using H3K27ac ChIP-seq and compare this data to existing H3K27ac datasets obtained from whole developing zebrafish embryos. After classifying a set of putative pancreas enhancers of interest the authors link these elements to target genes using HiChIP. The authors go on to functionally test a subset of these enhancers in zebrafish. Overall the data and analyses the authors performed will be a wide interest as will the functionally validated pancreatic enhancers.

The topic is interesting, the zebrafish validation work appears to be of high quality and the authors clearly make progress in defining and validating zebrafish pancreas enhancers. However, the manuscript is in general challenging to read and the analyses do not always appear to be systematic. While there are many grammatical issues that add uncertainty to the manuscript, I am most concerned with the statements and conclusions made around the concept of functionally equivalent enhancers. I am convinced that the authors discovered examples of such enhancers (based on their functional validations) but how these results are used to make broader conclusions are not always justified by the analysis methods used. As written and presented now the paper would be stronger without trying to force the concept of functional equivalency (which I realize is not what the authors intend).

Three major concerns are that the paper does not acknowledge or use the appropriate cross species analysis methods; that the analysis and selection of enhancers for functional validation does not appear to be systematic; and the use of pancreas related disease regions/genes does not appear to be systematic. I have done my best to be specific/helpful in my comments. However, this has required me to quote parts of the manuscript which is a bit tedious and is mostly used to indicate where the comments pertain. While in many instances I could guess what the authors are saying or want to say, I could not always be sure. I have spent more time than average reviewing this paper and my comments are more than usual and intended to be constructive as possible.

Response:

We thank Reviewer 2 for the input and for the constructive revision of our manuscript. We have extensively addressed all the points raised by Reviewer2 below, improving the manuscript, particularly regarding the three main points raised by the reviewer. We would like to note that the selection of enhancers for functional validation was done specifically focusing on the perspective of human pancreatic diseases, a message that in our perspective is present throughout the text and that, after the revision process, we consider to have been strengthened. We would also like to highlight that overall our assays were performed at 3 different degrees of depth and resolution: 1) Genome wide, using open chromatin (ATAC-seq) and H3K27ac marks to predict zebrafish pancreatic enhancers, 2) validation of the predictions by *in vivo* reporter assays and 3) functional validation by mutagenesis of the pancreatic enhancers. The coverage that we have used for the *in vivo* assays in 2 and 3 cannot be done at larger scale or genome wide because they are extremely time consuming. So in our selection we prioritized genomic loci that could be relevant to human pancreatic diseases, that is one of the main goals of this work. Finally, we would like to add that to facilitate the identification of the changes performed in the revised version of the manuscript addressing each specific point raised by Reviewer2, we have annotated the respective point in a “comment” in the manuscript version that contains “track changes”. E.g. for the changes performed to address point 2, we have added the following comment “Reviewer2, point 2”.

Specific points:

1) *When the authors start to describe their functional validations, the PsE and DevE concept never comes back. 10 regions with strong H3K27ac and 7 regions with low H3K27ac activity were tested. Are they all DevEs? Were any PsE elements tested to confirm they are indeed only active in adult pancreas?*

Response:

We thank Reviewer2 for raising this point. We would like to clarify that all sequences obtained, PsE and DevE, show H3K27ac signal in the adult pancreas, therefore, all these sequences have the potential to be adult pancreatic enhancers. The main goal to classify these adult pancreatic enhancers as PsE and DevE was to primarily understand the potential tissue specificity of these

putative enhancers, as stated in the manuscript (“To identify a subset of pancreatic enhancers with higher tissue-specificity, we compared the H3K27ac data from adult zebrafish pancreas to whole embryos at four developmental stages...”). As discussed above, regarding the question raised by Reviewer 1, point 3, we decided to use H3K27ac ChIP-seq data-sets performed in whole embryos at different developmental stages (Dome, 80% Epiboly, 24hpf and 48hpf) because whole embryos comprise many different types of differentiated and differentiating cells. Therefore, these datasets allow us to assess the potential activity of enhancers in already differentiated tissues as well as non-differentiated cells using a single model, without having to specifically dissect each organ from an adult zebrafish. Apart from this, the availability of datasets of H3K27ac in zebrafish adult tissues is limited, being restricted to brain, muscle, heart, intestine and testis (PMID: 27965291; PMID: 28245924; PMID: 28245924), which it would not allow us to take too many generalizations from such comparison. For this reason, we restricted the use of these groups mostly to the part of the manuscript that focus in the tissue specificity of the predicted enhancers.

Regarding the *in vivo* validation using enhancer assays, the main question we wanted to address was, if the H3K27ac dataset allowed us to predict active enhancers. We have now clarified this in the revised version of the manuscript. To address this question, we focused on the levels of H3K27ac signal, dividing testing sequences as having high and low H3K27ac signal. Within this dataset we find the *ptf1aE3* enhancer, that belongs to the PsE group. We have generated a stable transgenic reporter line for this enhancer. We have improved the revised version of the manuscript by describing the expression pattern of this enhancer in late whole larvae (17 days post fertilization), in young juveniles (2 months post fertilization) and in adults (2 years post fertilization; (Supplementary Fig. 10c). We can appreciate that the expression driven by *ptf1aE3* is highly specific, being restricted to pancreatic cells, observed in late whole larvae. In juveniles and adults, despite animals present opacity and some autofluorescence, GFP can be detected in the pancreas, co-localizing with the exocrine reporter *ela:mCherry* and surrounding the endocrine reporter *sst:mCherry*. Finally, we would like to clarify that, although this enhancer has pancreatic developmental roles, being active in the pancreatic progenitor domain (Fig. 5b), its activity is not detected in the H3K27ac dataset at 48hpf because it is so specific and restricted to the pancreatic progenitor domain. Overall, this is a clear case showing that our approach for the identification of tissue specific enhancers is effective. We have now improved the discussion of the manuscript adding this point.

2) What is the logic for choosing the zebrafish and human *arid1ab*/*ARID1A* regulatory elements and the *ptf1a* enhancer for functional testing? Do they fall into different categories? For example, are they both PsEs? I would assume the zP.E3 belongs to DevEs as it shows activity in larval fish. What about the other distal enhancers at this locus? For example in Fig. 4 there looks like the 4C data indicates an upstream enhancer that corresponds to an upstream enhancer in the human genome. I think for these two loci it would have been relevant to acknowledge and screen all putative zebrafish and human enhancers in zebrafish to reveal which one(s) drive similar expression patterns. This would allow the authors to more definitively make their case for functional equivalency.

Response:

We are very grateful for this comment as it allows us to improve the description of the rationale behind the selection of regulatory elements for functional validation. In this work, our selection of enhancers for *in vivo* validation was prioritized to explore the impact of cis-regulatory mutations in the development of human pancreatic diseases. We have clarified this point within the Results section, in the revised version of the manuscript. This was the main reason for the selection of *arid1ab*/*ARID1A* and *ptf1a*/*PTF1A* loci and respective enhancers, not because these fall in a specific DevE or PsE category. Following this reasoning, we focused on the landscapes of 2 genes relevant for the development of pancreatic diseases; *arid1ab*/*ARID1A* is a tumor suppressor gene that has been demonstrated to trigger pancreatic cancer (PMID: 29486633). However, no regulatory mutations associated with pancreatic cancer have been described, resulting in a good opportunity to further explore potential pancreatic cancer related enhancers. In turn, the *ptf1a*/*PTF1A* locus is one of the very few loci that have been described in humans as containing an enhancer that when mutated results in a severe and easily identifiable phenotype - pancreatic agenesis - which should be easily recapitulated if an equivalent mutation in an equivalent functional enhancer is induced in zebrafish. Indeed, we have demonstrated the induction of such phenotype with a deletion in the core of zP.E3.

As described previously, in response to Reviewer2's point1, zP.E3 is not within the DevE group. In the transgenic reporter line, zP.E3 becomes active between 24 and 48hpf, however it is

not detected in the 24 or 48hpf whole embryo H3K27ac dataset, because its activity is specifically restricted to the pancreatic progenitor domain, which constitutes an extremely small cell population in the whole embryo. We would also like to highlight that, as was discussed previously in response to Reviewer1's point 6 and 10, the main objective of describing DevEs and PsEs is to discriminate potential pancreas specific enhancers (PsEs), contrasting with enhancers detected in whole developing embryos (DevEs). The DevE group itself is not related with the identification of pancreatic developmental enhancers. We have now changed the description of DevE in the manuscript to "active in the differentiated adult pancreas and also broadly active in developing embryos during embryonic development (DevE)" in the "Zebrafish putative pancreatic enhancers share developmental roles" section of the Results, and we clarified the objective of these different groups in the revised version of the manuscript, as described in response to Reviewer2's point1.

Regarding this question "*What about the other distal enhancers at this locus [ptf1a locus]?*", as described in the manuscript, the zP.E1 and zP.E2 enhancers, predicted by our dataset, have been previously identified and validated by Pashos and colleagues (PMID: 23876428). Conversely, zP.E3 represented a novel enhancer that had not been previously described, which is why we focused our attention on this enhancer. Moreover, the combination of H3K27ac, ATAC-seq and HiChIP signals highlighted zP.E3 as the best candidate for a *ptf1a* distal enhancer. Apart from zP.E3, we have not found any other putative *ptf1a* distal enhancer in this locus (Fig.5a).

Regarding the following question "*For example in Fig 4 there looks like the 4C data indicates an upstream enhancer that corresponds to an upstream enhancer in human.*", we had previously tested this sequence for enhancer activity. This sequence is annotated in the original and revised manuscripts as E10 in Supplementary fig 4a and in Fig.2c. However, by mistake, we had not annotated this sequence in the landscape of *arid1ab*. We apologize for this oversight and we have corrected it in the revised version of the manuscript, including Fig.4a. This upstream sequence does not function as a pancreatic enhancer (Supplementary Table 4a, only 5,56% F0 larvae with GFP expression in the pancreas, p-value = 0.7955) and is now named zA.E1. Consequently, the previous zA.E3 is now named zA.E4 and the remaining sequences were also renamed accordingly (Fig.4a). Because, as mentioned in response to Reviewer1's point 11, out of the 4 putative *arid1ab* enhancers only zA.E4 displayed strong and reproducible enhancer activity in the pancreas (Supplementary Table 4a) we focused our efforts in uncovering its putative human equivalent.

We would also like to highlight that our strategy was to use the zebrafish genome to identify a pancreatic *arid1ab* enhancer that could point us to an equivalent functional sequence in humans. Using this rationale, we were able to identify a functional *ARID1A* pancreatic enhancer in humans, which we tested for enhancer activity *in vivo* and, in the revised version of the manuscript, *in vitro* in human cells (Fig.4c).

3) *The authors have 7 sets of enhancers: PsE, DevE, PsE+DevE, C1, C2, C3, C4. The authors examined the average expression of genes interacting with enhancers belonging to each of the categories (Fig2a and FigS1d). However, when presenting the ratio between the average expression of genes interacting with a given category of enhancers and the average expression of all genes throughout zebrafish development (Fig2b), it did not seem to be presented for all categories (it seems that DevE and PsE+DevE are missing). Similarly, for FigS1e, DevE is missing.*

Response:

We thank Reviewer2 for raising this point, helping us to clarify the rationale of this experiment. Indeed, in Fig.2b we represent only the PsE and C1-4 datasets. The main objective of Fig.2b is to summarize the main aspects of our data, for the sake of readability and interpretation. These datasets and the PsE+DevE dataset can be found in Supplementary Fig2e (formerly Fig.S1e) (with an increased degree of resolution (e.g. more developmental time-points). Regarding the choice of the groups to be represented in Fig.2b, we consider that PsE and C1-4 datasets are the most relevant, since they allow to clearly conclude that: 1) throughout zebrafish development (prior to the appearance of the differentiated pancreas), the average expression levels of the genes associated to PsE sequences is overall lower than any of the C1-4 subgroups that compose the DevEs group; and 2) the average expression levels of the genes associated to C1-4 subgroups overall recapitulate the activity dynamics of these enhancers during development, observed in Fig.1d. For instance C2, that shows highest average transcriptional levels of associated genes in early developmental stages (BDO:blastula to G75: 75%epiboly), also shows highest levels of H3K27ac at the Dome and 80% epiboly developmental time-points (Fig.1d), therefore showing a correlation between H3k27ac signal and transcription. We highlight this example in the revised version of the manuscript for clarification. Regarding the graphs plotted in Supplementary Fig2d, we have now included all the groups (PsE, DevE, PsE+DevE, C1, C2, C3, C4).

4) *The liftover method and alignments used are not the state of the art for zebrafish comparative genomics. Work done by many groups have established a comprehensive set conserved elements using methods including conserved synteny, ancestral reconstruction and intermediary species. These resources seem not to be used or acknowledged in the paper. While reference 19 (PMID: 23814184) is very relevant it seems to be used out of context in the introduction (this method does not simply take the nearest gene). Also the same reference (19; PMID: 23814184) is also used as a reference for the enrichment tool GREAT later in the paper which is clearly an error. I did not have time to cross-check all the references but this is a major concern in addition to my main point that the manuscript does not seem to acknowledge key literature in the experimental design and data interpretation. For example, a quick look at the *zptf1aE3* (1.8k) region in the context of existing zebrafish conserved non-coding elements resources, show it encompasses a zebrafish conserved non-coding element which has an orthologous human region 300 kb away (hg19: chr10:23579080-23579212) from the *hPTF1AE3*. This conserved element was not mentioned in the paper and contradicts the authors statement that there was not enough sequence identity to link *zptf1aE3* to the human genome. While this region does not appear to have the enhancer features than *hPTF1AE3*, it is not mentioned by the authors and its likely there are other examples of this in the dataset. While I would expect very few mammal-zebrafish conserved elements regardless of the alignments used, a more thorough analysis and more representative acknowledgement/citation of work in this area would be needed if the aim of this paper is to make statements about lack of sequence conservation of regulatory elements and functional equivalency.*

Response:

We thank the multiple questions that Reviewer2 points, giving us the opportunity to clarify them as following:

4.1) *“The liftover method and alignments used are not the state of the art for zebrafish comparative genomics. Work done by many groups have established a comprehensive set conserved elements using methods including conserved synteny, ancestral reconstruction and intermediary species. These resources seem not to be used or acknowledged in the paper.”*

In our initial approach to the alignment of zebrafish enhancer sequences with the human genome, we explored several datasets of identified zebrafish/human alignments. One of such datasets was the one obtained from the Bejerano lab (PMID: 23814184). However, we observed

that the alignments described in this dataset were relatively similar to the ones we obtained using liftOver. Using the liftOver method we observed that approximately 12.5% of predicted pancreatic enhancer sequences aligned to the human genome, while with the alternative method approximately 10.5% of predicted enhancers aligned with human sequences. This comparison can be observed in the graph below. This graph is similar to the one in Fig.3a, and shows the percentage of aligned zebrafish pancreatic enhancers to the human genome using both approaches, the liftOver (black bars) and the Bejerano’s dataset (grey bars). Since the output was not very different between approaches (i.e. regardless of the method used only a minority of sequences aligns with human genome, despite the liftOver method being less restricted), and because we were already using liftOver to convert genomic coordinates from one version of the genome to another, we decided to systematically use the liftOver method in this work. In this context, and following Reviewer2’s suggestion, we have now included the following reference in the revised version of the manuscript (PMID: 23814184).

Figure[reviewer2_point4]. Percentage of predicted zebrafish pancreatic enhancer sequences that can be aligned to the human genome through 2 different approaches.

4.2) “While reference 19 is very relevant it seems to be used out of context in the introduction (this method does not simply take the nearest gene). Also the same reference (19) is also used as

a reference for the enrichment tool GREAT later in the paper which is clearly an error. I did not have time to cross-check all the references but this is a major concern in addition to my main point that the manuscript does not seem to acknowledge key literature in the experimental design and data interpretation”.

We would like to clarify that we do not cite reference 19 (PMID: 23814184) in the introduction of the initial version of the manuscript. GREAT was referenced for the first time in the original manuscript in the Results section “Functional similarities between human and zebrafish pancreatic enhancers”. We also understand the perplexity of Reviewer2 regarding the citation of reference 19 (PMID: 23814184) in respect of the GREAT method. However, this was not an error. We have followed the authors’ suggestions regarding GREAT citations, which state: “If you use GREAT in the context of zebrafish, please cite: Michael Hiller, Saatvik Agarwal, Jim H. Notwell, Ravi Parikh, Harendra Guturu, Aaron M. Wenger, Gill Bejerano. "Computational methods to detect conserved non-genic elements in phylogenetically isolated genomes: application to zebrafish". Nucleic Acids Res., 2013. PMID 23814184 Why Cite GREAT”, according to the GREAT web platform <https://great-help.atlassian.net/wiki/spaces/GREAT/pages/655454/Citation>. Although we respect the GREAT authors’ request regarding the citation of their work, we agree with Reviewer2 that a more explicit citation of the GREAT software should be added. In this regard, we have now added the following reference to the revised version of the manuscript (PMID: 20436461). In addition, we have done a detailed reference cross-check in the revised version of the manuscript to ensure that all relevant literature is cited.

4.3) *“For example, a quick look at the zP.E3 (1.8k) region in the context of existing zebrafish conserved non-coding elements resources, show it encompasses a zebrafish conserved non-coding element which has an orthologous human region 300 kb away (hg19: chr10:23579080-23579212) from the hP.E3. This conserved element was not mentioned in the paper and contradicts the authors statement that there was not enough sequence identity to link zP.E3 to the human genome. While this region does not appear to have the enhancer features than hP.E3, it is not mentioned by the authors and its likely there are other examples of this in the dataset. While I would expect very few mammal-zebrafish conserved elements regardless of the alignments used, a more thorough analysis and more representative acknowledgement/citation of work in this area would*

be needed if the aim of this paper is to make statements about lack of sequence conservation of regulatory elements and functional equivalency.”

We thank Reviewer2 for raising this interesting question that we would like to further develop in depth, since it is directly related with the main focus of this work. Indeed, as Reviewer2 pointed out, the zP.E3 region that was tested for enhancer activity (1828bp) has a partial alignment (a little over 100bp) with the human genome in its 3' extremity. However, the zebrafish aligned region does not overlap with an open chromatin profile in zebrafish adult pancreatic cells, as observed by ATAC signal (Supplementary Fig.6a-b). We consider the aforementioned stretch of open chromatin (600-700bp) to be the core of the enhancer, and this is supported by the phenotypes observed for Deletion1 and Deletion2 of the enhancer (added to the revised version of the manuscript) (Supplementary Fig.9), neither of which overlaps with the region that aligns with humans. Deletion1 abrogates the core of the enhancer and results in a very dramatic pancreatic agenesis-like phenotypes, while Deletion2 only affects a very short region of the core (29 bp) and does not result in any observable pancreatic phenotype (Supplementary Fig.9). These results strongly suggest that the functional part of the enhancer is within its core, that does not coincide with the human aligned sequence.

Driven by the interesting question raised by Reviewer2, we further explored the alignment of this sequence with other mammals to determine if the human alignment could be extended further. Although an alignment with the human genome is only detected for a little over 100bp of the 1828bp of the sequence tested for enhancer activity, a more internal region of the enhancer, which partially overlaps with its core, can also be aligned with the mouse genome (approximately 340bp). And, when we traced the mouse sequence to the human genome (Supplementary Fig.6), we observed that, indeed, the zebrafish enhancer sequence can have a higher extended alignment with mammals. However, we would like to highlight that the alignment of the human and zebrafish sequences is very low, as observed in (Supplementary Fig.6). More importantly, the human sequence does not present marks of enhancer activity in progenitor or adult pancreatic cells, as mentioned by Reviewer2 (Supplementary Fig.6). This is an important observation, and we previously described this phenomenon in Fig.3b, where we observe that the vast majority of the zebrafish putative pancreatic enhancers that can be traced into the human genome by alignment, do not share marks of enhancer activity in human whole pancreatic tissue. We have now made changes in the revised version of the manuscript, as suggested by Reviewer2, to acknowledge and

clarify the alignment of zP.E3 with the human and mouse genomes, in particular, we added a new supplementary figure (Supplementary Fig.6) to illustrate these alignments and we improved the discussion to contextualize this observation. Also, as suggested by Reviewer2, in the revised discussion, we now acknowledge and cite several works that focus on the characterization of enhancers, including cases where: 1) conserved sequences can have divergent enhancer functions and 2) sequences that share no significant sequence identity can share functional equivalencies. We consider that, indeed, this observation illustrates very well the importance of the current work, showing that sequence conservation is a poor predictor of the specific function of cis-regulatory elements.

5) It is not clear how the authors defined syntenic blocks and their corresponding enhancers in methods section. For example, how can one rule out that the zA.E3 instead of zA.E2 is the functional equivalent of hA.E3?

Response:

Human/zebrafish syntenic blocks were defined by the existence of two aligned regions between both species that kept their position relative to each other. Pre-existing alignments available in the UCSC genome browser were used and we then searched for putative enhancer sequences within these blocks in both species. We have improved the methods section of the revised version of the manuscript with this description. After identifying putative enhancers within these blocks, we performed enhancer reporter assays to determine if the tested sequences have similar enhancer activity. Regarding the zA.E4/hA.E4 (formerly zA.E3/hA.E3) correspondence, we have excluded zA.E3 (formerly zA.E2) because this sequence did not display pancreatic enhancer activity in the enhancer reporter assays (sequence E10 in Fig.2c).

6) To confirm the functional equivalency, the authors should consider testing if the zA.E3 can rescue the hA.E3 deletion cell line and restore the expression level of ARIDIA.

Response:

We thank Reviewer2 for such an interesting suggestion. However, we would like to note that said experiment is technically very demanding. As previously expressed in the response to Reviewer1 (point 12.9), the deletion of the hA.E4 enhancer (previously zA.E3) was performed in populations of cells because we were not able to grow hTERT-HPNE cells in isolation, therefore

we were not able to generate clones for these deletions. The generation of clones would be a requirement for this elegant but very demanding experiment. This technical difficulty would make this experiment much less feasible.

7) To show that *zPtf1aE3* is the functional equivalent enhancer of *hPtf1aE3*, the author should show that the expression level of *PTF1A* gene is affected upon loss of the enhancer.

Response:

We thank Reviewer2 for the suggestion and we fully agree with the need to assess how the expression levels of *ptf1a* are affected by loss-of-function of the distal enhancer. Our first approach to determine the impact of zP.E3 deletion in the transcription of *ptf1a* was to perform an *in situ* hybridization with a *ptf1a* probe. We had synthesized a previously described probe against *ptf1a* (PMID: 15183727) (5'-GCGTCAAGAGGAAAAGATCG and 5'-CGACACAAACATGATTGCC; 1028bp product). However, we were not successful in detecting *ptf1a* mRNA. Control probes used in parallel worked properly, suggesting that the *in situ* protocol, which is standard in our laboratory, worked as expected. To bypass this problem, we instead performed qPCR in cells extracted from the ventral region of 24hpf embryos harboring a deletion of the enhancer. In brief, we disrupted the yolk of embryos using Ginzburg Fish Ringer's solution by gently pipetting up and down. We centrifuged the solution very briefly to recover the supernatant that contains ventral endoderm cells (including pancreatic progenitor cells) and discarded the embryo's body. We had previously tested this approach using the Tg(zP.E3:GFP) reporter line, which expresses GFP in pancreatic progenitors, and verified that the method successfully separated pancreatic progenitor cells from the remaining embryo using a very fast and simple process. We extracted total RNA from these cells and performed qPCR for *ptf1a*, housekeeping genes (*tbp* and *b2m*) and for *pdx1*, a specific marker of pancreas progenitor cells. Expression of *pdx1* precedes the expression of *ptf1a* in these cells and, therefore, is likely independent of *ptf1a* expression. Comparing zP.E3 deletion (Deletion 1) with siblings, we have observed a decrease in the levels of *pdx1*, as expected and consistent with our previously observation of a decreased number of pancreatic progenitor cells in zP.E3 deletion (Deletion 1), and we also observed a decreased expression of *ptf1a*. To take into account the loss of progenitor cells, we have normalized the expression of *ptf1a* by *pdx1*, still observing a significant decrease in

the transcription of *ptf1a*. These results are now presented in Supplementary Fig.9b of the revised version of the manuscript.

8) *Abstract states “Most disease-associated alleles overlap with non-coding cis-regulatory elements of DNA, suggesting that alterations in regulatory sequences contribute to pancreatic diseases.” Is this really true? I would argue that most disease associated alleles affect protein structure/function. I would agree that most GWAS hits fall in non-coding regions and that understanding how this genetic variation impacts gene expression is essential. I am sure the authors know this, and I highlight it only as an example of where a lack of precise language impacts the scientific message.*

Response:

We thank Reviewer2 for helping us to improve the manuscript writing. We have now improved the sentence in the revised version of the manuscript to make our message clearer. The respective sentence has been modified to: “Most alleles uncovered by genome wide association studies with associations to pancreatic diseases such as diabetes, overlap with non-coding sequences of DNA, many containing epigenetic marks of cis-regulatory activity in pancreatic cells. This suggests that alterations in regulatory sequences can contribute to pancreatic diseases.”

9) *There are also many instances in the manuscript where two different ideas are presented within a single sentence. Also there are many examples where two adjacent sentences do not have a clear conceptual link. I am not a professional writer, but as a scientist interested in the authors work, I found this to be an issue when trying to follow the logic of the paper. Examples of this can first be seen in the abstract: “Most disease-associated alleles overlap with non-coding cis-regulatory elements of DNA, suggesting that alterations in regulatory sequences contribute to pancreatic diseases. However, the interspecies identification of equivalent cis-regulatory elements required for in vivo testing face fundamental challenges, including lack of sequence conservation.” [a) how does the first part of the sentences connect to pancreas and b) what is the connection between disease associated alleles and interspecies studies?]*

Response:

We thank Reviewer2 for bringing this to our attention. To clarify the highlighted sentence, we meant that several GWAS studies described alleles that overlap with non-coding sequences, many of these sequences containing epigenetic marks of cis-regulatory activity, suggesting that

alterations in regulatory sequences can contribute to pancreatic disease. This hypothesis, namely that the alterations in regulatory sequences can contribute to pancreatic disease, should be validated and demonstrated. *In vitro* assays can be performed; however, *in vivo* modeling is preferred since regulatory sequences can affect multiple cell types and diseases tend to have more complex phenotypes than what can be explained *in vitro* by single cell types. On the other hand, the loss-of-function assays of cis-regulatory elements in *in vivo* models faces fundamental challenges, including lack of sequence conservation. With the approach we used in this work, combining analysis of ATAC-seq, ChIP-seq, 4C-seq and HiChIP-seq in zebrafish and human pancreatic cells, we were able to identify interspecies functionally equivalent cis-regulatory elements, regardless of sequence conservation. To clarify this issue, we have made the following adjustment to the manuscript: the sentence pointed out by the reviewer now reads “Most alleles uncovered by genome wide association studies and associated to pancreatic diseases such as diabetes, overlap with non-coding sequences of DNA, many containing epigenetic marks of cis-regulatory activity in pancreatic cells. This suggests that alterations in regulatory sequences can contribute to pancreatic diseases. Animal models can give a major contribution to explore the role of these non-coding alterations in disease. However, the interspecies identification of equivalent cis-regulatory elements required for *in vivo* testing face fundamental challenges, including lack of sequence conservation.”

10) *A dramatic societal burden of pancreatic diseases is mentioned however this statement would better be supported by citations or specific diseases and numbers.*

Response:

We thank Reviewer2 for the comment and we agree that the statement of this sentence should be better supported by citations. The study of *Mattiuzzi et al. (2020)* (PMID: 32542083) described that the incidence, prevalence, and mortality of pancreatic cancer increased by 55%, 63% and 53%, respectively, during the last 25 years and future projections proposed that its burden may double during the next 40 years. In 2019, a study from GBD 2017 Pancreatic Cancer Collaborators (PMID: 31648972) suggested that the development of screening programs for early detection and more effective treatment strategies for pancreatic cancer are needed to control the increasing numbers of deaths for pancreatic cancer. Additionally, a recent study (PMID: 33058868) showed an increasing incidence and mortality trends in pancreatic cancer, especially among women and

populations with 50 years or older, but also among younger individuals. Regarding diabetes, it is a long-term condition with major impact on the quality of life of individuals, with increased risks for chronic complications and with high costs for society (PMID: 31518657, PMID: 28847479). It is among the top 10 causes of death in adults and its prevalence is estimated to be 9,3% in 2019, just under half a billion people worldwide (PMID: 31518657), increasing to 19.3% of people aged 65-99 years (PMID: 32068097), and it is increasing dramatically also under 40 years old (PMID: 28847479). It is projected to increase by 25% in 2030 and by 51% in 2045 (PMID: 31518657). We added these references and made adjustments to the manuscript accordingly with Reviewer2's suggestion.

11) Intro: *“However, evidences that support this hypothesis are very limited, and for the most of them, they are inferred by in vitro assays.” What hypothesis is being referred to here?*

Response:

We appreciate the comment made by Reviewer2 and we agree that the original sentence should be better explained in the main text. Many Genome-Wide Association Studies (GWAS) have identified several non-coding alleles associated with pancreatic diseases that overlap with CREs signatures, suggesting that the disruption of CREs may be one of the genetic causes for the development of pancreatic diseases, such as the case of diabetes and pancreatic cancer. This hypothesis, that CREs-alterations can cause pancreatic diseases, has been explored in several works demonstrating an enrichment of diabetes variants at adult human islet enhancers (PMID: 24127591, PMID: 24413736, PMID: 30181159, PMID: 31064983, PMID: 31253982) and experimental reports have shown a regulatory function for specific islet enhancer variants (PMID: 20118932, PMID: 28684635, PMID: 29625024, PMID: 30181159). We have adjusted the manuscript to make this point more explicit.

12) *“Yet, the identification of interspecies functionally equivalent CREs face unsolved fundamental challenges, such as low conservation of interspecies non-coding sequences (10) and, for the minority of CREs that are conserved, their fast-evolving functionality (11)”. This is an important statement and deserves more elaboration/clarification and citations. I think this concept of shadow enhancers, which is not acknowledged by the authors, directly relates to enhancer evolution and the regulatory elements one would naturally expect to find around transcription*

factors that have been shown to play a key role in pancreas development in humans and fish. Citing work that has compared regulatory elements between human and fish would also be warranted (e.g early work in zebrafish with RET enhancers is but one clear example of this concept PMID: 16556802 that was not acknowledged). One could almost get the impression that the functional equivalency was a new concept being proposed. I certainly believe that more work is needed on this topic and the authors have provided additional examples.

Response:

We thank Reviewer2 for bringing this to our attention and we agree that our statement in this particular paragraph deserves more clarification and citations.

It is widely accepted that evolutionary forces drive genome architecture, since sequences that remain highly conserved between divergent organisms are likely to be functional. In several studies sequence conservation of non-coding sequences has been successfully used to find enhancers, many with interspecies orthologous identities (PMID:19212405, PMID:2117797). However, it has also been demonstrated that this type of approach is insufficient to identify all enhancers within a genome and between species (PMID: 22143240). In a study by Rubin and colleagues (PMID: 17803355), several ultraconserved sequence enhancers were deleted in mice with no significant phenotypic outcome, suggesting that extreme sequence constraint does not necessarily reflect crucial functions. Additionally, McCallion and colleagues (PMID: 18071029) observed that enhancer regions with no apparent sequence conservation display regulatory function, demonstrating that the non-coding functional component of vertebrate genomes may far exceed estimates predicated by evolutionary constraints. As pointed out by Reviewer2, the RET locus is also an excellent example of such observation (PMID: 16556802). In this context, we have now improved the introduction of the revised version of the manuscript, including references to most of these works.

Shadow enhancers, also known as redundant enhancers, are found in a wide range of organisms and have an important role in developmental patterning. Importantly, shadow enhancers are redundant only in that they have overlapping activity patterns and are not necessarily functionally identical (PMID: 27863239). This concept could explain the non-complete penetrance of pancreatic agenesis we observed when deleting the distal *ptf1a* enhancer in zebrafish. We have now improved the revised manuscript including the shadow enhancer concept in the discussion chapter.

13) *It seems like the authors do not really integrate their data generated but rather generate a series of important comparisons using specific datasets. This is of course fine, however I wondered why the ATAC-seq data was collected but did not seem to be used in the downstream analysis. Presumably the ATAC-seq data would give more refined putative enhancer regions for functional testing.*

Response:

We thank Reviewer2 for giving us the opportunity to clarify this point. In this work we used the ATAC-seq data to select the putative enhancers regions to test *in vivo* as well as to design the amplification primers and the small guide RNAs (sgRNAs) used in the CRISPR-cas9 assays. As this information had not been made clear in the original Material and Methods section, we clarified it in the revised manuscript by adding more detailed information [Material and Methods section, I) Experimental procedures: Generation of plasmids for enhancer assays, Cas9 target design; sgRNA synthesis and mutant generation; CRISPR-Cas9 in human cell lines]. The ATAC-seq data was also used to check if the *arid1ab* and *ptfla* promoters genes are active in the pancreas, as demonstrated in Fig. 4 and 5 Finally, to refine our transcription factor analysis, we filtered our putative pancreatic enhancers identified by H3K27ac ChIP-seq data with the ATAC-seq signal, as described in the Results section and in the Material and Methods section [Material and Methods, II) Processing and Bioinformatic analysis: Transcription factor binding motifs enrichment section].

14) *“Half of the Zebrafish Pancreas Enhancers Share Developmental Roles” – this should say putative enhancers.*

Response:

We acknowledge this comment by Reviewer2 and we altered the manuscript accordingly: “Zebrafish Putative Pancreatic Enhancers Share Developmental Roles”.

15) *“Interestingly, DevE presented 4 clusters (C1-4) with different activity dynamics during development (Fig.1d; Fig.S1b; Table S1).” The authors should clarify what is the interesting aspect of this.*

Response:

We appreciate the comment by Reviewer2 and agree that the original sentence should be better explained in the main text.

We compared the data from adult zebrafish pancreas to whole embryos datasets (done to 48hpf), in order to identify a subset of pancreatic enhancers with higher tissue-specificity. We found that 7115 (48.2%) were active only in the differentiated adult pancreas (PsE; Fig.1d; Supplementary Table 1d) while the remaining 7638 (51.8%) were also active during embryonic development (DevE). These results suggest that, apart from their activity in the adult pancreas, these enhancers function in other cell types. C1 and C4 show profiles invariably active in all developmental stages, compatible with a putative ubiquitous activity, while C2 and C3 show different states of activity during development, that may correspond to a dynamic state of repression (C2) and activation (C3) or, alternatively, differences in the abundance of cells where these enhancers are active during development. We have now better explained this in the revised version of the manuscript. We consider this observation to be very interesting, but to be fair, we have now deleted the “interesting” appreciation from the text, letting that appreciation be freely formulated, or not, by the reader.

16) *“We found that PsE-associated genes have a higher average expression in a variety of pancreatic cell types when comparing to all transcribed genes, contrasting with transcription in the muscle (Fig.2a, Table S3).” It would be important to know how many such genes were detected using the HiChIP versus the previous analysis that used a defined list of pancreas expressed genes. (It would be interesting to know how using HiChIP alters the list of enhancers near pancreas genes of interest compared to the proximal gene analysis).*

Response:

We appreciate this comment of the Reviewer2 and regarding this point, we agree that it would be interesting to know how using HiChIP alters the list of genes associated with enhancers, compared to the proximal gene analysis. In this context we have performed the suggested analysis and made them available for Reviewer2’s appreciation.

Briefly, the initial analysis was done using GREAT to assign genomic regions (in this case PsE-enhancers) with the nearest genes, using the GREAT basal plus extension rule. This proximity-based association resulted in 4809 genes associated with PsE. Using HiChIP for the active promoter mark H3K4me3 in adult pancreas, we were able to associate 6174 genes to PsE. We have

displayed this overlap between the two datasets in the venn diagram below (A:VENNY), showing that 1726 genes interact with PsE based on the 2 methods (HiChIP and GREAT). We then looked at the average expression of the genes associated by each method and their overlap; that is all GREAT-associated genes (4809 genes), all HiChIP-associated genes (6174 genes) and the subset of GREAT+HiChIP-associated genes (1726). Pancreatic tissues show a higher average expression of genes associated to PsE by HiChIP (A: 6174 genes in total) compared to average of all transcribed genes, while the average expression in the muscle is lower than the average expression of all genes strongly hinting at the specific role of HiChIP-PsE-associated genes in pancreas (Fig.2a). We observed the same profile in pancreatic tissues for GREAT-associated genes (A: 4809 genes in total) and we presented these profiles below (B: GREAT-expression). Additionally, we also show the average expression of genes associated with PsE by both methods (C: GREAT&HiChIP-expression, 1726 genes in total).

Despite the higher average expression of GREAT-PsE-associated genes in pancreatic tissues, the muscle control reveals that such associations lose specificity for the pancreas, as the expression in muscle is higher compared to the average of all genes (B: GREAT-expression). In contrast, the average expression of HiChIP-PsE-associated genes (Fig.2a) shows a higher specificity for pancreas, as mentioned above, suggesting their higher usage in pancreas. As for the average expression of genes associated by GREAT and HiChIP (C, below), we observe an even higher expression in pancreatic cell types while maintaining a slightly lower expression in muscle, although not as low as that for all HiChIP-associated genes shown in Fig.2a, but the number of genes is very reduced (1726 genes in total).

Despite the higher average expression in pancreatic tissues for GREAT associations (B, below), GREAT and HiChIP methods for prediction of enhancer-gene associations are quite different. GREAT performs associations up to 100Mb distances and for input coordinates overlapping high gene density regions (up to 6kb distant) it connects all genes in that region to each overlapping input coordinate (putative active enhancer). However, for distances above 6kb, the rule imposes a maximum of 2 genes interacting with an input coordinate, which can deflate the number of associations compared to HiChIP, in which no such limit exists and more interacting genes per enhancer may be reported. In addition, as GREAT is not based on experimental evidence for interactions, it may lack sensitivity to pinpoint condition, time-point or cell-type specific interactions. This contrasts with HiChIP, whose resolution does frequently not go beyond 5kb,

resulting in all genes within a 5kb chunk being associated with all enhancers in the interacting 5kb chunk, no matter what distance they are apart. Nevertheless, HiChIP is based on experimental data and we observed that it performs well when assessing the average expression level of the detected genes, when compared with GREAT. For these reasons, we focus our analysis in the interactions supported by HiChIP (Fig.2a).

A: VENNY

B: GREAT-expression, 4809

C: GREAT&HiChIP-expression, 1726

Figure[reviewer2_point16]. Overlap between enhancer-target gene prediction methods. A) Venn diagram depicting the number of genes that interact with PsE based on the 2 methods

(HiChIP and GREAT). HiChIP PsE-interacting genes (≥ 1 interactions, ≤ 100 kb distance): 6174; GREAT PsE-interacting genes (≤ 100 kb distance): 4809; $\sim 64\%$ of GREAT genes (3083) are not observed via the HiChIP interactions data. $\sim 72\%$ of HiChIP genes (4448) are not observed via the GREAT associations data. B) Average expression of GREAT-associated genes (4809 genes). C) Average expression of genes associated to PsE by both methods (172 6genes).

17) *“Performing a similar assay using the transcriptome of whole zebrafish embryos from 18 developmental stages (20), genes associated to DevE have shown an increased average expression comparing to all transcribed genes, with a similar dynamic to the enhancer activation during development (Fig.2b; Fig.S1e).” It is not clear what it means to perform a similar assay. Was this HiChIP-seq? If this was a typo and it was meant to say analysis, it is not clear how one can do a similar analysis if the specific data used are not clearly defined.*

Response:

We appreciate this comment from Reviewer2 and agree that the original sentence should be better explained in the main text. Our intention was to say ‘an analysis’. Additionally, in the revised version of the manuscript we now describe what the analysis consists of, and which data was used. These modifications are in the section of the Results “Functional similarities between human and zebrafish pancreatic enhancers”.

18) *In the abstract "Among several disease-associated sequences, we identified a zebrafish ptf1a distal enhancer whose deletion generates pancreatic agenesis, demonstrating the causality of this condition in humans." The logic of this sentence is unclear and as written suggests that pft1a zebrafish enhancer was disease associated (from the manuscript citation there was a human enhancer that was clearly shown to be disease-causing based on human genetics and functional studies). Since others have shown that homozygous human mutations in a distal PTF1A enhancer lead to pancreatic agenesis, it is not clear how studying a non-orthologous zebrafish enhancer near the ortholog of PTF1A can further demonstrate the disease causality of the homozygous deletion in the distal human enhancer. Given the demonstration that homozygous mutations in one human PTF1A enhancer causes pancreatic agenesis (PMC4131753), I fully agree the authors chose a relevant and exciting model to study. However looking over the human sequence coordinates given (either in hg19 or hg38) suggests that the authors did not test the actual*

enhancer that where mutations were demonstrated to cause the human disease (PMC4131753: positions 23508305A>G, 23508363A>G, 23508365A>G, 23508437A>G and 23508446A>C on chromosome 10). The choice for not including the actual disease associated enhancer in the comparative analysis of this locus and validation strategies should be explained given the emphasis on the disease association in the abstract and the overall presentation of the paper. Given the clear pancreatic expression and function of this pancreatic transcription factor it is not surprising that mouse and human both have enhancers that drive reporter gene expression in the zebrafish pancreas. However, it is still of interest knowing what the repertoire of pancreatic enhancers are and if there are potentially redundant enhancer or “shadow enhancers” underlying PTF1A/ptf1a regulatory logic and a more systematic analysis of the enhancers within the authors regions of interest.

Response:

We appreciate the comments by Reviewer2 and we would like to address them in separate points:

18.1) *In the abstract "Among several disease-associated sequences, we identified a zebrafish ptf1a distal enhancer whose deletion generates pancreatic agenesis, demonstrating the causality of this condition in humans." The logic of this sentence is unclear and as written suggests that ptf1a zebrafish enhancer was disease associated (from the manuscript citation there was a human enhancer that was clearly shown to be disease-causing based on human genetics and functional studies). Since others have shown that homozygous human mutations in a distal PTF1A enhancer lead to pancreatic agenesis, it is not clear how studying a non-orthologous zebrafish enhancer near the ortholog of PTF1A can further demonstrate the disease causality of the homozygous deletion in the distal human enhancer.*

We agree that the original sentence should be better explained in the abstract. In the revised version of the manuscript we now specify that this demonstration is *in vivo*, and we clearly acknowledge the previous association between the enhancer disruption and pancreatic agenesis. Respecting the format of *Nature Communications*, we did not add a reference to the abstract, nevertheless the work of Weedon and colleagues is cited in the introduction and throughout the manuscript. The respective sentence now reads: “Among other disease-associated sequences, we identified a zebrafish *ptf1a* distal enhancer whose deletion causes pancreatic agenesis, a phenotype

previously described in human patients that display mutations in a distal enhancer of *PTF1A*, demonstrating the causality of this condition *in vivo*".

Weedon and colleagues have very elegantly demonstrated a correlation between the distal *PTF1A* enhancer deletion and pancreatic agenesis in humans using human genetics and *in vitro* functional studies. In our work we add an extra layer of resolution to this demonstration introducing an *in vivo* model that recapitulates the phenotype observed in humans. We would like to highlight the importance of the *in vivo* modeling in at least 2 main points: 1) It expands the limitations usually inherent to less complex models as the *in vitro* ones, in particular in developmental processes, where pancreatic progenitor cells are actively interacting by signaling with a multitude of other cell types, and 2) it allows to better understand the dynamics of the development of the observed phenotype.

18.2) *Given the demonstration that homozygous mutations in one human PTF1A enhancer causes pancreatic agenesis (PMC4131753), I fully agree the authors chose a relevant and exciting model to study. However looking over the human sequence coordinates given (either in hg19 or hg38) suggests that the authors did not test the actual enhancer that where mutations were demonstrated to cause the human disease (PMC4131753: positions 23508305A>G, 23508363A>G, 23508365A>G, 23508437A>G and 23508446A>C on chromosome 10). The choice for not including the actual disease associated enhancer in the comparative analysis of this locus and validation strategies should be explained given the emphasis on the disease association in the abstract and the overall presentation of the paper.*

We apologize for this mistake. The coordinates were incorrectly annotated in Supplementary Table 4a, likely due to a human error in the transposition of the information when compiling the respective Supplementary Table 4a. Nevertheless, we would like to clarify that the enhancer sequence amplified and tested in our work is the same sequence used in the luciferase assays of the paper highlighted by Reviewer2 (PMC4131753, GRCh37/hg19 chr10:23507926+23508669, 744bp). We made available below, for Reviewer2's appreciation, a diagram representing the landscape of *PTF1A*, having annotated the sequence tested by Weedon and colleagues, the sequence tested in our work and the respective SNP sites mentioned by the reviewer (PMC4131753: positions 23508305A>G, 23508363A>G, 23508365A>G, 23508437A>G and 23508446A>C on chromosome 10). Additionally, we would like to highlight that the list of

primers annotated in Supplementary Table 4 from the initial version of the manuscript, used to amplify the human sequence, unambiguously target the respective human enhancer. We have corrected the human coordinates in Supplementary Table 4a in the revised version of the manuscript and we have confirmed that all the remaining coordinates are correct. We thank Reviewer2 for noticing such incongruence and helping us to improve the manuscript.

Figure [reviewer2_point18]. Human PTF1A landscape (above) and zoom-in to the human distal PTF1A enhancer region (hP.E3) depicting ChIP-seq density plots for the enhancer mark H3K4me1 (black), for FOXA2 (blue), and for PDX1 (green) from human embryonic pancreatic progenitors. The hg19 genomic coordinates of the sequence tested by Weedon et al. (PMID: 24212882) and the sequence tested in the current work (Bordeira-Carriço et al.) are depicted below the zoom-in ChIP-seq tracks, along with pancreatic agenesis-associated point mutations (PMID: 24212882) (purple lines) labeled by the the final 3 digits from the hg19 coordinates (referring to positions 23508305A>G, 23508363A>G, 23508365A>G, 23508437A>G, and 23508446A>C on chromosome 10 respectively).

18.3) *Given the clear pancreatic expression and function of this pancreatic transcription factor it is not surprising that mouse and human both have enhancers that drive reporter gene expression in the zebrafish pancreas. However, it is still of interest knowing what the repertoire of pancreatic enhancers are and if there are potentially redundant enhancer or “shadow enhancers” underlying PTF1A/ptf1a regulatory logic and a more systematic analysis of the enhancers within the authors regions of interest.*

There are already several studies across three vertebrate species (mouse, human, and zebrafish) that show that *Ptf1a* expression is regulated by multiple enhancers, scattered throughout the *ptf1a/PTF1A* regulatory landscape, including: 1) a highly conserved autoregulatory enhancer, 2) a proximal downstream early-acting enhancer, and 3) a series of tissue-specific distal downstream enhancers, which likely include redundant enhancers (summarized in PMID: 34125483). We agree with Reviewer2 that understanding the regulatory landscape of *ptf1a/PTF1A* in more detail is very interesting, however, to properly address these questions it would require multiple experiments including mutations in multiple enhancers to functionally understand the requirement of each one both individually and in combinations, including putative redundant functions. Although very interesting, and a worthy subject for a project of its own, this would diverge from the main focus of the current work, which is to explore pancreatic enhancers potentially associated with disease using zebrafish as a tool, and to establish an approach to allow the establishment of a correlation between zebrafish and human enhancers, resorting to some examples as a proof-of-principle.

We would like to highlight that we annotated the previously described enhancers in the *ptf1a* regulatory landscape (Fig.5; zP.E1 and zP.E2). Additionally, and because we agree with the suggestion of Reviewer2, that enhancer redundancy and the possibility of the existence of shadow enhancers is very interesting and important to discuss, we have improved the discussion of the revised version of the manuscript to include this subject.

19) *“Overall, we show that chromatin profiling can help to uncover interspecies functional equivalency of cis-regulatory elements”. This claim suggests that a systematic analysis was performed and presented in the paper. I do not see how chromatin profiling can do this. I do agree that the authors have done excellent work to demonstrate that specific human enhancers at syntenic loci in human and zebrafish can drive similar expression, but a lot of the decisions on*

what to validate likely came from the authors intuition/prior knowledge rather than from the data they generated.

Response:

We appreciate Reviewer2's concern regarding the clarification of this sentence, allowing us to improve the manuscript. We now have changed this sentence to be more specific and detailed to avoid misinterpretations. We believe that this sentence now summarizes properly the presented work. Indeed, we have profiled the chromatin of zebrafish pancreatic cells, including open chromatin, presence of H3K27ac and interactions and combining with *in vivo* reporter assays we have determined interspecies enhancers that contain a similar regulatory information and that regulate the same target gene. As proof-of-principle we selected 2 of the validated enhancers for this subsequent approach, largely based on their putative target genes and the role of the human orthologue gene in pancreatic disease: specifically, *PTF1A* is determinant for pancreas development and function, and *ARID1A* has been implicated in the development of pancreatic cancer.

20) *“To determine if the identified regulatory sequences are active pancreatic enhancers, we have performed in vivo reporter assays for 10 regions with strong H3K27ac signal and 7 with low levels of this mark (Fig.2c-d; Fig.S2a). From the first set, we have found that 5 out of 10 tested sequences ($H3K27ac: -\log_{10}(p\text{-value}) \geq 35$) are pancreatic enhancers (50%; Fig.2c-d and Table S4). In contrast, from the regions with low H3K27ac signal, only 1 out of 7 tested (14%, Fig.S2a) showed strong and reproducible pancreatic enhancer activity ($-\log_{10}(p\text{-value}) < 35$). These results validate the robustness of the enhancers prediction based on chromatin state.”* It is not clear how these regions were chosen for testing and what defines high and low levels of H3K27ac. Without appreciating the criteria for selecting these regions the statistical tests performed are not as informative as it could be and would be difficult to reproduce. I think it is well established that cell-type enhanced H3K27ac signal is a useful criteria for selecting enhancer active in a particular cell-type. However, its not clear here how this select set of 17 regions were chosen. (i.e. to what extent was evolutionary conservation considered or ATAC-seq signal or the literature on nearby genes considered). The reason I ask this is that the authors use these examples to make broad conclusions about predicting enhancers with “functional equivalency”. To what

extent would studying putative enhancers with H3K27ac signal that surround orthologous genes with known pancreas expression or disease associations, with or without evidence for 3D chromatin interactions, serve as a predictor of functional equivalency?

Response:

We thank Reviewer2 for bringing these points to our attention, allowing us to clarify them.

Regarding the selection of sequences for the *in vivo* validation by reporter assays, we had to follow restricted rules of prioritization, since the *in vivo* testing is highly demanding in time and resources. We selected the sequences to test by *in vivo* enhancer reporter assays based on the overlap between H3K27Ac ChIP-seq and ATAC-seq signal, within the genomic landscape of pancreas relevant genes, as described in Materials and Methods section. Specifically, we chose genomic landscapes of genes known to be crucial for pancreas function and/or with putative roles in pancreatic diseases (*AMY1A*, *CPA1*, *CPA5*, *CTRB1*, *PTF1A*, *PRSS1*, *ARID1A* and *TP53*). This selection was important for the subsequent objectives of this work, that involved the phenotypic characterization of the loss-of-function of the validated enhancers, since we needed to have a relatively clear idea about the outcome of the loss-of-function of the enhancer, in contrast to selecting regulatory landscapes of genes with unknown pancreatic functions. We would like to highlight that the main focus of this work is to address the impact of cis-regulatory mutations in pancreas function and human disease.

Regarding the H3K27ac signal, we agree with Reviewer2 that "... it is well established that cell-type enhanced H3K27ac signal is a useful criteria for selecting enhancers active in a particular cell-type.". For this reason, we divided sequences tested for enhancer activity in two groups, based on their H3K27ac signal. We considered the enhancers with the top10 H3K27ac signal as the "high H3K27ac group", and the remaining 7 as the "low H3K27ac group". We considered this to be a fair division since it corresponded to approximately the midpoint of the range of H3K27ac values ($-\log_{10}(p\text{-value})$ maximum value:92 and minimum value:19). Nevertheless we agree that the way we present these results may suggest that the value $-\log_{10}(p\text{-value}) < 35$ is a specific threshold for a better prediction of pancreatic enhancers, which is not the case, nor do we attempt to state this. To avoid this misinterpretation, we have now improved the description of this part of the Results. Additionally, to make clear the point that H3K27ac signal is a useful criterion for the selection of the best candidate sequences to be enhancers, consistent with other published works (PMID:

32728240; PMID: 24360275), we have now included a graph displaying the distribution of H3K27ac signal intensity within the group of “validated enhancers” and “non-enhancers” (Supplementary Fig. 3c), showing that validated enhancers have on average higher H3K27ac levels than the predicted sequences that tested negative for pancreatic enhancer activity.

Regarding the last point raised by Reviewer2 “...*To what extent would studying putative enhancers with H3K27ac signal that surround orthologous genes with known pancreas expression or disease associations, with or without evidence for 3D chromatin interactions, serve as a predictor of functional equivalency?*”: The identification of pancreatic enhancers associated to orthologue genes can help to identify functionally equivalent enhancers. We would like to highlight that the vast majority of the zebrafish pancreatic enhancers that present some sequence conservation between human and zebrafish, do not share marks for enhancer activity in human pancreatic tissue: only 12.49% of putative zebrafish pancreatic enhancers could be directly aligned to the human genome, and only 10.23% of these human sequences also overlapped with H3K27ac signal from human pancreas, suggesting that sequence conservation, in these cases, is not directly related with the pancreatic enhancer activity, as discussed in the response to Reviewer2, point 4. For this reason, and in the absence of enough sequence conservation to trace orthologue enhancers, it is instead necessary to identify where the pancreatic enhancers are located in the genome and which are their target genes. The identification of pancreatic enhancers can be done by looking at chromatin availability and epigenetic marks associated with enhancers, such as H3K27ac, as we have done in this work. The H3K27ac signal or chromatin availability *per se* is not an evidence of functional equivalency, but it is a relevant predictor because equivalent functional enhancers are expected to be active in the same tissue, and to control orthologous gene/genes. *In vivo* reporter assays can help to address if similar regulatory information is present in putative functional equivalent enhancers, which translates into a similar expression pattern of reporter expression. Finally, we would like to clarify that this type of analysis alone, is not a predictor of functional equivalency. Nevertheless, by studying these sequences, we can validate functional equivalency, in particular by the addition of a comparison of the phenotypes associated with the loss-of-function of such sequences in each species, a comparison that we established for the *ptfl1a/PTF1A* distal enhancer.

21) Page 6 “Although only a minority of interspecies aligned sequences shared H3K27ac signal (Total pancreas data set: 229 out of 1842; PsE: 116 out of 1052; DevE: 113 out of 790), there is a clear enrichment comparing to random sequences (Fig.3c), although not showing a higher average sequence conservation score (Fig.3b). These results suggest that pancreatic enhancer function is not a strong constraint for sequence conservation.” There is a lot of information here that would benefit from further explanation. Intuitively, sequences that can be aligned between fish and human should show more sequence constraint than random sequences unless such a large window was taken that it obscures such sequence constraint. Could the ATAC-seq peak within the H3K27ac peaks be used for this calculation? What does “pancreatic enhancer function is not a strong constraint for sequence conservation” mean?

Response:

We agree that this sentence is not clear in the manuscript, we have improved it and we thank Reviewer2 for raising this point. The respective sentence now reads: “Only a minority of interspecies aligned sequences shared H3K27ac signal (total pancreas data set: 227 out of 1842; PsE: 115 out of 1052; DevE: 112 out of 790). The human sequences, that shared H3K27ac signal with zebrafish, did not show a higher average conservation score than the aligned sequences that showed H3K27ac signal in zebrafish alone (Fig.3b and Supplementary Fig.3e; Average sequence conservation score for H3K27ac non-shared vs shared signal, Pancreas: 0.40vs0.36, PsE:0.42vs0.41, DevE:0.36vs0.34). Notwithstanding the low absolute numbers of aligned sequences that share H3K27ac signal in human and zebrafish pancreas, these sequences represent a clear enrichment compared to the overlap obtained by randomized set of sequences in the human genome (3.21 times higher for pancreas, 2.79 times higher for PsE, 3.76 times higher for DevE and 1.76 times higher for embryo, Fig.3c; Supplementary Table 3q). Overall, these results suggest that pancreatic enhancer function is not a strong constraint for sequence conservation”. In the same direction, we have now clarified that all the compared sequences are aligned between both species, zebrafish and human, but one subset shares H3K27ac mark in pancreatic cells of both species, while the other subset of sequences has pancreatic H3K27ac mark in zebrafish alone. For these two subsets of sequences, the average conservation score is very similar. Therefore, because H3K27ac is a mark for enhancer activity, aligned sequences that share H3K27ac in human and zebrafish should have a higher conservation score, assuming that pancreatic enhancer function entails strong evolutionary constraint. However, we observed that both sets of sequences had

similar average conservation scores. For this reason, we concluded that "...pancreatic enhancer function is not a strong constraint for sequence conservation".

In regard to the question "Could the ATAC-seq peak within the H3K27ac peaks be used for this calculation?", we have done an exercise in which only the portions of H3K27ac peaks that overlapped with ATAC-seq signal were considered. Indeed, using ATAC-seq to filter the H3K27ac ChIP-seq peaks leads to higher conservation scores, at the cost of fewer of such sequences aligning with human (Supplementary Fig.3e). These results are now included in Fig. S3e. Similar to what was previously observed in the H3K27ac peaks not filtered by ATAC (Fig.3b), the ATAC-filtered peaks did not show a higher average conservation score in the aligned sequences with H3K27ac zebrafish/human shared signal than in the aligned sequences that showed H3K27ac signal in zebrafish alone (Average sequence conservation score for ATAC restricted H3K27ac non-shared vs shared signal, Pancreas: 0.50vs0.38, PsE: 0.55vs0.27, DevE: 0.46vs0.47; Number of aligned sequences restricted to the ATAC peak and with shared H3k27ac: Pancreas: 73, PsE:33, DevE:40). Additionally, Fig.3c has been approached in more detail in the answer to question 5 from Reviewer1.

Finally, we would like to state that during the revision process we found a small mistake in the number of interspecies aligned sequences, that we corrected in the revised version of the manuscript (Original version: total pancreas data set: 229 out of 1842; PsE: 116 out of 1052; DevE: 113 out of 790. Revised version: total pancreas data set: 227 out of 1842; PsE: 115 out of 1052; DevE: 112 out of 790).

22) *"Then, we wanted to assess whether functionally equivalent pancreatic CREs might exist between human and zebrafish, despite an overall lack of sequence conservation. To explore this possibility, we analysed if the human ortholog genes coupled to each cluster of zebrafish enhancers were enriched for human pancreatic diseases" Until this point, Clusters of enhancers or super enhancers were not mentioned and it seems odd that the authors did not acknowledge prior comparative work comparing human and zebrafish super enhancers (PMID: 27965291). It would be relevant to know how calling super-enhancers in each species, linking them to target genes with HiChIP; and then looking into aspects of conservation/functional equivalency and disease*

association compares to the current stratification of H3K27ac regions where adult zebrafish H3K27ac peaks were compared to a reference dataset of bulk zebrafish embryo H3K27ac ChIP-seq?

Response:

We do agree that extra information should be present to introduce the super enhancers to the reader. We have improved the revised version of the manuscript by adding a short introduction to super enhancers, including respective references. We would like to clarify that, indeed, in this experimental setting we have called super enhancers in zebrafish and human, using standard computational methods (please see the Materials and Methods section). Then we linked super enhancers to genes. This link was done using GREAT instead of HiChIP to be consistent with the approach for human and zebrafish datasets, since HiChIP data for the human whole pancreas is not available, with HiC being the closest type of available datasets (PMID:27851967). To address the possible functional equivalency, we explored gene ontology terms for the human and zebrafish associated genes, to better understand if the super enhancers were controlling similar types of genes. Additionally, functional equivalent enhancers are expected to control the expression of orthologous genes, therefore, we observed the overall correspondence between zebrafish and human orthologue genes. We have now improved the revised version of the manuscript by better explaining the rationale of these analyses.

23) *While performing TF analysis is an essential step and should be performed, the analysis as presented is challenging to follow and in places seems arbitrary. For example, why were 140 motifs chosen? It was not clear in the main text or methods what human data was used. Ideally the data type and a citation of the original publications of the data used in the manuscript would be given.*

Response:

We thank Reviewer2 for bringing this to our attention. As mentioned by Reviewer2, TF motif analysis is an important step, it helps us to understand if the transcriptional machinery operates similarly in human and zebrafish pancreatic enhancers. We performed a search for motif enrichment for TF binding sites (TFBS) in regions of open chromatin identified by ATAC-seq within putative pancreatic enhancers for zebrafish whole pancreas (ZP), human whole pancreas

(HP), zebrafish embryos (D80 and 24HPF) and human heart ventricle (V). We added the last two datasets (zebrafish embryos (D80 and 24HPF) and human heart ventricle (V)) to the analysis as controls for the zebrafish and human pancreas, respectively, as we do not expect to find a large overlap of enriched TF motifs in these datasets with the one obtained from the pancreatic enhancers datasets. From the resulting analysis we selected the top 140 enriched motifs for each tissue. We required a smaller set of motifs and opted to select 140 to proceed to the next part of the analysis. Although the choice for the population size was arbitrary, several parameters were considered: 1) The population size should not be so small that the probability of overlapping motifs among the different tested groups would be very small, nor too large, ideally smaller than 50% of all the motifs tested for enrichment. 140 motifs correspond to 32% (1/3) of the total dataset of motifs used for the motif enrichment discovery and 2) For all groups, all the enriched motifs within the top 140 had a *p-value*<0.01. Using this population size, we proceeded to the comparisons. We observed that the majority of shared motifs were found in zebrafish (ZP) and human (HP) pancreas datasets (ZP,HP:98, versus ZP,D80:63 and HP,D80:61) (Fig.3g), while comparisons with the human ventricle (V) showed that ZP,V was the second largest group (ZP,V:89) (Supplementary Fig.4b-f). These results suggest that a similar set of TFs operate predominantly in human and zebrafish pancreatic enhancers.

Because there are groups of TFs known to be important in pancreas function and development in mammals, we reasoned that they should be overrepresented in the group of enriched motifs detected in zebrafish and human pancreatic enhancers. Therefore, we decided to focus on 25 TFs known to be important to pancreas development and function. As expected, we found that the majority of these TFs were within the ZP,HP overlapping datasets, regardless of the compared groups (Supplementary Fig.4d-fj), suggesting that the same set of pancreas relevant TFs is operating in both zebrafish and human pancreatic enhancers.

Following Reviewer2's suggestion, we have now improved this section of the Results, clarifying the TF motif analysis. Regarding the public available datasets for human pancreas and ventricle datasets, as well as zebrafish embryos (D80, dome and 80%epiboly and 24HPF), this information was summarized and made available in Supplementary Table 4t-u, which is now referenced in the Material and Methods section. Because this information was not visible enough, and to address Reviewer2's comment, we have now added a reference to T Supplementary able 4

in the main text which reads “Datasets summarized in Supplementary Table 4t-u”. Additionally, we added to the main text all the references to the publications where these datasets were retrieved.

24) *“These results suggest that the same set of TFs might operate in zebrafish and human pancreas enhancers.” I would think that it would be expected a priori that vertebrate pancreas TFs would be shared between species and that there would be ample literature to cite and suggest targets. It would be useful to clarify and/or make such expectations in the introduction.*

Response:

Following Reviewer2’s suggestion, we have improved the revised version of the manuscript by adding several references to TFs important for pancreas function and development. Additionally, we now introduce this part of the results by stating a clear expectation regarding the presence of enriched TFBS in human and zebrafish pancreatic enhancers. The respective sentence now reads: “Several TFs, such as Ptf1a, Pdx1, Pax6 and Sox9, are known to be important for pancreas function or development in several vertebrate species, including human and zebrafish (PMID: 32894307, PMID: 24413736, PMID: 25915126, PMID: 34125483). As shown above, human and zebrafish pancreatic enhancers are enriched for many shared TFBS, therefore it is reasonable to expect that many of these TFBS correspond to known pancreas TFs.”

25) *Details about the number of replicates for RNA-seq for adult zebrafish were not given. The DATA tab in the supplemental spreadsheet does not indicate which biological replicates for the authors own data as well as public data were used.*

Response:

We thank Reviewer2 for the comment. We have clarified this aspect by adding the number of biological replicates to Supplementary Table 4g. Regarding our own adult zebrafish datasets of RNA-seq, we used 4 biological replicates for whole pancreas samples, 4 biological replicates endocrine samples and 2 biological replicates muscle samples. Supplementary Table 4g was also improved to better discriminate which RNA-seq datasets were generated by our lab and from other labs, including references to those works.

26) *There seems to be information missing regarding how the authors processed their own ChIP-seq data as well as published ChIP-seq data (it would not be possible to redo this analysis with the information given). How were biological replicates handled? It is mentioned for the H3K27ac that biological replicates were compared but not how the data was used (e.g. were the peaks combined or were the replicates combined prior to peak calling.) There does not appear to be any mention of the use of input controls for ChIP-seq peak calling analysis which is highly unusual. This should be clarified and the justification for not using input controls should be clearly made in the method if this is not the case and how this impacts the data analysis. Were input controls also not used for the human data and developmental datasets utilized by the authors? This seems unusual.*

Response:

We thank Reviewer2 for the comments. We have now improved the Material and Methods section of the revised version of the manuscript, adding more details regarding ChIP-seq data processing. These changes are annotated in Material and Methods section, II) Processing and Bioinformatic analysis.

ChIP-seq peak calling analysis was performed without a control input sample. MACS, which is the software used to identify peaks, has been developed and tested to be used with or without control input, making the use of controls not mandatory. Even without a control input, MACS can accurately detect enriched ChIP regions. This has been described in the MACS original work (PMID: 18798982) and in works further developed by their authors (PMID: 21633945). During peak calling we have skipped the model building step by setting `-nomodel` in the command line, as recommended by MACS's authors (PMID: 21633945). Several works published have implemented a similar type of approach, opting for not using input controls for peak calling. We have collected a list of 10 published works as examples of similar approaches described: (PMID: 30922236; PMID: 32883883; PMID: 27381023; PMID: 26697317; PMID: 26823433; PMID: 27049946; PMID:17512414; PMID: 29509191; PMID: 29146583; PMID: 22138689).

To address the concern of the identification of false positive peaks, we have now analyzed a “blacklist” of potentially problematic genomic regions, generated by several input samples in different zebrafish tissues. The used datasets were the following: DCD002894SQ, DCD002921SQ, DCD003653SQ, DCD003654SQ, DCD003671SQ and DCD002742SQ. MACS was used in these datasets for peak calling, using the same conditions described in Materials and

Methods section of the current manuscript, and peaks were selected when present in at least 5 out of 6 datasets. This generated 156 peaks, from which 102 overlap with only 69 peaks from the list of 14753 putative enhancers, representing less than 0,5% of the total dataset. This is now highlighted in the Materials and Methods section of the revised version of the manuscript. Importantly, we have confirmed that none of these regions overlapped with selected putative enhancers used for *in vivo* validation by enhancer reporter assays or genomic deletions. The blacklist of peaks is now included in Supplementary Table 1o and the unreliable peaks that overlap with our dataset is now annotated in Supplementary Table 1a. Regarding the embryo ChIP-seq datasets from the work by Bogdanovic and colleagues (PMID:22593555), an input control was not used by these authors (<https://www.ncbi.nlm.nih.gov/geo/query/acc.cgi?acc=GSE32483>). To maintain exactly the same conditions, we processed human samples also without an input control. Using the same conditions as above, we have used a published human blacklist of unreliable peaks (PMID:31249361) and observed that these represent 192 out of 102548 of the human called peaks, representing as little as 0.2% of the identified peaks. We have now added this information to the Material and Methods section, to clearly show that unreliable peaks represent only a very small fraction of the used datasets.

27) The DisGeNET analysis seems to be a central aspect of defining regions for study in this paper. It is not clear in the main text how pancreas disease genes were obtained. Reading the methods, it was not clear how many genes the authors obtained and how the putative enhancers associated with these genes compares with the set of zebrafish pancreatic enhancers. Given the prominence of the disease angle in the abstract, I think this idea should be more formally and systematically introduced and used in the paper.

Response:

We thank Reviewer2 for raising this point which we clarify in the next lines. The main rationale for the use of DisGeNET was to determine if the human orthologues of the zebrafish genes targeted by the pancreatic enhancers were enriched for roles associated with pancreatic diseases. This was important for several reasons: 1) This shows that the orthologous genes in human are functionally active in pancreatic tissue, therefore it is likely that these genes have CREs that define their expression in this tissue, which could be potentially functionally equivalent enhancers to the

zebrafish ones. This hypothesis is further developed in the subsequent parts of the manuscript, but we consider to be the best starting point to approach this question. 2) We demonstrate that identified zebrafish regulatory landscapes belong to potentially biomedical relevant genes. 3) The zebrafish regulatory landscapes can be further explored from the pancreatic disease point of view. As suggested by Reviewer2, we have now improved the introduction of this idea in the respective results section.

To clarify the doubts raised by Reviewer2, we would like to better explain in detail the experimental procedure we have followed. Pancreatic diseases and their associated genes were selected from the file containing all gene-disease links from DisGeNET (all_gene_disease_associations.tsv, downloaded from the DisGeNET website, v6.0) filtering for associations with a score >0.1 to exclude those based only on text-mining. The disease search term used was “pancrea*”, followed by manually filtering for pancreas-related diseases and their human associated genes. In the next step, we used the homology table between zebrafish and human genes (exported from the Ensembl BioMart for DanRer10) to derive the set of zebrafish genes for each pancreatic disease. We required a minimum of 15 genes relating to a disease to avoid significant gene set enrichments only due to small group ratios without real over/under representations. This search yielded 16 pancreatic diseases containing a total of 836 zebrafish homologs (306 of them interacting with pancreatic enhancers, 36.6%) of human genes associated with pancreatic diseases (Supplementary Table 3r). To check whether the genes interacting with various enhancer clusters (Embryo only, C1, C2, C3, C4, PsE) are enriched for pancreas disease-association, we performed hypergeometric tests for gene set enrichment (R phyper function, X: number of genes in disease A_i and in enhancer set B_i ; M: number of genes in disease A_i , N: non-disease genes – number of zebrafish protein coding genes minus M; K: number of genes in enhancer set B_i). For each disease with ≥ 15 zebrafish genes homologous to the original human genes associated to pancreatic diseases from DisGeNET, we checked if their overlap with each enhancer set was higher than expected by chance, using the set of zebrafish protein coding genes extracted from Ensembl BioMart as background gene set. Indeed, we observe various pancreatic disease-association enrichments on several clusters, namely C1, C3, C4 and PsE, whereas embryo and C2 show no or low enrichment, suggesting the link between regulation of late development and adult pancreas function and pancreatic diseases. Accordingly, we have now improved the Materials and Methods "Disease association enrichment of genes from different enhancer clusters".

28) *The authors used H3K4me3 HiChIP to capture the enhancers that interacts with gene promoters and RNA-seq as a read out for gene expression. In Fig.1b, they should show a track of RNA-seq signal as well for their gata6 example.*

Response:

We appreciate Reviewer2's comment and agree that it would be beneficial to show RNA-seq data for the regions depicted in Fig.1b. We have modified the figure accordingly: above the H3K4me3 HiChIP in panel b, we added a track of adult zebrafish exocrine pancreas RNA-seq signal (in green). The panel now shows (top to bottom) ChIP-seq for H3K27ac, ATAC-seq, RNA-seq showing that *gata6* is actively transcribed in pancreas, and the heatmap representation of HiChIP interaction signal, showing a putative enhancer highlighted in blue interacting with the *gata6* promoter.

30) *In Figure 2b, the author examined the HC/AIIG ratio across development. It would be interesting to add in an adult timepoint to see if there is a significant increase for the genes interacting with PsE.*

Response:

We thank Reviewer2 for the useful comment and we agree with the Reviewer2's assessment that adding an adult timepoint to the graph would be interesting. Accordingly, we analyzed the ratio between average expression of genes interacting with pancreas-specific enhancers (PsE, C1, C2, C3 and C4) and the average expression of all genes throughout zebrafish development, adding 5 new datapoints that correspond to adult tissues: Whole pancreas, Endocrine, Acinar, Duct and Muscle (figure below). The first 4 new datapoints correspond to adult pancreatic tissues while the last one corresponds to muscle. As predicted by Reviewer2, there is a very strong increment in the HC/AIIG ratio for the genes that interact with all described enhancers clusters for at least one pancreatic tissue, while the HC/AIIG ratio remains very low for the genes that interact with all described enhancers clusters when analyzing their expression in the muscle. We have now included this data in a graph in Supplementary Fig. 2d.

Figure[reviewer2_point30]. Ratio between the average expression of genes interacting with pancreas-specific enhancers (PsE, C1, C2, C3 and C4 clusters) and the average expression of all genes throughout zebrafish development (BDO: blastula, dome; G50: gastrula, 50% epiboly; GSH: gastrula, shield; G75: gastrula, 75% epiboly; S1-4: segmentation, 1-4 somites; S14-19: segmentation, 14-19 somites; S20-25: segmentation, 20-25 somites; PP5: pharyngula, Prim-5; PP15: pharyngula, Prim-15; PP25: pharyngula, Prim-25; HLP: hatching, long-pec; LPM: larval, protruding-mouth; LD4: larval, day 4; LD5: larval, day 5) and different pancreatic zebrafish tissues (whole pancreas, endocrine, acinar, duct and muscle).

31) *In theory genes interacting with PsE, even if they are expressed in development, they are not activated by PsEs. In Figure 2b, the author still observed a higher expression level of these genes compared to all genes in certain developmental stages (e.g. S20-25). Are these PsE genes being regulated by other sets of enhancers such as DevEs? What is the percentage of enhancers that are overlapped between PsE associated genes and DevE associated genes? For those overlapped genes, how do their gene expression change during development?*

Response:

We thank Reviewer2's interesting question. To address this question, we have performed the following analysis: We determined how many genes associated with PsEs and DevEs are shared,

meaning that they share at least 1 PsE and 1 DevE enhancer. We found that there are 6174 genes associated to PsE and 5449 associated to DevE (HiChIP; $\leq 100\text{kb}$ distance and ≥ 2 interactions per chunk), of which 2783 genes are shared (45.05% of the PsE-associated genes or 51.07% of the DevE-associated genes). These results are graphically represented in a Venn diagram below. Therefore, as predicted by Reviewer2, there are a considerable number of genes that contain at least 1 PsEs and 1 DevE in their landscapes, which can explain the results observed in Fig.2b, regarding the high HC/AllG ratio (HC/AllG ratio explained in detail in the previous point 30 raised by Reviewer2) for the group of genes associated to PsEs in some developmental stages (e.g. S20-25). Following, as suggested by Reviewer2, we looked at the average gene expression and HC/AllG ratio in different developmental times of the genes controlled by PsE alone, DevE alone and by PsE and DevE (PsE+DevE). As expected, we found that the genes interacting with DevE alone have a higher HC/AllG ratio than the genes interacting with PsE alone. However, the genes interacting with both PsE and DevE (PsE+DevE) showed the highest HC/AllG ratios. We present the respective graphs below to share with Reviewer2. These results suggest that genes with more complex regulatory landscapes tend to have higher levels of expression. This can be better visualized when addressing the remaining question of Reviewer2, *“What is the percentage of enhancers that are overlapped between PsE associated genes and DevE associated genes?”*. We observed that the 3391 PsE-only-associated genes interact with an average of 1.43 enhancers (median of 1 enhancer, with a maximum of 12 enhancers), the 2666 DevE-only-associated genes interact with 1.45 enhancers on average (median of 1 enhancer, with a maximum of 15 enhancers), while the 2783 PsE-and-DevE-associated-genes interact with an average 2.19 enhancers (median of 2 enhancers, with a maximum of 22 enhancers).

Figure [reviewer2_point31]. A) Venn diagram depicting the overlap of genes interacting with PsE and DevE enhancers. B) For each of the 3 gene subgroups shown in A (PsE: 3391, DevE:2666, PsE+DevE:2783) the average gene expression (transcripts per million, TPM) detected by RNA-seq from zebrafish embryos at different developmental stages (0 to 120hpf) is shown. The horizontal bars swap in the order of all genes (AllG) and genes interacting with enhancers from the respective subgroup (PsE, DevE, DevE+PsE) as detected by HC in adult zebrafish pancreas.

32) *Is it possible for the authors to show the percent sequence identity between human and fish for the two enhancers they tested (A.E3 and P.E3)? Just wanted to get an idea of how poorly conserved they are. Or is it simply they don't align?*

Response:

To address this question we have performed BLAT and BLAST analysis for the zebrafish and human enhancer sequences located in the landscapes of *ptfla/PTF1A* (zP.E3 and hP.E3) and *arid1ab/ARID1A* (formerly zA.E3 and hA.E3, now named zA.E4 and hA.E4).

Nucleotide sequence analysis of hP.E3 and zP.E3 using BLAST yielded a Needleman-Wunch (NW) score of -2078.0, 32.0% identities, and 59.0% gaps (Figure a, bellow). To understand whether these values represent sequence similarity, we performed 5 control BLAST alignments, using random sequences with the same length. The average values for the control alignments were: -2188.8 (± 87.4) NW score, 30.4 (± 1.8)% identities, and 60.6 (± 2.2)% gaps (figure b, bellow). Thus, given that the BLAST scores obtained for the two functionally equivalent enhancers are similar to the scores yielded by random sequences, we concluded that the zP.E3 and hP.E3 sequences are as similar as randomly aligned sequences.

Likewise, BLAST between zA.E4 and hA.E4 yielded a NW score of -2235.0, 34% identities, and 54% gaps. We performed 5 control BLASTS with random sequences with the same length, and the average values of the controls were: -2285.2 (± 30.1) NW score, 33.8 (± 0.4)% identities, and 54.2 (± 0.4)% gaps (figure c, bellow). The scores obtained for zA.E4 and hA.E4 were very close to those obtained for the controls (figure d, bellow), suggesting that these sequences are as similar as randomly aligned sequences.

Finally, we performed BLAT searches of zA.E4 and zP.E3 in the human genome and hA.E4 and hP.E3 in the zebrafish genome. In all cases we obtained a low nucleotide match, and the genomic hits did not overlap with the genomic coordinates of *Arid1a* or *Ptfla* regulatory landscapes.

Figure[reviewer2_point32]: a) Nucleotide sequence analysis of hP.E3 (pink) and zP.E3 (blue) using BLAST. b) Nucleotide sequence analysis of hP.E3 control (pink; hCont) and zP.E3 control (blue; zCont) using BLAST. The controls were randomly generated with the same length of the sequence. c) Nucleotide sequence analysis of hA.E43 (violet) and zA.E43 (blue) using BLAST. d) Nucleotide sequence analysis of hA.E43 control (violet; hCont) and zA.E43 control (blue; zCont) using BLAST. The controls were randomly generated with the same length of the sequence.

PTF1A

Results of a blat of the human sequence (hP.E3) in zebrafish genome (danRer10). ptf1a is located at chr2:29,727,237-29,728,531 coordinates of the zebrafish genome (danRer10):

SCORE	START	END	QSIZE	IDENTITY	CHROM	STRAND	START	END	SPAN
32	500	568	736	97.20%	Chr7	+	10559846	10559915	70
29	618	687	736	96.8%	Chr12	+	15090612	15090702	91
26	484	518	736	72.5%	Chr13	+	37366353	37366381	29
22	679	702	736	95.9%	Chr12	-	38728368	38728391	24
22	109	132	736	87%	Chr13	+	5947284	5947306	23
21	708	728	736	100%	Chr11	-	22561591	22561611	21
21	703	723	736	100%	Chr15	+	42273328	42273348	21
20	38	57	736	100%	Chr13	-	3667757	3667776	20

Result of a blat of the zebrafish sequence (zP.E3) in human genome (hg19). PTF1A is located at chr10:23,481,460-23,483,181 coordinates of the human genome (hg19):

SCORE	START	END	QSIZE	IDENTITY	CHROM	STRAND	START	END	SPAN
27	842	873	1828	93.8%	Chr2	+	194789796	194789828	33
25	192	224	1828	85.2%	chr4	-	140880631	140880661	31
23	418	445	1828	80.8%	chr4	-	31257896	31257921	26

ARID1A

Results of a blat of the human sequence (hA.E4) in the zebrafish genome (danRer10). arid1ab is located at chr19:14,490,137-14,573,059 coordinates of the zebrafish genome (danRer10):

SCORE	START	END	QSIZE	IDENTITY	CHROM	STRAND	START	END	SPAN
23	1760	1783	2053	100%	chr24	-	24304786	24304811	26
20	1219	1238	2053	100%	chr1	-	53897941	53897960	20

Result of a blat of the zebrafish sequence (zA.E4) in the human genome (hg19). ARID1A is located at chr1:27,022,522-27,108,601 coordinates of the human genome (hg19):

SCORE	START	END	QSIZE	IDENTITY	CHROM	STRAND	START	END	SPAN
48	537	789	934	62.3%	chr1	-	90758130	90758182	53
41	415	561	934	97.7%	chr1	+	7540318	7540473	156
31	415	557	934	57.6%	chr1	+	218741692	218741751	60

30	537	786	934	47.1%	chr17	+	50130295	50130332	38
30	532	561	934	100%	chr1	+	243131887	243131916	30
29	537	785	934	43.4%	chr2	-	33218468	33218497	30
29	655	735	934	96.8%	chr1	-	2253578	2253697	120
28	661	703	934	83.8%	chr4	+	47670111	47670551	441
28	537	570	934	93.6%	chr1	+	179673446	179673479	34
28	534	561	934	100%	chr1	+	44714364	44714391	28
27	534	561	934	100%	chr17	+	1179153	1179184	32
26	534	560	934	100%	chr1	-	72389038	72389072	35
26	534	559	934	100%	chr2	+	15921279	15921304	26
26	534	561	934	96.5%	chr16	+	18807106	18807133	28
26	536	561	934	100%	chr1	+	50089666	50089691	26
25	537	561	934	100%	chr2	-	42339287	42339311	25
25	529	557	934	80.8%	chr2	-	1669686	1669711	26
25	537	561	934	100%	chr15	-	102283693	102283717	25
25	537	561	934	100%	chr1	-	203275933	203275957	25
25	537	561	934	100%	chr1	-	154371367	154371391	25
25	537	561	934	100%	chr1	-	117960313	117960337	25
25	533	562	934	96.3%	chr1	-	26455277	26455308	32
25	537	561	934	100%	chr1	-	11094007	11094031	25
25	537	561	934	100%	chr1	+	205300156	205300180	25
25	712	737	934	100%	chr1	+	190677893	190677925	33
25	537	561	934	100%	chr1	+	92035145	92035169	25
25	537	561	934	100%	chr1	+	78716735	78716759	25
25	537	561	934	100%	chr1	+	10215135	10215159	25
25	537	561	934	100%	chr1	+	10254500	10254524	25
24	534	559	934	96.2%	chr2	-	23568876	23568901	26
24	537	560	934	100%	chr1	-	42647484	42647507	24
24	536	561	934	96.2%	chr6	+	1242639	1242664	26
23	537	559	934	100%	chr2	-	16336835	16336857	23
23	537	561	934	96%	chr11	-	5384672	5384696	25
23	537	561	934	96%	chr17	+	71708490	71708514	25
23	534	556	934	100%	chr1	+	12645138	12645160	23
22	535	556	934	100%	chr1	-	24411385	24411406	22
22	712	735	934	87%	chr1	+	232146809	232146831	23
22	537	558	934	100%	chr1	+	200194470	200194491	22
21	537	557	934	100%	chr2	-	51221387	51221407	21
21	572	592	934	100%	chrX	+	21894011	21894031	21
20	537	556	934	100%	chr2	-	16493593	16493612	20
20	537	556	934	100%	chr1	-	90872482	90872501	20
20	716	737	934	95.5%	chr1	-	49153801	49153822	22

20	537	556	934	100%	chr1	-	44557623	44557642	20
20	537	556	934	100%	chr1	+	227311550	227311569	20

33) *When the author deleted the enhancer for functional test, more rationale about the part of the enhancer, the TFBS motifs contained etc would be important. I am curious if and how the ATAC-seq data was used for designing these experiments.*

Response:

We thank Reviewer2 for the comment. To clarify this point we have built a figure, containing H3K27ac ChIP-seq signal, ATAC-seq signal, TFBS predictions (HOMER and JASPAR), and the generated zP.E3 deletions (Supplementary Fig.9a). Additionally, to address point 4 by Reviewer2, we have now included a new deletion (Deletion2) in the revised version of the manuscript, which is also annotated in Supplementary Fig.9a. In summary, the predicted Foxa2 and Pdx1 motifs are lost in Deletion1 mutants but not in Deletion2 mutants. Additionally, in Deletion1 most of the ATAC-seq peak is deleted, while in Deletion2 most of the ATAC peak remains intact. These differences in Deletion 1 and 2 correlate very well with the different resulting phenotypes, in which Deletion1 mutants display clear pancreatic abnormalities while Deletion2 mutants show mostly normal pancreas development, with no significant differences regarding the size of the pancreas, when comparing with controls (Supplementary Fig.9d). These results are now included in the revised version of the manuscript (Supplementary Fig.9).

34) *In the zebrafish zPtf1aE3 deletion line, are the Foxa2 and Pdx1 binding sites still present? The author used two pairs of gRNAs to generate zebrafish mutants. How many different alleles were recovered? Are the phenotypes consistent across independent lines?*

Response:

Regarding Foxa2 and Pdx1 predicted binding sites, they are not present in the Deletion1 homozygous mutant fish. We have added Supplementary Fig.9a to the revised version of the manuscript to clarify this point. This point was also discussed in more detail in point 33, raised by Reviewer2.

Regarding the CRISPR-Cas9-mediated deletion strategy, we first tested two pairs of sgRNAs (sgPair1 and sgPair2) targeting the enhancer to generate mutant F0 larvae. As depicted in Supplementary Fig.8, this strategy resulted in a pool of deletion alleles of varying lengths in

somatic cells (Supplementary Fig. 8a). We also verified that this pool included deletions spanning the predicted Foxa2 and Pdx1 binding sites (Supplementary Fig. 8a-b), and the F0 larvae displayed reduced pancreatic area at 8dpf compared to the controls (Supplementary Fig. 8c-e). This illustrates how multiple independent deletions of the enhancer can result in similar pancreatic phenotypes.

Furthermore, we have now generated a new deletion for zP.E3, describing in the revised version of the manuscript a total of 2 zebrafish mutant lines: Deletion1 (sgPair1) and Deletion2 (sgPair2) (Supplementary Fig. 9a). Homozygous Deletion1 resulted in reduced area of the pancreatic progenitor domain at 48hpf, and pancreatic agenesis phenotypes at 9dpf (Fig.5c-g) and 12dpf (Supplementary Fig.9c). Conversely, Deletion2 did not produce observable pancreatic phenotypes, neither in homozygous larvae, nor in 10dpf transheterozygous larvae containing both deletion alleles (Deletion1 and Deletion2) (figure below). These last observations correlate with the location of the different deletions, with Deletion1 overlapping with the predicted Foxa2 and Pdx1 binding sites and the ATAC-seq peak, while the adjacent Deletion2 leaves this sequence largely intact. These results suggest that the functional core of the enhancer is within Deletion1. We have now added the results obtained from Deletion 2 in the revised version of the manuscript.

Fig [Reviewer2_point34]. Normalized whole pancreas area of Tg(ela:mCherry) Deletion1 heterozygous (wt/del1) and Deletion1/Deletion2 transheterozygous larvae (del1/del2) at 10dpf. Individual values were normalized to the mean of the control group (wt/del1). Unpaired student's t-test (two-tailed), p-values<0.05 were considered significant. ns, non-significant.

35) Enhancers regulate tissue specific gene expression. Are the enhancers tested restricted to pancreas even though some of the genes they regulate are not pancreas specific? Are they activated elsewhere? For example, *arid1ab* could be broadly expressed in many tissues.

Response:

We thank Reviewer2 for the interesting question. Taking into account that approximately half of the predicted pancreatic enhancers (7638 out of 14753; Fig.1d) show epigenetic marks associated to enhancer activity in early embryonic stages, we consider that the existence of pancreatic enhancers also active in other tissues may be quite prevalent. When analyzing specifically the zA.E3 enhancers (zA.E3, now renamed zA.E4 in the revised version of the manuscript), we find it within the DevE cluster, suggesting that its activity may be broadly active in many different tissues (differentiated and/or undifferentiated). Motivated by Reviewer2's question, and to better clarify this point, we performed *in vivo* enhancer reporter assays for zA.E4, searching expression in a total of 5 tissues in 11 days post fertilization (dpf) F0 zebrafish larvae and, aside from the pancreas, we also found a consistent expression of the reporter gene in the digestive tract (specifically in the hindgut and cloacal epithelium). Expression was also detected in the heart, although less consistent than in the hindgut, and in the cloacal epithelium. These results are depicted below for Reviewer2's appreciation. In addition, we have further explored the zebrafish transgenic line for the zP.E3 enhancer, asking if reporter gene is restricted to pancreatic tissue in 17dpf larvae, 2 month juveniles and 2 years old adults. We found that the vast majority of the GFP expression is located in the pancreatic tissues in all analyzed time points, suggesting that zP.E3 is quite tissue specific for pancreatic tissues. These results are in agreement with the presence of the zP.E3 enhancer within the PsE cluster of enhancers. This detailed description of the zP.E3 enhancer is now included in Supplementary Fig.10 of the revised version of the manuscript.

Figure [reviewer2_point35]: zA.E4 enhancer drives GFP expression in tissues of the digestive tract. Above: Percentage of F0 zebrafish larvae (11 dpf) showing GFP expression following *in vivo* transient transgenesis reporter assays for the zA.E4 sequence in comparison with the empty Z48 vector (control). A total of 5 tissues were observed; the exocrine pancreas (empty Z48 n=53, zA.E4 n=28), the heart (control n=8, zA.E4 n=8), the liver (control n=8, zA.E4 n=8), the hindgut (control n=8, zA.E4 n=8), and the cloaca (control n=9, zA.E4 n=8). Values are represented as percentages and compared by Chi-square test (p -values < 0.05 were considered significant). Below: Representative confocal image of the *in vivo* transient transgenesis reporter assays showing expression of GFP (green) in 11 dpf zebrafish hindgut and cloacal epithelium, in comparison with

the control. GFP expression can also be seen in the cells of the pronephric duct. Nuclei were stained with DAPI (blue). Images were captured with a Leica SP5II confocal microscope.

Reviewer3

In this paper the authors perform a combined analysis using ATAC-seq, ChIP-seq, 4C-seq and HiChIP-seq in zebrafish and human pancreatic cells to identify interspecies functionally equivalent cis-regulatory elements. This is an innovative work that utilizes the power of inter species comparison of chromatin traits and conservation of them to identify regulatory elements regulating pancreatic genes linked to human diseases. Overall the study is robust and well presented and I would recommend it for publication in Nature Communications.

Minor comments.

1) *There are a few places in the manuscript that the authors refer to interaction points of chromatin. For instance “To bypass these limitations, we profiled the chromatin state of zebrafish pancreas cells and interaction points” in lines 61,62. I assume the authors refer to chromatin interactions but this should be clarified in several parts of the text.*

Response:

We thank Reviewer3’s comment, which is entirely correct, as the text was intended to refer to chromatin interactions. We have therefore altered the manuscript (main text, Material and Methods and Figure captions) to clarify this point, in particular:

The sentence pointed out by the reviewer now reads: “To bypass these limitations, we profiled the chromatin state of zebrafish pancreas cells and chromatin interaction points.”

Accordingly, modifications were also done in the following sentences: “we explored the chromatin state and chromatin interaction points of zebrafish whole pancreas, to gather information about endocrine and exocrine cells, and to compare it to human data sets...” [Results section: Zebrafish putative pancreatic enhancers share developmental roles]; “To improve the enhancer to gene association, we used H3K4me3 HiChIP to detect chromatin interactions between active promoters and putative enhancers in the zebrafish adult pancreas” [Results section:

Functional similarities between human and zebrafish pancreatic enhancers]; “HiChIP analysis from paired-end fastq files to pairs of interacting chromatin fragments were performed using a custom python script...” [Methods section: HiChIP-seq analysis]; and “heat map for chromatin interactions with gata6 promoter detected by HiChIP for H3K4me3 from whole pancreas (below)” [Fig.2b caption]. Also, to facilitate the identification of the changes performed in the revised version of the manuscript addressing each specific point raised by Reviewer3, we have annotated the respective point in a “comment” in the manuscript version that contains “track changes”. E.g. the changes for this particular point are annotated as “Reviewer3, point 1”.

2) In lines 103 to 113 the authors state “We found that PsE-associated genes have a higher average expression in a variety of pancreatic cell types when comparing to all transcribed genes, contrasting with transcription in the muscle (Fig.2a, Table S3). Similar results were obtained when analysing genes associated to the remaining clusters of pancreatic enhancers (PsEs+DevE, DevE and C1-4; Fig.S1d). Performing a similar assay using the transcriptome of whole zebrafish embryos from 18 developmental stages 20, genes associated to DevE have shown an increased average expression comparing to all transcribed genes, with a similar dynamic to the enhancer activation during development (Fig.2b; Fig.S1e). These results suggest that DevE enhancers control gene expression in the adult differentiated pancreas and during development.”

- This paragraph is very confusing and hard to understand. The authors need to clarify it. Particular points are:

Response:

We have addressed this point in the specific points raised below.

3) PsE associated genes have a higher expression in pancreatic cell types compared to all transcribed genes, contrasting with the muscle. Why do the authors compare the PsE associated genes transcription with the muscle? Are the authors measuring the expression of the PsE associated genes in the muscle as well? This is unclear in the text.

Response:

We thank Reviewer3 for the comment and we agree with the Reviewer3's assessment that the use of the muscle as our control tissue was not clear. In the current work we profiled the regulatory landscape of adult zebrafish pancreas using known signatures of active enhancers, therefore

predicting pancreatic enhancers active in the adult zebrafish pancreas. Additionally, we performed HiChIP for H3K4me3 to identify gene-enhancer chromatin interaction points to predict the enhancer's target genes. If these predictions are correct, namely the identified adult pancreatic enhancers and the genes they are interacting with, two reasonable outcomes should be expected: the first is that, in pancreatic tissues, the average expression level of the genes interacting with adult pancreatic enhancers should be higher than all genes in the genome. This is so because this list of genes should be enriched for genes expressed in the pancreas, and their expression should be high as a consequence of the activity of the associated enhancers. The second expected outcome is that the list of genes associated with adult pancreas specific enhancers should not have higher expression levels than all genes in the genome in tissues that are not closely related to the pancreas. This is so because the list of genes associated with adult pancreas specific enhancers should not be biased to be expressed in the unrelated tissue. To assess this second expected outcome, in the current experiment, we have used the transcriptome of a mesoderm derived tissue, the muscle. As expected, we found that the average expression in the muscle of the genes associated with adult pancreas specific enhancers is lower than the average expression of all genes, suggesting that these genes are not biased to be expressed in the muscle, but they are biased to have lower expression levels in this tissue. These results suggest that most likely the pool of the genes associated with adult pancreas specific enhancers contain several genes that may be pancreas specific and, therefore inactive, in the muscle. These analyses are important to validate our datasets of adult pancreatic enhancers and their association to target genes. Following Reviewer3's advice, we have now improved the highlighted sentence in the revised version of the manuscript, which now reads: "We found that, compared to all genes, PsE-associated genes have a higher average expression in multiple pancreatic cell types (Fig.2a, Supplementary Table 3b). As expected, these expression results contrast with the lower average expression level of the PsE-associated genes compared to all genes in a distantly related control tissue as the muscle (Fig.2a, Supplementary Table 3b)."

4) *What do the authors mean with "the remaining clusters of pancreatic enhancers" and what does the PsE+DevE category refers to? This should be better explained in the main text.*

Response:

We thank Reviewer3 for this comment and agree that the original sentence should be better explained. In the original text, the phrase "the remaining clusters of pancreatic enhancers" was

intended to include all the other subclusters of pancreatic enhancers that have been identified in the manuscript. The total dataset of zebrafish adult pancreatic enhancers has been divided into PsE and DevE. In addition, the DevE cluster has been subdivided into C1 to C4. Therefore, all the identified clusters and subclusters are: PsE, DevE, C1, C2, C3 and C4. We have now clarified this point in the revised version of the manuscript. Regarding the PsE+DevE category, it refers to “The total dataset” of zebrafish adult pancreatic enhancers. Because we have not used a specific nomenclature for this dataset, we considered that a representation of the sum of PsE and DevE could be an intuitive way to represent this dataset, however we agree that, using this nomenclature without previous clarification may be misleading and may confuse the reader. We have improved the current sentence, which now reads: “Similar results were obtained when analysing genes associated to the other identified clusters of pancreatic enhancers, specifically, DevE, C1 to C4 and the total dataset of pancreatic enhancers altogether (PsEs+DevE; Supplementary Fig. 2c), showing higher expression levels in differentiated pancreatic cells, and lower expression levels in the muscle, when compared to all transcribed genes.”.

5) “...genes associated to DevE have shown an increased average expression comparing to all transcribed genes, with a similar dynamic to the enhancer activation during development (Fig.2b; Fig.S1e). These results suggest that DevE enhancers control gene expression in the adult differentiated pancreas and during development.” What do the authors mean by enhancer activation? Clarify in the main text. PsE are the one’s driving gene expression in the adult pancreas preferentially over DvE correct? The last sentence of the paragraph is confusing.

Response:

We agree that we should make this sentence clearer, and we thank Reviewer3 for bringing this point to our attention. Regarding the meaning of “enhancer activation” in the context of the highlighted sentence, we were referring to the differences in H3K27ac signal in the 4 different developmental time points analyzed, which we inferred to be an indicator of “enhancer activation”. We acknowledge that this may be misleading, and we have now improved this segment of the text in the revised version of the manuscript. In detail: After the identification of the total dataset of zebrafish adult pancreatic enhancers we explored if these adult pancreatic enhancers could be very specific of pancreatic tissues or could be found active more broadly. We addressed this by analysing the presence of H3K27ac marks in whole embryos, as discussed in detail to address the

questions raised by Reviewer1 point3 and Reviewer2 point1. We found that out of the 14753 identified adult pancreatic enhancers, 7255 also presented H3K27ac mark in whole embryos in at least 1 of the 4 developmental stages profiled for H3K27ac. This result indicates that approximately half of the adult pancreatic enhancers are also active in other cell types, rather than being exclusively active in pancreatic cells. Next, we wanted to better understand the nature of the enhancers active in whole embryos. Ubiquitous enhancers are expected to show a constant presence of H3K27ac marks while tissue-specific, in other tissues than the pancreas, or developmental enhancers, should present variable H3K27ac marks in the 4 developmental time points analyzed. To understand this, we applied a self-clustering algorithm to cluster the different sequences in regard to the H3K27ac availability in the 4 developmental time points (Fig.1d). We found 4 different clusters, C1 to C4, that presented different profiles in the 4 developmental time points analyzed. C1 and C4 showed a constant presence of H3K27ac in the 4 developmental time points, varying only in their total amount (high in C1 and low in C4) while C2 and C3 presented a progressive decrease (C2) and increase (C3) of H3K27ac presence in the 4 developmental time points. We inferred that the changes in H3K27ac signal corresponded to a variation in the activity of enhancers, which, as explained above, we have now corrected in the revised version of the manuscript.

Regarding the second doubt presented by Reviewer3 “PsE are the one’s driving gene expression in the adult pancreas preferentially over DevE correct?”. This question can be addressed by analyzing the average expression levels of the DevE associated genes (Supplementary Fig.2c, panel “DevE”), compared to the PsE associated genes (Fig.2a). We made available these two graphs below to facilitate a side-by-side comparison. Indeed, we can observe that the average levels of expression of the genes associated with PsE are higher than the ones associated with DevE in all the pancreatic cell types analyzed. One possible explanation for these results, as suggested by Reviewer3, is that PsE drives gene expression in the adult pancreas preferentially over DevE. To further test this possibility, we analyzed the average expression levels of the genes associated with DevE and PsE for genes that contain enhancers of only one of these categories, DevE alone or PsE alone, in their landscapes. In this analysis, the differences between DevE and PsE become even more striking, further supporting Reviewer3’s hypothesis that PsE drives gene expression in the adult pancreas preferentially over DevE.

Figure [reviewer3_point5] - Average gene expression determined by RNA-seq from pancreatic cells (acinar, duct, endocrine), whole pancreas and muscle (control) for all genes (AllG) and for genes interacting with pancreas specific enhancers (PsE, upper-left plot) and developmental enhancers (DevE, upper-right plot) as detected by HC in adult zebrafish pancreas. Average gene expression determined by RNA-seq from pancreatic cells (acinar, duct, endocrine), whole pancreas and muscle (control) for all genes (AllG) and for genes interacting with pancreas specific enhancers (PsE only, down-left plot) or developmental enhancers (DevE only, down-right plot) as detected by HC in adult zebrafish pancreas.

6) The authors state in lines 115-117, “From the first set, we have found that 5 out of 10 tested sequences ($H3K27ac: -\log_{10}(p\text{-value}) \geq 35$) are pancreatic enhancers (50%; Fig.2c-d and Table S4).” Can the authors comment on the fact that 50% of $H3K27ac$ peaks are not pancreatic enhancers and if these elements have other features (accessibility, chromatin contacts) that could explain this phenomenon? Have they checked for other enhancer features as active transcription

or p300 abundance? It would be interesting to include a comment on this matter somewhere in the paper as the field more or less assumes that a peak of H3K27ac will be an enhancer or promoter..

Response:

We thank Reviewer3 for raising this point and we want to clarify that we only used H3K27ac and ATAC-seq data to identify putative pancreatic enhancers. One considerable limitation of the zebrafish as a model, regarding ChIP-seq for some classical markers for enhancers, such as p300, is the lack of optimal ChIP-seq grade antibodies that can recognize the zebrafish proteins. This is the particular case for p300 that, to our knowledge, no antibody has not yet been described to recognize p300a (curated information summarized in <https://zfin.org/ZDB-GENE-080403-16#transcripts>) or p300b (curated information summarized in <https://zfin.org/ZDB-GENE-080403-15#transcripts>). Regarding the enhancer reporter assays, we would like to state that during the revision process we have detected that the graph in Fig.2c did not correctly represent the *p-value*<0.05 for sequence E6, despite having been correctly annotated in the original Supplementary Table 4. The description of the results in the main text was based on the interpretation of the graph in Fig.2c and for this reason we have described 50% of the tested sequences to have shown to be pancreatic enhancers, which is incorrect. The correct value is 60% and this is now corrected in the revised version of the manuscript. We would like to highlight that sequences E7 to E10 in Fig.2c, although not having as consistent results as sequences E1 to E6, present a higher percentage of larvae with GFP expression in the pancreas than the negative control (E1 to E6: 5.56 to 16.67% versus Negative Control: 3.49%). One possible interpretation for these results is that the *in vivo* enhancer reporter assay has some limitations in sensibility regarding the validation of pancreatic enhancers. Nevertheless, the clear advantages associated with the use of *in vivo* enhancer reporter assays (e.g. representation of multiple cellular types) made this our system of choice to use in this work.

Histone modifications and chromatin accessibility, as we used in this work, have been described as effective tools for identifying enhancers (PMID: 32728240, PMID: 22763441, PMID: 24360275). Previous studies showed that regions with stronger H3K27ac validated more frequently in transgenic reporter assays and the percentage of validated enhancers found in our work (60%) is similar to the percentages found in other studies: e.g. in PMID: 32728240 60% of the sequences in the highest H3K27ac enrichment rank tier displayed reporter expression in the expected tissue and in PMID: 24360275 63-67% of the tested sequences had enhancer activity in

the predicted tissue at the predicted timepoint (based on H3K27 enrichment). We have now modified the highlighted sentence to reflect this, as suggested by Reviewer3 and to address a point raised by Reviewer2 (point 20).

7) It would be important for the authors to discuss why if they can detect TFBS enriched at the enhancers in both human and zebrafish the sequence conservation seems to be low. Also in this regard it would be interesting for the author's to comment what features are preserved then, if not sequence, between the enhancers across species.

Response:

We greatly appreciate this very interesting comment by Reviewer3. Indeed, as Reviewer3 pointed out, in the current work we found that, although TFBS are enriched at the enhancers in both human and zebrafish, sequence conservation seems to be low. These two findings are apparently incoherent since TFBS are sequence dependent, however there are several important points to consider that can unify these two apparent irreconcilable observations:

1) Regarding what is known about TFBS, the consensus is usually very short with a considerable degree of nucleotide variation. This potential nucleotide variation for TFBS for the same TF is currently broadly represented by position weight matrices that define the probability of finding a nucleotide in a certain consensus for a specific TF. This permission in terms of sequence variability impacts negatively in the identification of strict sequence conservation.

2) Enhancers are frequent sequences that combine the binding of several TFs. Several models have been proposed regarding the rules for TF binding within enhancers. One such model is the *billboard model* (reviewed in PMID: 15696541, and summarized in PMID: 27968730), that proposes that the position and organization of TFBSs within an enhancer sequence is flexible. Therefore, within this conceptual framework, during evolution enhancers can suffer a reshuffle in the position of the different TFBS maintaining their function. This potential flexibility could also impact negatively in the identification of strict sequence conservation in enhancers of divergent species.

3) Sequences with ancestral unrelated functions can be recruited to the regulatory landscape of genes to act as enhancers. A striking example of this type of event is the repurpose of coding sequences to work as enhancers (PMID: 27863239). These types of events open the possibility of

the recruitment of functionally redundant enhancers that might replace the ancestral ones, another mechanism that could explain the lack of sequence conservation in functionally equivalent enhancers in divergent species.

These 3 examples, that represent the mechanisms that can operate together during evolution, nucleotide alterations within TFBS; reshuffling of TFBSs within enhancers; and substitution of total sequences of enhancers by acquisition of redundant enhancers in the same regulatory landscape, can help to explain the observed low sequence conservation with a similar TFBS code in pancreatic enhancers.

Following Reviewer3's suggestion, we have now improved the discussion of the revised version of the manuscript, in line with the above explanation.

REVIEWER COMMENTS

Reviewer #1 (Remarks to the Author):

I would like to acknowledge the authors for addressing my comments in detail and by improving the manuscript substantially during the revision period.

However, I still have a few concerns on the wording and level of claim in some sections, which I list below:

1. Title – the authors are not necessarily uncovering disease-related enhancers (e.g. as the authors clarify, the ARID1A enhancer does not contain known PDAC risk variants) thus I suggest the authors re-edit the title to be more accurate in relation with the datasets generated. For example, "(...) to uncover and investigate disease-relevant enhancers" would be more appropriate.

2. There are some scientific inaccuracies in the abstract, which I invite the authors to revise once more. In "Most alleles uncovered by genome-wide association studies with associations to pancreatic diseases such as diabetes", consider rephrasing by "Most alleles uncovered by genome-wide association studies of diseases and traits involving pancreatic dysfunction such as diabetes" (diabetes is a multi-organ disease not restricted to the pancreas).

"(...) we identified a zebrafish ptf1a distal enhancer whose deletion causes pancreatic agenesis, a phenotype previously described in human patients that display mutations in a distal enhancer of PTF1A, demonstrating the causality of this condition in vivo." – The fact that multiple humans who have homozygous deletion/point mutation of this enhancer have ablation of pancreas formation (Weedon et al. 2014, Gabbay et al. 2017, Evliyaoğlu et al. 2018, Dermirbilek et al. 2020) is a much better proof of causality of the condition in vivo. Please revise the abstract and elsewhere in the manuscript where there a claim of demonstrating causality. Proof of this is the fact that the enhancer sequence has already been added onto the panel of routinely analysed regions in individuals with suspected monogenic diabetes: <https://www.diabetesgenes.org/tests-for-diabetes-subtypes/targeted-next-generation-sequencing-analysis-of-45-monogenic-diabetes-genes/>.

In "(...) contributing to the prediction of new disease-associated enhancers and their role in human disease." – the term association implies some statistical test (such as a GWAS), which is not necessarily the case of what the authors are identifying. Thus replacing "associated" by "relevant" may be a more nuanced and correct way of referring to this set of CREs. This comment is also relevant for the last sentence of the Introduction.

3. Introduction – The sentence "However, in vivo evidence of the role of CREs' mutations in the development of pancreatic diseases is still scarce" is slightly misleading. The authors should either cite the papers where the PTF1A enhancer mutations have been reported or instead say "evidence from in vivo models".

4. Introduction – in "strongly associated with pancreatic agenesis 35" the authors are advised to rephrase "strongly associated" with "leads to" pancreatic agenesis/hypoplasia", as association seems to refer to a GWAS. Citing the additional more recent papers that reported mutations in this enhancer leading to pancreatic agenesis (see point 2) is also advised.

5. Page 11 – the authors refer to a potential DNA damage response interference due to ablation of the enhancer from ductal cells. While I agree with the statement, it will not be obvious to a reader from outside the PDAC field. The authors are advised to expand a little the justification of choosing ARID1A for modelling (above in the same section) referring to its properties/role in DNA damage response. This should be supported by an appropriate citation, such as PMID: 26069190.

6. Discussion – “Although GWAS make invaluable contributions in this field, these studies are limited by the size of analysed populations and by the frequency of alleles within these populations”. This point is highly controversial and the authors offer no basis to support claims that their approach may fill the gaps of GWAS and other approaches to investigate common disease. There is not a single GWAS variant tested in this manuscript. Particularly in the current setting where GWAS with over a million participants are now a reality (PMID: 32541925). The authors are invited to revise this section and tone down their claims of potential uses of their approach.

Inês Cebola

Reviewer #2 (Remarks to the Author):

We thank the authors for their thorough and thoughtful response to our comments. I still recommend looking into grammatical issues and sentences where the meaning is unclear. I have indicated a few examples below along with a final question/clarification about the motif analysis.

Minor clarification/grammar comments:

- e.g. page 4 line 4: “, and to compare it” would read better as “and compared it”
- grammatical issues in “However, in vivo evidence of the role of CREs’ mutations in the development of pancreatic diseases is still scarce”
- for consistency 36,5 to 92,1 should be 36.5 to 92.1. There are other examples like this in main text and supplementary figures.
- pg 8 “reproducible pancreatic enhancer activity ” should be “reproducible evidence of pancreatic enhancer activity ”
- “Overall, these results suggest that pancreatic enhancer function is not a strong constraint for sequence conservation.” This sentence is still very confusing and ideally it can be restated.
- figure 3b legend does not explain the two groups for each enhancer category. I assume this has to do with H3K27ac signal. This should be made clear.
- p11 - “To better address this hypothesis, we ” This is a new section of the paper and which hypothesis is being referred to is not clear.
- p11 - “enhancer in ARID1A expression” should be “on ARID1A”
- bottom of page 11. Should ARID1A be italicized?
- page 12 “demostrated” should be demonstrated.
- page 15 Ptf1a should be ptf1a if referring to zebrafish.

The description of the motif enrichment strategies and comparisons are still a bit challenging to follow as presented. In particular the rationale on page 10:

"As shown above, human and zebrafish pancreatic enhancers are enriched for many shared TFBS, therefore it is reasonable to expect that many of these TFBS correspond to known pancreas TFs. To test this hypothesis we have selected 25 motifs from TFs known to be required for pancreatic function and development and found that the majority of these were within the ZP,HP overlapping datasets, regardless of the compared groups (Supplementary Fig.4d-f)."

To address the hypothesis it seems the most straightforward thing to do would be to test whether the shared TFBS identified are enriched for pancreatic TFs. But as written it sounds like the authors only talk about the presence of these pancreatic motifs in the different enhancer categories. I would assume that within any set of enhancers (H3K27ac regions) you would find these pancreas TF motifs, and that looking for their presence alone would not address this hypothesis referred to regarding their enrichment. I was expecting to see the authors report on the enrichments within different categories.

However, looking again at Supp fig 4d-f it seems like this may be what was done, as pvalues are given for Supp4 d, e, f. Is it fair to say that except for when the authors split the data up using the ventricle data, the ZP/HP regions were not significantly enriched for the pre-selected pancreas TFs? Separating enhancer categories using a different adult tissue could reasonably allow for a more refined pancreas-specific enhancer set. If so, it would make sense for the authors to acknowledge this in the main text when addressing their hypothesis.

- Is it necessary to correct for multiple testing if the same set of sequences are split different ways and tested for enrichments?

Reviewer #3 (Remarks to the Author):

The authors have answered my concerns in full. I recommend the paper for publication in Nature Communications

** See Nature Research's author and referees' website at www.nature.com/authors for information about policies, services and author benefits.

Point-by-point response to reviewers:

Reviewer1

I would like to acknowledge the authors for addressing my comments in detail and by improving the manuscript substantially during the revision period.

However, I still have a few concerns on the wording and level of claim in some sections, which I list below:

1) *Title – the authors are not necessarily uncovering disease-related enhancers (e.g. as the authors clarify, the ARID1A enhancer does not contain known PDAC risk variants) thus I suggest the authors re-edit the title to be more accurate in relation with the datasets generated. For example, “(...) to uncover and investigate disease-relevant enhancers” would be more appropriate.*

Response:

We thank Reviewer1 for the constructive comment. Following the suggestion of Reviewer1, we now improved the title of the manuscript to “Multidimensional chromatin profiling of zebrafish pancreas to uncover and investigate disease-relevant enhancers”.

To facilitate the identification of the changes performed in the revised version of the manuscript addressing each specific point raised by Reviewer1, we have annotated the respective point in a “comment” in the manuscript version that contains “track changes”. E.g. this particular change has been annotated as “Reviewer1, point 1”.

2) *There are some scientific inaccuracies in the abstract, which I invite the authors to revise once more.*

2.1) *In “Most alleles uncovered by genome-wide association studies with associations to pancreatic diseases such as diabetes”, consider rephrasing by “Most alleles uncovered by*

genome-wide association studies of diseases and traits involving pancreatic dysfunction such as diabetes” (diabetes is a multi-organ disease not restricted to the pancreas).

Response:

We thank Reviewer1 for raising this point. As suggested by Reviewer1, we have now rephrased this sentence to “Most alleles uncovered by genome-wide association studies of diseases and traits involving pancreatic dysfunction, such as diabetes, overlap with non-coding sequences of DNA, many containing epigenetic marks of cis-regulatory elements active in pancreatic cells.”.

2.2) *“(…) we identified a zebrafish *ptf1a* distal enhancer whose deletion causes pancreatic agenesis, a phenotype previously described in human patients that display mutations in a distal enhancer of *PTF1A*, demonstrating the causality of this condition *in vivo*.” – The fact that multiple humans who have homozygous deletion/point mutation of this enhancer have ablation of pancreas formation (Weedon et al. 2014, Gabbay et al. 2017, Evliyaoğlu et al. 2018, Dermirbilek et al. 2020) is a much better proof of causality of the condition *in vivo*. Please revise the abstract and elsewhere in the manuscript where there a claim of demonstrating causality. Proof of this is the fact that the enhancer sequence has already been added onto the panel of routinely analysed regions in individuals with suspected monogenic diabetes: <https://www.diabetesgenes.org/tests-for-diabetes-subtypes/targeted-next-generation-sequencing-analysis-of-45-monogenic-diabetes-genes/>.*

Response:

As suggested by Reviewer1, we have now improved the manuscript eliminating the claim of the demonstration of the causality of the pancreatic agenesis phenotype. In the example that Reviewer1 points out, we have changed the sentence from “we identified a zebrafish *ptf1a* distal enhancer whose deletion causes pancreatic agenesis, a phenotype described in human patients that display mutations in a distal enhancer of *PTF1A*, demonstrating the causality of this condition *in vivo*.” to “we identified a zebrafish *ptf1a* distal enhancer whose deletion causes pancreatic agenesis, a phenotype previously found to be induced by mutations in a distal enhancer of *PTF1A* in humans, further supporting the causality of this condition *in vivo*.”. We

have also reinforced the human causality statement in several points of the manuscript, annotated in a “Reviewer1, point2.2” comment in the manuscript version that contains “track changes”.

2.3) *In “(...) contributing to the prediction of new disease-associated enhancers and their role in human disease.” – the term association implies some statistical test (such as a GWAS), which is not necessarily the case of what the authors are identifying. Thus replacing “associated” by “relevant” may be a more nuanced and correct way of referring to this set of CREs. This comment is also relevant for the last sentence of the Introduction.*

Response:

We thank Reviewer1 for raising this point. We agree that the term “association” pointed out by Reviewer1 is not the most appropriate term, given the context, and we appreciate the suggested alternative and we altered the manuscript accordingly.

3) *Introduction – The sentence “However, in vivo evidence of the role of CREs’ mutations in the development of pancreatic diseases is still scarce” is slightly misleading. The authors should either cite the papers where the PTF1A enhancer mutations have been reported or instead say “evidence from in vivo models”.*

Response:

We thank Reviewer1 for raising this point. Following Reviewer1 suggestion, we have now changed this sentence to “However, evidence from *in vivo* models of the role of CREs’ mutations in the development of pancreatic diseases is still scarce^{21–23}.”

4) *Introduction – in “strongly associated with pancreatic agenesis 35” the authors are advised to rephrase “strongly associated” with “leads to” pancreatic agenesis/hypoplasia“, as*

association seems to refer to a GWAS. Citing the additional more recent papers that reported mutations in this enhancer leading to pancreatic agenesis (see point 2) is also advised.

Response:

We thank Reviewer1 for raising this point. To avoid misinterpretations regarding GWAS, we have followed Reviewer1's suggestion changing the highlighted sentence. We have also added references of three more recent papers reporting mutations in this enhancer leading to pancreatic agenesis. This sentence now reads: "Additionally, we explored the regulatory landscape of *PTF1A*, known to contain a human distal enhancer whose deletion leads to pancreatic agenesis/hypoplasia 35-38 and found a zebrafish distal *ptf1a* enhancer that contains similar regulatory information to its human counterpart."

5) Page 11 – the authors refer to a potential DNA damage response interference due to ablation of the enhancer from ductal cells. While I agree with the statement, it will not be obvious to a reader from outside the PDAC field. The authors are advised to expand a little the justification of choosing ARID1A for modelling (above in the same section) referring to its properties/role in DNA damage response. This should be supported by an appropriate citation, such as PMID: 26069190.

Response:

We thank Reviewer1 for giving us the opportunity to clarify this point and we agree that we should elaborate on the choice of *ARID1A*. *ARID1A* is a subunit of the SWI/SNF (also called BAF) chromatin remodeling complex, which modulates chromatin accessibility during processes such as transcription, DNA replication, and DNA repair. *ARID1A* is frequently mutated genes in several human cancers, including pancreatic cancer (~3.6% of pancreatic cancers) (PMID: 23644491). This overrepresentation of *ARID1A* mutations in cancer, is likely due to its key function in DNA double-strand breaks processing. As DNA double-strand breaks drive cancer development by creating genomic instability, DNA double-strand break repair genes are important tumor suppressors and their disruption is often associated with multiple cancers (PMID: 23644491, PMCID: PMC7004434). *ARID1A*, is recruited to DNA double-strand breaks

and facilitates/accelerates double-strand break end resection (or 5'-3' degradation), which, in turn is a crucial step to for damage repair by homologous recombination, hence suppressing tumorigenesis (PMID: 26069190). As suggested by Reviewer1, we have now added a very short description of *ARIDIA* in the context of pancreatic cancer, to improve the justification of the selection of this gene.

6) Discussion – “Although GWAS make invaluable contributions in this field, these studies are limited by the size of analysed populations and by the frequency of alleles within these populations”. This point is highly controversial and the authors offer no basis to support claims that their approach may fill the gaps of GWAS and other approaches to investigate common disease. There is not a single GWAS variant tested in this manuscript. Particularly in the current setting where GWAS with over a million participants are now a reality (PMID: 32541925). The authors are invited to revise this section and tone down their claims of potential uses of their approach.

Response:

We thank Reviewer1 for the opportunity to clarify this point. We agree that the way this sentence is formulated might be misleading. It was not our intention to say that all the current GWAS are limited by the size of the studied populations, which is not the case as for instances in the case of T2D studies, that as Reviewer1 has accurately pointed out, currently involves more than a million participants. Our intention was to say that the small size of the studied population and low frequency of trait associated alleles, as well as potential low phenotypic penetrance of some alleles, may limit the outcome of GWAS. To avoid misinterpretations and because it is not the aim of the current manuscript to explore the limitations of GWAS, we have now shortened this sentence, highlighting the fact that the approach we describe might be suitable to identify disease relevant CREs, a term suggested by Reviewer1 and that we agree to be more accurate and toned down.

Reviewer2

We thank the authors for their thorough and thoughtful response to our comments. I still recommend looking into grammatical issues and sentences where the meaning is unclear. I have indicated a few examples below along with a final question/clarification about the motif analysis.

1) Minor clarification/grammar comments:

- *e.g. page 4 line 4: “, and to compare it” would read better as “and compared it”*
- *grammatical issues in “However, in vivo evidence of the role of CREs’ mutations in the development of pancreatic diseases is still scarce”*
- *for consistency 36,5 to 92,1 should be 36.5 to 92.1. There are other examples like this in main text and supplementary figures.*
- *pg 8 “reproducible pancreatic enhancer activity ” should be “reproducible evidence of pancreatic enhancer activity ”*
- *“Overall, these results suggest that pancreatic enhancer function is not a strong constraint for sequence conservation.” This sentence is still very confusing and ideally it can be restated.*
- *figure 3b legend does not explain the two groups for each enhancer category. I assume this has to do with H3K27ac signal. This should be made clear.*
- p11 - “To better address this hypothesis, we ” This is a new section of the paper and which hypothesis is being referred to is not clear.*
- p11 - “enhancer in ARID1A expression” should be “on ARID1A”*
- *bottom of page 11. Should ARID1A be italicized?*
- *page 12 “demostrated” should be demonstrated.*

-page 15 Ptf1a should be ptf1a if referring to zebrafish.

Response:

We thank Reviewer2 for giving us the opportunity to correct these issues in the manuscript. We carefully read Reviewer2's suggestions and corrected the manuscript accordingly. Regarding the comment on page 11 about ARID1A italicization, since we are talking about ARID1A protein, it should not be italicized. In this sentence we have changed "expression" to "levels" to avoid a possible misinterpretation regarding gene expression.

To facilitate the identification of the changes performed in the revised version of the manuscript addressing each specific point raised by Reviewer2, we have annotated the respective point in a "comment" in the manuscript version that contains "track changes". E.g. these particular changes have been annotated as "Reviewer2, point 1".

2) The description of the motif enrichment strategies and comparisons are still a bit challenging to follow as presented. In particular the rationale on page 10:

"As shown above, human and zebrafish pancreatic enhancers are enriched for many shared TFBS, therefore it is reasonable to expect that many of these TFBS correspond to known pancreas TFs. To test this hypothesis we have selected 25 motifs from TFs known to be required for pancreatic function and development and found that the majority of these were within the ZP,HP overlapping datasets, regardless of the compared groups (Supplementary Fig.4d-f)."

To address the hypothesis it seems the most straightforward thing to do would be to test whether the shared TFBS identified are enriched for pancreatic TFs. But as written it sounds like the authors only talk about the presence of these pancreatic motifs in the different enhancer categories. I would assume that within any set of enhancers (H3K27ac regions) you would find these pancreas TF motifs, and that looking for their presence alone would not address this hypothesis referred to regarding their enrichment. I was expecting to see the authors report on the enrichments within different categories.

However, looking again at Supp fig 4d-f it seems like this may be what was done, as p-values are given for Supp4 d, e, f. Is it fair to say that except for when the authors split the data up using the ventricle data, the ZP/HP regions were not significantly enriched for the pre-selected pancreas TFs? Separating enhancer categories using a different adult tissue could reasonably allow for a more refined pancreas-specific enhancer set. If so, it would make sense for the authors to acknowledge this in the main text when addressing their hypothesis.

- Is it necessary to correct for multiple testing if the same set of sequences are split different ways and tested for enrichments?

Response:

We thank Reviewer2 for giving us the opportunity to clarify this point. We agree that the rational of the highlighted sentence in page 10 “*As shown above , regardless of the compared groups (Supplementary Fig.4d-f).*” is a bit challenging to follow as presented. We have now improved this sentence to: “*As shown above, human and zebrafish pancreatic enhancers are enriched for many shared TFBS, therefore it is reasonable to expect that many of these TFBS are from TFs known to have an important pancreatic function. To test this hypothesis, we have selected 25 TFs known to be required for pancreas function and development and calculated the distribution of the respective TFBS motifs within the previously identified enriched motifs described in Supplementary Table 3t. We found that the majority of the TFBS motifs from the pancreatic TFs were within the ZP,HP overlapping datasets, regardless of the compared groups (Supplementary Fig.4d-f).*”.

Indeed, we have performed the assay suggested by Reviewer2 “*To address the hypothesis it seems the most straightforward thing to do would be to test whether the shared TFBS identified are enriched for pancreatic TFs.*”. In summary, in this assay we describe the distribution of TFBS motifs of known pancreatic TFs in the groups of enriched TFBS motifs presented in Figure3g and Supplementary Figure4 b and c.

Regarding the second point raised by Reviewer2, “*Is it necessary to correct for multiple testing if the same set of sequences are split different ways and tested for enrichments?*” This type of multiple testing correction (e.g. Bonferroni) is usually applied in confirmatory studies

and the nature of this study is more exploratory. However, following the suggestion of the Reviewer2, we applied the Bonferroni correction in these comparisons. This correction did not change the statistical significance. Finally, we also corrected some inaccuracies that we have found during the revision process: the statistical test applied in this analysis was the Fisher exact test, however, in the legend of Supplementary Figure4 described the chi-square test; in Supplementary Figure 4f, the percentage of group V is 11.75% and not 6.67%.

Reviewer3

The authors have answered my concerns in full. I recommend the paper for publication in Nature Communications

Response:

We would like to earnestly thank Reviewer3 for her/his time, for providing valuable comments that led to a greatly improved manuscript, and for recommending our paper for publication in *Nature Communications*.

REVIEWERS' COMMENTS

Reviewer #1 (Remarks to the Author):

Once again I would like to thank the authors for improving the manuscript during the revision, addressing my concerns in full. I recommend the paper for publication in Nature Communications.

Reviewer #2 (Remarks to the Author):

Thank you for clarifying. I have no further comments.

** See Nature Research's author and referees' website at www.nature.com/authors for information about policies, services and author benefits

We are submitting a revised version of our manuscript entitled "Multidimensional chromatin profiling of zebrafish pancreas to uncover and investigate disease-relevant enhancers" (NCOMMS-20-35219C) for consideration in Nature Communications. We have addressed all Editorial Board comments and suggestions. Essentially, we addressed all minor textual changes and requests to the manuscript file, and we submitted a marked-up version clearly indicating the changes. We also added hyperlinks in all accession codes provided in the Data Availability Statement. We also linked our Gitlab to the repository Zenodo, in order to obtain a DOI and we cited the Gitlab repository in the Code Availability statement and in our reference list. We provided a figure exemplifying the flow cytometry gating strategy in the Supplementary Information and we provided complete source data for all figures included in the Supplementary Information. We hope you agree that our manuscript is now suitable for publication in Nature Communications.